# Acceleration via silver stepsize on Riemannian manifolds with applications to Wasserstein space

**Jiyoung Park**
Department of Statistics
Texas A&M University
wldyddl5510@tamu.edu

**Abhishek Roy**
Department of Statistics
Texas A&M University
abhishekroy@tamu.edu

**Jonathan W. Siegel**
Department of Mathematics
Texas A&M University
jwsiegel@tamu.edu

**Anirban Bhattacharya**
Department of Statistics
Texas A&M University
anirbanb@stat.tamu.edu

## Abstract

There is extensive literature on accelerating first-order optimization methods in an Euclidean setting. Under which conditions such acceleration is feasible in Riemannian optimization problems is an active area of research. Motivated by the recent success of silver stepsize methods in the Euclidean setting, we undertake a study of such algorithms in the Riemannian setting. We provide the new class of algorithms determined by the choice of vector transport that allows the silver stepsize acceleration on Riemannian manifolds for the function classes associated with the corresponding vector transport. As a core application, we show that our algorithm recovers the standard Wasserstein gradient descent on the 2-Wasserstein space and, as a result, provides the first provable accelerated gradient method for potential functional optimization problems in the Wasserstein space. In addition, we validate the numerical strength of the algorithm for standard benchmark tasks on the space of symmetric positive definite matrices.

## 1  Introduction

Consider the Riemannian optimization problem

$$\min_{x \in N} f(x), \tag{1.1}$$

where $N \subseteq M$ is a *geodesically convex* subset of a Riemannian manifold $M$, and $f : N \to \mathbb{R}$ is a continuously differentiable *geodesically convex* functional. A popular approach to solve (1.1) is via Riemannian gradient descent (RGD) [ZS16] given by,

$$x_{n+1} = \exp_{x_n} \left( -\eta_n \operatorname{Grad} f(x_n) \right), \tag{1.2}$$

where $\exp_x(\cdot)$ is the exponential map at $x$, $\eta_n$ is the stepsize at iteration $n$, and $\operatorname{Grad}$ denotes the Riemannian gradient. It is known that for geodesically convex and smooth functionals $f$, constant stepsize RGD has an $O(1/n)$ convergence rate as in Euclidean spaces [ZS16].

A natural follow-up question is whether one can find first-order algorithms that achieve an *accelerated* convergence rate. This is motivated by the success of accelerated first-order methods in Euclidean settings, most notably Nesterov's method [Nes83], which uses momentum to achieve an $O(1/n^2)$ rate for convex and smooth objectives. Extensive efforts have been made to achieve the same

39th Conference on Neural Information Processing Systems (NeurIPS 2025).

accelerated rate using similar acceleration in Riemannian optimization problems under various settings [LSC$^+$17, ZS18, AS20, Sie21, AOBL21, CB22, MR22, KY22, HMJG23]. However, these works typically rely on additional constraints, stronger assumptions, or modifications to the basic gradient descent update (1.2). For example, [LSC$^+$17] involves an intractable nonlinear operator. The analysis in [HMJG23] relies on a submanifold structure and establishes acceleration only in the asymptotic regime. All the other algorithms require both upper and lower sectional curvature bounds. We refer to [dST21] for a general survey of momentum-based acceleration methods, and to [KY22, Sections 1, 2] for Riemannian variants.

On the other hand, there is a line of work showing that, in the Euclidean case, an accelerated convergence rate is possible by using a carefully designed *dynamic stepsize schedule* without any modification to vanilla gradient descent. This idea goes back to [You53]; for quadratic functions, choosing $\eta_n$ to be Chebyshev stepsizes in gradient descent achieves the $O(1/n^2)$ rate. Generalizing this idea to general convex and smooth functions, [Alt18, AP24b, AP24c, BA24] introduced the *silver stepsize* schedule—a carefully designed stepsize sequence that guarantees an improved convergence rate of $O(1/n^{\log_2 \rho})$, where $\rho = 1 + \sqrt{2}$. While slower than the $O(1/n^2)$ rate of Nesterov's acceleration, this method significantly outperforms constant stepsize gradient descent and shows that standard gradient descent, with a carefully designed stepsize schedule, can achieve meaningful acceleration. Its further studies–including the optimality of the stepsize and generalization to arbitrary number of iterations–are active area of research in the field of optimization [Gri24, GSW24]. Motivated by the success of the dynamic stepsize schedule in the Euclidean case, in this work, we pose the following question:

> Is it possible to accelerate Riemannian gradient descent by using the dynamic stepsize schedule?

**Main contribution**    Towards addressing the above question, we make the following contributions.

1. We introduce a family of algorithms, *vector-transported Riemannian gradient descent* (VTRGD), parameterized by the choice of vector transport $\mathcal{VT}$. This framework enables silver step-size acceleration on Riemannian manifolds for function classes defined relative to $\mathcal{VT}$. To formalize these classes, we define the notions of $\mathcal{VT}$-geodesic convexity (resp. $L$-smoothness) with base point $b \in M$.

2. We show if a function is (i) $\mathcal{VT}$-geodesically $L$-smooth with base $b$, and (ii) $\mathcal{VT}$-geodesically convex with all base, then VTRGD with silver stepsize schedule achieves the *accelerated* convergence rate of $O(1/n^{\log_2 \rho})$, and the rate of $\exp(-O(n/\kappa^{\log_\rho 2}))$ when $f$ is in addition geodesically *strongly* convex with condition number $\kappa$. These rates match the corresponding rates in the Euclidean case, and provides the acceleration with minimal assumptions on the manifold. In particular, we avoid the curvature assumption and diameter assumption typically required for momentum-based accelerated RGDs.

3. We show when $\mathcal{VT} = \Gamma$, the *parallel transport*, then $\Gamma$-geodesic convexity and $\Gamma$-geodesic smoothness are satisfied for some non-trivial Riemannian optimization problems. In particular, we show our algorithm coincides with the classical Wasserstein gradient descent in 2-Wasserstein space, a space where previous accelerated algorithms fail. Hence, our method provides the first provable accelerated result for (usual) Wasserstein gradient descents, particularly for potential function optimization problem. We also provide numerical illustrations in the space of symmetric positive definite matrices.

## 2  Background

**Riemannian manifolds.**    In this section, we review basic concepts of Riemannian manifolds while deferring more rigorous details to Appendix A.1. At a point $x$ on a manifold $M$, tangent vectors are the velocity vectors of smooth curves on $M$ that pass through $x$. The tangent space $T_x M$ is the vector space consisting of all such tangent vectors at $x$. A Riemannian manifold is a manifold equipped

with an inner product $\langle \cdot, \cdot \rangle_x$ for each tangent space $T_x M$, called a Riemannian metric. For $x, y \in M$, the distance $d(x, y)$ is the infimum of the length of all piecewise continuously differentiable curves from $x$ to $y$. A Riemannian gradient of the differentiable function $f : M \to \mathbb{R}$ at $x$ is a tangent vector $\mathrm{Grad}\, f(x) \in T_x M$ satisfying $d_v f(x) = \langle \mathrm{Grad}\, f(x), v \rangle_x$ for all $v \in T_x M$. Here, $d_v f(x)$ is a directional derivative of $f$ at $x$ along the direction $v$. For $(x, v) \in TM$, where $TM := \coprod_{x \in M} T_x M$ denotes the tangent bundle, a smooth curve $\gamma_v : [0, 1] \to M$ with $\gamma_v(0) = x$ and $\gamma'_v(0) = v$ is called a (constant speed) geodesic if it has the locally minimum length with zero acceleration. The exponential map $\exp_x : T_x M \to M$ is a map defined by $\exp_x(v) = \gamma_v(1)$. $\exp_x(v)$ transports the point $x$ in the direction of the tangent vector $v$, following the geodesic $\gamma_v$. It is known that $\exp_x$ is a local diffeomorphism in some neighborhood $U$ of $0 \in T_x M$. Hence, $\exp_x$ allows the inverse on $U$, which is called the logarithmic map $\log_x : \exp_x(U) \to T_x M$. While the exponential and logarithmic maps are always locally well-defined, they may not be globally well-defined.

A parallel transport $\Gamma(\gamma)_{t_0}^{t_1} : T_{\gamma(t_0)} M \to T_{\gamma(t_1)} M$ is a way to transport a tangent vector along the curve $\gamma$ parallelly. If $\gamma$ is a geodesic curve such that $\gamma(0) = x, \gamma(1) = y$, then we simply denote $\Gamma(\gamma)_0^1$ as $\Gamma_x^y$, a (geodesic) parallel transport from $T_x M$ to $T_y M$. One can generalize the notion of the parallel transport by *vector transport* [AMS08, GGH$^+$21, WDPY24]. For any $x, y \in M$, a vector transport $\mathcal{VT}_x^y : T_x M \to T_y M$ is an operator which maps a tangent vector $v \in T_x M$ to another tangent space $T_y M$, satisfying $\mathcal{VT}_x^x = id$. Typical examples include the adjoint of the differential of the exponential map $\mathcal{VT}_x^y = (\mathrm{d}\exp_x)^*_{\log_x y}$ and parallel transports $\mathcal{VT}_x^y = \Gamma_x^y$.

Lastly, we introduce the notion of geodesic convexity and smoothness.

**Definition 2.1** (Geodesic convexity). *We say $N \subseteq M$ is a geodesically convex subset of $M$ if for all $x, y \in N$ there exists a geodesic $\gamma$ such that $\gamma(0) = x, \gamma(1) = y$, and $\gamma(t) \in N$ for all $t \in [0, 1]$. We say a differentiable function $f : N \to \mathbb{R}$ is geodesically $\alpha$-strongly convex if for all $x, y \in N$*

$$f(y) \geq f(x) + \langle \mathrm{Grad}\, f(x), \log_x y \rangle_x + \frac{\alpha}{2} d^2(x, y).$$

*If the above inequality holds with $\alpha = 0$, then $f$ is said to be geodesically convex.*

**Definition 2.2** (Geodesic smoothness). *We say $f$ is geodesically $L$-smooth if for all $x, y \in M$*

$$f(y) \leq f(x) + \langle \mathrm{Grad}\, f(x), \log_x y \rangle + \frac{L}{2} d^2(x, y).$$

Some literature use the $L$-Lipschitz continuity of the gradient function as the definition of geodesic $L$-smoothness which in fact implies Definition 2.2 (see Definition A.17 and Lemma A.18). However, we adopt the above definition used in [KY22, Section 3.1], as it is more directly related to our $\mathcal{VT}$ extension introduced in Section 3.

**Silver stepsize in Euclidean space**    In this section, we present the silver stepsize schedule [AP24c] for Euclidean optimization problem. Consider the problem (1.1) where $N \equiv \mathbb{R}^d$, and $f$ is convex and $L$-smooth. A standard approach is gradient descent, which updates via $x_{n+1} = x_n - \eta \nabla f(x_n)$ for a fixed stepsize $\eta$. In contrast, the silver stepsize schedule is a sequence of dynamic stepsizes $\{\eta_n\}_{n \in \mathbb{N}}$. For $n = 2^k - 1$ where $k \in \mathbb{N}$, $\{\eta_n\}_{n \in \mathbb{N}}$ is given by the following inductively constructed sequence:

$$\eta^{(k+1)} = [\eta^{(k)}, 1 + \rho^{k-1}, \eta^{(k)}], \tag{2.1}$$

where $\rho = 1 + \sqrt{2}$. We set $\eta_0 = \rho - 1$. For example, for $k = 1, 2, 3$, $\eta^{(k)}$ has the following form:

$$\eta^{(1)} = [\sqrt{2}], \quad \eta^{(2)} = [\sqrt{2}, 2, \sqrt{2}], \quad \eta^{(3)} = [\sqrt{2}, 2, \sqrt{2}, 2 + \sqrt{2}, \sqrt{2}, 2, \sqrt{2}].$$

In Euclidean optimization, the silver stepsize was recently shown to improve the convergence rate of the gradient descent from $O(1/n)$ to $O(1/n^{\log_2 \rho})$ [AP24c]. The philosophy of the silver stepsize extended to arbitrary $n$, with the name *long stepsize* [Gri24, GSW24]. For the brevity of the paper, for this work we focus on the silver stepsize.

# 3   Silver stepsize VTRGD: Assumptions and Preliminaries

In this section, we state the assumptions on the manifold and objective function required to solve problem (1.1) using silver stepsize RGD (1.2).

**Assumption 3.1** (Assumptions for Riemannian manifold)**.**

1. $M$ *is a complete Riemannian manifold,* i.e., *any two points are connected by some geodesic.*

2. $N \subseteq M$ *is open, geodesically convex subset.*

3. *Exponential maps and logarithmic maps are all well-defined and computationally tractable on $N$.*

**Assumption 3.2** (Assumptions on the objective)**.** *We make the following assumptions on $f : N \to \mathbb{R}$.*

1. $f$ *is geodesically convex and has a global minimizer $x_* \in N$.*

2. *All the iterates of our algorithms are well defined and remain inside $N$.*

3. *For a given linear vector transport $\mathcal{VT}$, There exists a constant $L > 0$ and $b \in N$ such that for all $x_i, x_j$ in the RGD trajectory, $i, j = 0, 1, 2, \cdots, *$,*

$$
\begin{aligned}
Q_{ij;b} := 2L(f(x_i) - f(x_j)) - 2L \left\langle \mathcal{VT}_{x_j}^b \operatorname{Grad} f(x_j), \log_b x_i - \log_b x_j \right\rangle_b \\
- \left\| \mathcal{VT}_{x_i}^b \operatorname{Grad} f(x_i) - \mathcal{VT}_{x_j}^b \operatorname{Grad} f(x_j) \right\|_b^2 \geq 0.
\end{aligned}
\tag{3.1}
$$

**Remark 3.3.** *Assumptions 3.1 and 3.2, excluding equation (3.1), are standard in the Riemannian optimization literature [AOBL21, KY22, HMJG23], and ensure well-behaved RGD iterates. While we additionally assume (3.1), we do not require the curvature bound or diameter bound on $N$ typically assumed in (momentum-based) RGD algorithms.*

Some comments on (3.1) are in order. In an Euclidean space, since all tangent spaces are the same, *i.e.*, $\mathcal{VT}_{x_i}^b = id$, (3.1) holds for any $x_i, x_j, b \in \mathbb{R}^d$ if and only if $f$ is convex and $L$-smooth [Nes14, Theorem 2.1.5]. However, on Riemannian manifolds, (3.1) does not directly match with standard geodesic convexity and smoothness. To provide the interpretation of (3.1) in Riemannian manifold as in Euclidean space, we introduce new notions of convexity and smoothness, which we dub $\mathcal{VT}$-*geodesic convexity (smoothness)*.

**Definition 3.4** ($\mathcal{VT}$-geodesic convexity)**.** *A functional $f : N \to \mathbb{R}$ is called $\mathcal{VT}$-geodesically convex with base $b \in M$ if for all $x, y \in N$, we have,*

$$
f(y) \geq f(x) + \left\langle \mathcal{VT}_x^b \operatorname{Grad} f(x), \log_b y - \log_b x \right\rangle_b.
\tag{3.2}
$$

$f$ *is called $\mathcal{VT}$-geodesically convex if (3.2) holds for all $b \in N$.*

**Definition 3.5** ($\mathcal{VT}$-geodesic $L$-smooth)**.** *A functional $f : M \to \mathbb{R}$ is called $\mathcal{VT}$-geodesically $L$-smooth with base $b \in M$ if for all $x, y \in M$ we have,*

$$
f(y) \leq f(x) + \left\langle \mathcal{VT}_x^b \operatorname{Grad} f(x), \log_b y - \log_b x \right\rangle_b + \frac{L}{2} \left\| \log_b y - \log_b x \right\|_b^2.
\tag{3.3}
$$

$f$ *is called $\mathcal{VT}$-geodesically $L$-smooth if (3.3) holds for all $b \in M$.*

We interpret $\mathcal{VT}$-geodesic convexity (*resp.* $L$-smoothness) for some representative choices of $\mathcal{VT}$. One canonical example is $\mathcal{VT}_b^x = (d\exp_b)_{\log_b x}^*$. For such $\mathcal{VT}$, the $\mathcal{VT}$-geodesic convexity (*resp.* $L$-smoothness) with base $b$ is equivalent to the (Euclidean) convexity (*resp.* $L$-smoothness) of the function $F(v) := f(\exp_b(v))$ defined on the tangent space. Another natural choice of $\mathcal{VT}$ is the parallel transport $\Gamma$. In the optimal transport literature, $\Gamma$-geodesic convexity is already a popular concept, namely *generalized geodesic convexity* [AGS08, San14, SKL20, DBCS23] which is broadly applied for studying the proximal operator, $\Gamma$-convergence in 2-Wasserstein space, and Riemannian

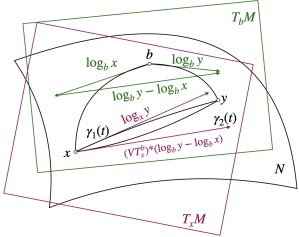

Figure 1: Geometric illustration of $\mathcal{VT}$-geodesic convexity. Usual geodesic convexity (*resp. L*-smoothness) means for any $x, y \in N$, the function is convex (*resp. L*-smooth) along the geodesic curve $\gamma_1(t)$. On the other hand, $\mathcal{VT}$-geodesic convexity ($L$-smoothness) with base $b$ implies the function is convex along a curve $\gamma_2(t)$. Note a curve $\gamma_2$ exists for general cases (see Remark C.7).

gradient methods [CMRS20, ACGS21]. Motivated from this fact, when $\mathcal{VT} = \Gamma$, we will dub $\Gamma$-geodesic convexity (*resp.* smoothness) by *generalized geodesic convexity (*resp. *smoothness)*. Since $\mathcal{VT} = \Gamma$ will be our main application in Section 5, we introduce more detail about generalized geodesic convexity in Appendix C.

If the function is $\mathcal{VT}$-geodesically convex (*resp. L*-smooth), then it is geodesically convex (*resp. L*-smooth) by taking $b = x$. However, a function being $\mathcal{VT}$-geodesically convex (*resp. L*-smooth) with *single base* $b$ is *not* a strictly stronger condition than geodesic convexity (*resp. L*-smoothness) and is incomparable. For example, for $\mathcal{VT} = \Gamma$, *i.e.*, parallel transport, the function $x \mapsto \frac{1}{2}d^2(x, p)$ on a non-Euclidean Hadamard manifold, a manifold with non-positive curvature, is generalized geodesically 1-smooth with base $p$, while it is not geodesically smooth (see Example C.10 and [CK25]).

**Remark 3.6** (Geometric interpretation of $\mathcal{VT}$-geodesic convexity)**.** *Intuitively, for any three points $x, y, b \in N$, $\mathcal{VT}$-geodesic convexity requires $f$ to be convex along a curve from $x$ to $y$, where the initial velocity is measured in the tangent space at a third point $b \in M$ (see Figure 1 for the detail). This generalizes standard geodesic convexity, which corresponds to the special case $z = x$.*

We now establish the relationship between (3.1) and convexity and smoothness, as in Euclidean space. Proposition 3.7 provides a sufficient condition for (3.1).

**Proposition 3.7.** *Let $f$ be a $\mathcal{VT}$-geodesically $L$-smooth with base $b \in N$, $\mathcal{VT}$-geodesically convex, and for all $x, y \in N$, define $z := \exp_b \left( -\frac{1}{L} \left( \mathcal{VT}_y^b \operatorname{Grad} f(y) - \mathcal{VT}_x^b \operatorname{Grad} f(x) \right) + \log_b y \right) \in N$. Then $f$ satisfies (3.1) for all $x_i, x_j \in N$.*

We provide the proof in Appendix B.1. The condition $z \in N$ is technical and generally requires case-specific verification. We emphasize that, as a special case of Proposition 3.7 with $b = y$, *i.e.*, under standard geodesic smoothness, one obtains a Riemannian analogue of the *co-coercivity* type inequality [Nes14, Theorem 2.1.5], a fundamental property of convex $L$-smooth functions in Euclidean space. See Appendix B.1.1 for further discussion.

At first glance, (3.1) and the corresponding Proposition 3.7 may appear to be merely a technical assumption imposed for the convenience of the proof. However, we will later demonstrate in Section 5 that (3.1) admits meaningful applications, particularly when $\mathcal{VT} = \Gamma$. In this regard, our notion of $\mathcal{VT}$-geodesic convexity and smoothness is not introduced solely for technical convenience; rather, it can be viewed as the generalization of existing concepts and allows practical applications.

# 4 Main Results

In this section, we present our main convergence results for silver stepsize acceleration on Riemannian manifolds.

**Proposed Algorithm: VTRGD**  While classical RGDs are defined by the exponential map between consecutive updates as in (1.2), such construction turned out to be incapable to achieve the convergence for dynamic stepsize methods like silver stepsize. The main reason is because dynamic stepsize methods require the relationship between *non-consecutive* itereates, which is not feasible in standard RGD (1.2). To overcome this bottleneck, we propose *Vector Transported RGD (VTRGD)*. VTRGD is defined as follows. First, choose a vector transport $\mathcal{VT}$. Next, choose an arbitrary base point $b \in M$ which satisfies (3.1) (e.g., a base $b$ that makes $f$ $\mathcal{VT}$-geodesically $L$-smooth with base $b$). Then, VTRGD makes the following update:

$$x_{n+1} = \exp_b \left( \log_b x_n - \frac{\eta_n}{L} \mathcal{VT}_{x_n}^b \operatorname{Grad} f(x_n) \right). \tag{4.1}$$

Since we assumed that the exponential map, logarithmic map, and Riemannian gradients are tractable in Assumption 3.1, (4.1) provides a tractable algorithm whenever $\mathcal{VT}$ is tractable. In particular, we focus on the case when $\eta_n$ is the silver stepsize. The below theorem is our main theorem, showing that silver stepsize VTRGD achieves the accelerated convergence rate.

**Theorem 4.1.**  *Let Assumption 3.1. 3.2 be true and $n = 2^k - 1$. Then, for VTRGD (4.1) with silver stepsizes $\eta_n/L$ (2.1), we have,*

$$f(x_n) - f(x_*) \le r_k L \left\| \log_b x_0 - \log_b x_* \right\|_b^2, \qquad r_k = \left( 1 + \sqrt{4\rho^{2k} - 3} \right)^{-1}.$$

We provide the full proof in Appendix B.2. Since $r_k \asymp n^{-\log_2 \rho} \approx n^{-1.2716}$, Theorem 4.1 shows an accelerated convergence rate than constant stepsize RGD on Riemannian manifolds, which is $O(n^{-1})$ [KY22, Appendix D]. For $n \ne 2^k - 1$, the error may oscillate and not always remain in a low regime. However, the constraint $n = 2^k - 1$ is not a practical issue, since the number of iterations can be freely chosen. Moreover, while the theorem is formally stated for $n = 2^k - 1$, the oscillation happens when we encounter the huge stepsize, which is known a priori. Thus, as long as the iteration count avoids these spike points (specifically, when $n = 2^k$), the iterates remain stable. Our numerical experiments confirm this behavior; see Appendix D.

**Strongly convex case**  While the Euclidean silver stepsize method allows the strong convexity variant [AP24b], it relies heavily on the use of the interpolating inequality [AP24b, Eqation (2.6)] which does not seem to have the direct interpretation in Riemannian setting, as well as suffers the saturation (non-accelerating) regime after the certain number of the iterates. On the other hand, it turns out that silver stepsize VTRGD can be extended to $\mathcal{VT}$-geodesically $\alpha$-strongly convex functionals with base $b$ by the use of well-known restarting method [OC15]. The method proceeds as follows:

1. Perform $m$ steps of gradient descent starting from an initial point $x_0$ to obtain $x_m$.

2. Restart from $x_m$ with the stepsize reset to $\eta_0$, and run $m$ additional steps to obtain $x_{2m}$.

3. Repeat this process $\ell$ times, each time restarting from the most recent iterate with the stepsize reset to $\eta_0$. The total number of iteration will be $n = m\ell$.

For fixed $n$, choosing $m$ and $\ell$ appropriately yields the optimal convergence rate for strongly convex objectives. Notably, this approach remains valid in the Riemannian setting with silver stepsize VTRGD.

**Theorem 4.2.**  *Consider the same setting of Theorem 4.1. In addition, let $f$ be geodesically $\alpha$-strongly convex with base $b$ with the condition number $\kappa := L/\alpha$. Set $k^* = \lceil \log_\rho \kappa \rceil + 1$. For any $\ell \in \mathbb{N}$, consider the above restarting scheme with $m = 2^{k^*} - 1$, so that the total number of iteration is $n = \ell(2^{k^*} - 1)$. Then, for any $\ell \in \mathbb{N}$ and $n = \ell(2^{k^*} - 1)$,*

$$\left\| \log_b x_n - \log_b x_* \right\|_b^2 \le \exp\left( -\log(\rho/2) n / \kappa^{\log_\rho 2} \right) \left\| \log_b x_0 - \log_b x_* \right\|_b^2$$

*In particular, the algorithm finds an $\epsilon$-approximate solution, i.e., $\left\| \log_b x_n - \log_b x_* \right\|_b^2 \le \epsilon$ within $O\left( \kappa^{\log_\rho 2} \log(1/\epsilon) \right)$ iterations.*

We provide the proof in Appendix B.2.1. Although the left-hand side is $\|\log_b x_n - \log_b x_*\|_b^2$ rather than the usual objective $d^2(x_n, x_*)$, on any non-negatively curved Riemannian manifold it always upper-bounds the squared distance (see [ZS16, Lemma 5]). Hence, Theorem 4.2 yields a convergence rate for the squared distance at least on non-negatively curved Riemannian manifolds. Since constant stepsize RGD finds an $\epsilon$-approximate solution for strongly convex objectives in $O(\kappa \log(1/\epsilon))$ iterations [KY22, Appendix D], our algorithm achieves an improved rate in the strongly convex setting as well. While the algorithm requires to pick the certain number of iterations, again in practice this is not problematic as one can freely choose the number of iterates. Also, unlike [AP24b], the restarting method avoids suffering saturation regime.

We conclude this section with a remark on the choice of $b$. Although the convergence rates of our VTRGD algorithms in Theorems 4.1 and 4.2 are independent of the choice of the base point $b$, the choice of $b$ still influences the algorithm's performance through the constant factor.

# 5 Applications

At first glance, (3.1) and (4.1) may appear to be merely technical assumptions and consequences introduced for the convenience of the proof, without further implications. However, in this section, we demonstrate that VTRGD in fact coincides with the standard Riemannian gradient descent in the 2-Wasserstein space under the canonical choice of $\mathcal{VT} = \Gamma$.

## 5.1 Optimization on the 2-Wasserstein Space

A key advantage of our algorithm over existing methods is that it achieves the acceleration without requiring the curvature bounds. This feature makes our analysis particularly suitable for the 2-Wasserstein space, which admits a Riemannian structure but lacks an upper curvature bound (see Lemmas A.33 and A.42). While the acceleration has been studied in the continuous-time setting [CCT18, WL22], no discrete-time algorithm with provable acceleration guarantee was previously available. To the best of our knowledge, our method provides the first theoretically guaranteed accelerated algorithm in the 2-Wasserstein space for a widely used family of functionals, the *potential functional*.

We briefly introduce the 2-Wasserstein geometry (see Appendix A.2 for details). Let $\mathcal{P}_{2,ac}(\mathbb{R}^d)$ denote the set of probability measures on $\mathbb{R}^d$ with finite second moments and absolutely continuous with respect to the Lebesgue measure, $\mathcal{L}^2(\mu)$ be the space of square-integrable functions from $\mathbb{R}^d \to \mathbb{R}^d$ under $\mu \in \mathcal{P}_{2,ac}(\mathbb{R}^d)$, and $T_{\#\mu}$ denotes a pushforward of $\mu$ by $T$. For any $\mu, \nu \in \mathcal{P}_{2,ac}(\mathbb{R}^d)$, the 2-Wasserstein metric is defined as:

$$W_2^2(\mu, \nu) := \min_{T \in \mathcal{L}^2(\mu) \text{ s.t. } T_{\#\mu} = \nu} \mathbb{E}_{x \sim \mu} \left[ \|T(x) - x\|^2 \right]. \tag{5.1}$$

The well-definedness (precisely, the existence of the minimum $T$) is guaranteed by Brenier Theorem [Bre91]. The map $T_{\mu,\nu}$ achieving the minimum in (5.1) is called an *optimal transport map* from $\mu$ to $\nu$. The metric space $(\mathcal{P}_{2,ac}(\mathbb{R}^d), W_2)$, called the 2-Wasserstein space, admits a Riemannian structure with tangent space $T_\mu \mathcal{P}_{2,ac}(\mathbb{R}^d) \subset \mathcal{L}^2(\mu)$ and the Riemannian metric given by the $\mathcal{L}^2(\mu)$ inner product. The exponential map is defined by $\exp_\mu(v) = (id + v)_{\#\mu}$, and the logarithmic map is defined by $\log_\mu \nu = T_{\mu,\nu} - id$. The Wasserstein gradient at $\mu$ is defined as the operator satisfying $\partial_t|_{t=0} \mathcal{F}(\mu_t) = \langle \text{Grad}_{W_2} \mathcal{F}(\mu), v_0 \rangle$, where $\mu_t$ is any sufficiently regular curve of measures with $\mu_0 = \mu$ and $v_t \in \mathcal{L}^2(\mu_t)$ is the velocity vector satisfying the continuity equation $\partial_t \mu_t + \text{div}(\mu_t v_t) = 0$. This definition is the exact analogy of the definition of Riemannian gradient that is defined by $d_v f(x) = \langle \text{Grad} f(x), v \rangle$. A natural vector transport in 2-Wasserstein space is a transport map, *i.e.*, $\mathcal{VT}_\mu^\nu = T_{\nu,\mu}$, where $T_{\nu,\mu}$ is a map satisfying $(T_{\nu,\mu})_{\#\nu} = \mu$. If $T_{\nu,\mu}$ is the *optimal* transport map, this is in fact an *un-projected* parallel transport in 2-Wasserstein space. In particular, when one uses the *optimal* transport map, Definition 3.4, 3.5 becomes generalized geodesic convexity (*resp.* smoothness) in 2-Wasserstein space [AGS08, San14, SKL20].

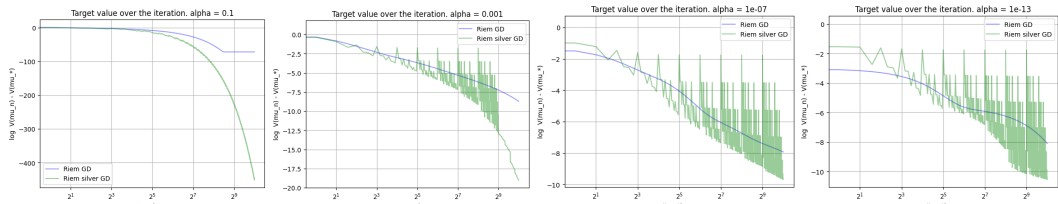

Figure 2: Comparison between silver stepsize method and RGD for potential functional optimization in $BW(\mathbb{R}^d)$, with different convexity parameters. We set $\ell = 2^{\left\lfloor \log_2 \left( \frac{2^{10}-1}{2^{k^*}-1} \right) \right\rfloor}$ and $n = \ell(2^{k^*} - 1)$, where $k^*$ being the optimal sub-iterate derived in Theorem 4.2. **Columns**: From left to right, each column corresponds to $\kappa = 10^1, 10^3, 10^7, 10^{13}$.

Perhaps surprisingly, VTRGD recovers the standard Wasserstein gradient descent (WGD) when $\mathcal{VT}$ is chosen to be *any* transport map, since for all $b \in \mathcal{P}_{2,ac}(\mathbb{R}^d)$ and any transport map $T_{b,\mu_n}$

$$\mu_{n+1} = (T_{b,\mu_n} - \frac{\eta_n}{L} \operatorname{Grad}_{W_2} \mathcal{F}(\mu_n) \circ T_{b,\mu_n})_{\#b} = (id - \frac{\eta_n}{L} \operatorname{Grad}_{W_2} \mathcal{F}(\mu_n))_{\#\mu_n}. \qquad (5.2)$$

Hence in 2-Wasserstein space, *silver stepsize Wasserstein gradient descent* guarantees the accelerated convergence, analogous to Theorem 4.1, and 4.2.

**Corollary 5.1** (Acceleration by silver stepsize WGD). *Let $\mathcal{F} : \mathcal{P}_{2,ac}(\mathbb{R}^d) \to \mathbb{R}$ to be a functional on 2-Wasserstein space. Set $n = 2^k - 1$, and let $\mu_n$ be a WGD update (5.2) with $\eta_n$ being the silver stepsize. If $\mathcal{F}$ satisfies (3.1) with some $b \in \mathcal{P}_{2,ac}(\mathbb{R}^d)$ and $\mathcal{VT}_b^{x_i} = T_{b,\mu_i}$, and all $\mu_i$ and $(id - \frac{1}{L} \left( \operatorname{Grad}_{W_2} \mathcal{F}(\mu_i) - \operatorname{Grad}_{W_2} \mathcal{F}(\mu_j) \circ T_{\mu_i,\mu_j} \right)_{\#\mu_i}$ admit the densities, we get*

$$\mathcal{F}(\mu_n) - \mathcal{F}(\mu_*) \le r_k L \left\| T_{b,\mu_0} - T_{b,\mu_*} \right\|_b^2 .$$

*Suppose $\mathcal{F}$ is, in addition, geodesically $\alpha$-strongly convex with the condition number $\kappa = L/\alpha$. Then, under the same setup as in Theorem 4.2,*

$$W_2^2(\mu_n, \mu_*) \le C_\kappa \exp \left( -\log(\rho/2)n/\kappa^{\log_\rho 2} \right) \left\| T_{b,\mu_0} - T_{b,\mu_*} \right\|_b^2 .$$

*In particular, the algorithm finds an $\epsilon$-approximate solution in squared 2-Wasserstein distance within $O\left( \kappa^{\log_\rho 2} \log(1/\epsilon) \right)$ iterations.*

In particular, when the condition (3.1) holds for all $b \in \mathcal{P}_{2,ac}(\mathbb{R}^d)$, one can substitute the right hand sides $\left\| T_{b,\mu_0} - T_{b,\mu_*} \right\|_b^2$ to $W_2^2(\mu_0, \mu_*)$ by choosing $b = \mu_0$ and letting $T_{\mu_0,\mu_*}$ be an optimal transport map. While Corollary 5.1 may appear to be a straightforward application of our main theorems, it in fact requires careful attention to the underlying geometry, since the 2-Wasserstein space is not *geodesically complete* [PZ20]. This obstacle can nevertheless be resolved by exploiting specific properties of the 2-Wasserstein geometry. The proof is deferred to Appendix B.3.

**Remark 5.2** (Optimality is not needed). *In classical Wasserstein gradient descent, one typically needs to impose the condition $id - \frac{\eta_n}{L} \operatorname{Grad}_{W_2} \mathcal{F}(\mu_n)$ to be convex, which restricts the size of steps to be less or equal to $1/L$; the convexity condition is essential for the gradient update to be the* optimal *transport map. However, our proof does not use the optimality of the update step. Therefore, we do not require the map $id - \frac{\eta_n}{L} \operatorname{Grad}_{W_2} \mathcal{F}(\mu_n)$ to be convex, and it justifies the choice of large stepsizes.*

**Remark 5.3** (Simplification of the regularity condition). *The regularity condition–existence of densities among iterates–can be simplified as follows. If the transport map $T_{\mu_i,\mu_j} \in C_{loc}^{1,1}(\mathbb{R}^d)$, and $\operatorname{Grad}_{W_2} \mathcal{F}(\mu) = \nabla h$ for some $h \in C_{loc}^{1,1}(\mathbb{R}^d)$, which is in fact the case for many of Wasserstein gradient algorithms, then the regularity condition boils down to $I - \frac{\eta_n}{L} \nabla \operatorname{Grad}_{W_2} \mathcal{F}(\mu)$ and $I - \left( \frac{1}{L} \nabla \operatorname{Grad}_{W_2} \mathcal{F}(\mu_i) - \nabla \operatorname{Grad}_{W_2} \mathcal{F}(\mu_j) \circ T_{\mu_i,\mu_j} \right)$ being invertible. See Appendix B.3 for the detail.*

Motivated by Corollary 5.1, we provide the numerical experiments on WGD. We set $M = \mathcal{P}_{2,ac}(\mathbb{R}^d)$ and $N$ as the Bures-Wasserstein space $BW(\mathbb{R}^d)$, the space of non-singular Gaussian distributions

in $\mathbb{R}^d$ equipped with Wasserstein geometry. $BW(\mathbb{R}^d)$ is a geodesically convex subset of $\mathcal{P}_{2,ac}(\mathbb{R}^d)$ [LCB$^+$22]. Moreover, $BW(\mathbb{R}^d)$ can be identified with a product Riemannian manifold of mean vectors and covariance matrices, *i.e.*, $\mathbb{R}^d \times \mathrm{SPD}(d)$ where $\mathrm{SPD}(d)$ denotes the space of symmetric positive definite matrices of $\mathbb{R}^{d \times d}$. We introduce more detail of $BW(\mathbb{R}^d)$ geometry in Appendix A.2.1. As our objective functional, we consider the *potential energy functional*:

$$\mathcal{V}(\mu) := \mathbb{E}_{x \sim \mu}[V(x)]$$

where $V : \mathbb{R}^d \to \mathbb{R}$. Such functional appears frequently in applications, for example as the component of variational inference. Using the explicit formula of $\mathrm{Grad}_{\mathrm{BW}}\, \mathcal{V}(m, \Sigma)$ [DBCS23] in (5.2), we obtain the following silver stepsize RGD in $BW(\mathbb{R}^d)$ for $\mathcal{V}(\mu)$:

$$
\begin{aligned}
m_{n+1} &= m_n - \frac{\eta_n}{L}\mathbb{E}_{X \sim N(m_n, \Sigma_n)}[\nabla V(X)], \\
\Sigma_{n+1} &= \left(I - \frac{\eta_n}{L}\mathbb{E}_{X \sim N(m_n, \Sigma_n)}[\nabla^2 V(X)]\right)\Sigma_n\left(I - \frac{\eta_n}{L}\mathbb{E}_{X \sim N(m_n, \Sigma_n)}[\nabla^2 V(X)]\right).
\end{aligned}
\tag{5.3}
$$

The following proposition shows that the potential functional is an instance of Corollary 5.1 whenever $V$ is convex and $L$-smooth.

**Proposition 5.4.** *If $V$ is convex and $L$-smooth in $\mathbb{R}^d$, then $\mathcal{V}$ satisfies (3.1) with any $b \in \mathcal{P}_{2,ac}(\mathbb{R}^d)$ and any transport map $\mathcal{V}\mathcal{T}_\mu^b = T_{b,\mu_i}$ under both the Wasserstein and Bures-Wasserstein geometries. In particular, the results in Corollary 5.1, with $\mathcal{F} = \mathcal{V}$, hold by substituting $W_2^2(\mu_0, \mu_*)$ for $\|T_{b,\mu_0} - T_{b,\mu_*}\|_b^2$.*

We provide the proof in Appendix B.3. By Corollary 5.1, the silver stepsize WGD (5.3) achieves the accelerated convergence rate whenever $I - \eta_n\mathbb{E}[\nabla^2 V(X)]/L$ is invertible. For our experiment, we considered quadratic $V(x) = \frac{1}{2}(x - m_*)^T \Sigma_*^{-1}(x - m_*)$ defined on $\mathbb{R}^{10}$, with $m_*, \Sigma_*$ being a randomly generated vector and symmetric positive definite matrix respectively. Since $V$ is a strongly-convex quadratic function, by Proposition 5.4 $\mathcal{V}$ is generalized geodesically $\alpha$-strongly convex and geodesically $L$-smooth with $L = 1/\lambda_{\min}(\Sigma_*)$ and $\alpha = 1/\lambda_{\max}(\Sigma_*)$. To study the effect of the condition number $\kappa = L/\alpha$, we fix $L = 1$, and vary $\alpha$. Small $\alpha$ corresponds to convex case, and larger $\alpha$ stands for the strongly convex case. We choose $1/L$ as the stepsize for constant stepsize WGD [ZS16, KY22]. Figure 2 shows that the silver stepsize WGD outperforms constant stepsize WGD in both convex and strongly convex case. We provide further implementation detail (e.g., the specific distributions of $m_*$ and $\Sigma_*$) and additional experiments under various settings (e.g., different random seeds, number of iterations, stochastic gradients, and comparisons with various constant stepsizes) in Appendix D.

Beyond quadratic potential, or even in the absence of convexity and smoothness guarantees, we observed that our algorithm yields numerical improvements for other optimization problems in the 2-Wasserstein space. Specifically, we present additional experiments on logistic regression potential and mean-field training of neural networks in Appendix D.2.1 and D.2.2.

## 5.2 Optimization on Symmetric Positive definite matrices

While our main motivation is 2-Wasserstein space, we also provide the numerical experiments for the optimization problem in symmetric positive definite matrix (SPD) with $\mathcal{V}\mathcal{T} = \Gamma$ again. The tangent space is $\mathrm{Sym}(d)$, the space of symmetric matrices of $\mathbb{R}^{d \times d}$. On this space, there is a natural choice of the metric, called *affine invariant metric*. The metric is defined by:

$$d_{AI}(A, B) := \left\|\log A^{-1/2} B A^{-1/2}\right\|_F, \quad \langle S, R\rangle_A = \mathrm{tr}(A^{-1}SA^{-1}R).$$

This metric coincides with the Fisher-Rao metric between multivariate Gaussian distributions with fixed mean and covariance matrices $A$ and $B$ [Nie23]. Additionally, the Fréchet mean of SPD matrices with respect to $d_{AI}$ coincides with the geometric mean and plays an important role in many applications, such as diffusion tensor imaging [FAP$^+$05, PFA05, BH06, Ngu22, KPB25]. This metric induces the complete non-positively curved manifold on $\mathrm{SPD}(d)$ with the curvature bound $[-1/2, 0]$ [CB23]. While there is a problem setup which fully satisfies our assumption on this space

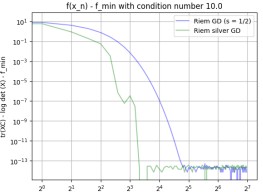
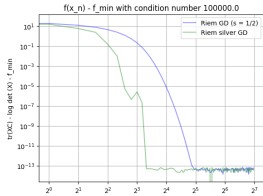

Figure 3: Optimization of $f(X) = \text{tr}(CX) - \log\det X$, with the different condition number on $C$. We set $b = I$ for VTRGD (4.1). We set $L = 2$ for both experiments. We plot $f - f_{\min}$, where $f_{\min}$ is the minimum value over all experiments. **Left:** $\kappa = 10$. **Right:** $\kappa = 10^5$.

(see Proposition D.1), to validate the wide applicability of our method we provide experiments on a representative benchmark problem which does not satisfy our assumptions. In addition, we provide more experiments with various setups on $\text{SPD}(d)$ in Appendix D.2.3, including a setup which does satisfies our assumption.

**Linear minus log-determinant**   The linear minus log-determinant is widely studied problem, as it can be considered as a generalization of linear semidefinite programming (SDP) problems as well as the log-likelihood of the Wishart distribution [BV04, WST10, HMJG21, MM24]. The problem is formulated as follows: for $C \in \text{SPD}(d)$, one optimizes the function

$$\min_{X \in \text{SPD}(d)} f(X) := \text{tr}(CX) - \log\det X.$$

Unfortunately, this problem does not satisfy our desired assumptions. In practice, however, standard RGD is nontheless applied to this problem [HMJG21]. We set $d = 50$. Since the problem is not geodesically $L$-smooth, one must choose $L$ manually; we set $L = 2$, which ensures the stability of both algorithms. For VTRGD with the silver stepsize (4.1), we set the base point to $b = I$. The results are summarized in Figure 3.

Codes for our experiments can be found at `https://github.com/wldyddl5510/VTRGD-Silver`.

# 6   Conclusion

In this work, we provide the new concept of convexity and smoothness on Riemannian manifolds, namely vector transported geodesic convexity and smoothness. Based on these new concepts, we propose vector transported Riemannian gradient descent (VTRGD) method (4.1), which enables silver stepsize acceleration to be feasible for the functions satisfying the $\mathcal{VT}$-geodesic convexity and $\mathcal{VT}$-geodesic smoothness with a base. Albeit under a different notion of convexity and smoothness, our algorithm is the first tractable accelerated algorithm in Riemannian manifolds, without imposing the curvature assumption or diameter assumption. As our core application, in 2-Wasserstein space VTRGD coincides with standard gradient descent in 2-Wasserstein space. Particularly, our framework yields the acceleration for potential energy functional optimization problem.

We conclude the paper with some open questions: 1. The implication of $\mathcal{VT}$-geodesic convexity and smoothness is not direct. We are unaware how restrictive or favorable these conditions will be. That said, since this notion is well studied in 2-Wasserstein geometry and there are nontrivial examples that satisfy these assumptions, it seems worthy of further investigation. 2. Whether one can obtain silver stepsize acceleration of standard RGD for general problems remains an open question. 3. We restricted our attention to deterministic gradient descent. Stochastic or proximal version still remains the open. 4. There are extensions of the silver stepsize; showing for arbitrary iterates $n$ or showing the optimality [Gri24, GSW24]. Also, for specific classes of functions in the Euclidean setting, [AP24a] proposed a stepsize schedule for gradient descent that achieves the fully accelerated rate $O(1/n^2)$, matching that of momentum methods. Extending these ideas to the Riemannian setting would be an intriguing direction for future work.

## Acknowledgments

JP acknowledges Bofan Wang for catching an algebraic flaw in an earlier version of the manuscript. JP additionally acknowledges Hyunwoong Chang and Jaewook J. Suh for helpful discussions on this topic. JP acknowledges support from NSF DMS-2210689 and NSF DMS-2424305. AB acknowledges NSF DMS-2210689 for supporting this project. JS ackowledges support from NSF DMS-2424305 and the ONR MURI grant N00014-20-1-2787.

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

# Appendix

## A  Preliminaries

### A.1  Riemannian geometry

In this appendix, we introduce key concepts in Riemannian geometry briefly discussed in Section 2. We mainly mention the known results, and omit the proof and well-definedness of definitions. For detail, interested reader can find relevant material in textbooks, e.g., [Lee12, Lee18, Bou23].

A Riemannian manifold is a manifold equipped with an inner product for each tangent space, called a Riemannian metric.

**Definition A.1** (Riemannian manifold). *A Riemannian manifold $(M, g)$ is a real smooth manifold equipped with a Riemannian metric $g$ which assigns to each $p \in M$ a positive-definite inner product $g_p(v, w) = \langle v, w \rangle_p$ on the tangent space $T_p M$.*

Often, this tangent space $T_p M$ is conveniently expressed in the form of the vector field, which takes a point in a manifold as an input and returns a tangent space vector at that point. Formally, the vector field of $M$ is defined as follows:

**Definition A.2** (Vector field). *A map $X : C^\infty(M) \to C^\infty(M)$ is called a smooth vector field if it is a derivation, i.e., $X$ satisfies*

$$X.(fg) = X.(f)g(\cdot) + f(\cdot)X.(g).$$

*Here $\cdot \in M$ is the input of the function.*

As the name derivation indicates, one can think of the vector field as a directional derivative along the direction of the vector field. The following familiar example may help.

**Example A.3** (Vector field in $\mathbb{R}^d$). *For $f \in C^\infty(\mathbb{R}^d)$, $p \in \mathbb{R}^d$, and $v \in \mathbb{R}^d$, think of a directional derivative of $f$ at $p$ along direction $v$, $d_v f(p)$. If we fix $p$ and view $f$ as a variable input, then $v \in T_p M$ can be identified with the functional $f \mapsto d_v f(p)$. In other words, by defining $X_p(f) := d_v f(p)$, the value of vector field $X_p$ at each point $p \in M$ can be identified as a tangent vector $v \in T_p \mathbb{R}^d$.*

From now on, we will write $X$ as a vector field, and this will mean a function $X_p(f) = d_v f(p)$ where $v = X_p$. For the definition of a directional derivative in general manifolds, we refer to [Lee12]. We write $\mathfrak{X}(M)$ as a set of smooth vector fields on $M$.

One of the fundamental structure of a manifold is an affine connection, a concept that connects tangent spaces of different points of the manifold.

**Definition A.4** (Affine connection). *Let $M$ be a manifold, and $\mathfrak{X}(M)$ be the set of all smooth vector fields on $M$. An operator $\nabla_\cdot \cdot : \mathfrak{X}(M) \times \mathfrak{X}(M) \to \mathfrak{X}(X)$ is called an affine connection if for all $f \in C^\infty(M)$ and $X, Y \in \mathfrak{X}(M)$ it satisfies the following properties:*

    *1. $\nabla_{fX} Y = f \nabla_X Y$, i.e. linear in the first variable.*

2. $\nabla_X(fY) = (d_X f)Y + f\nabla_X Y$, that is, $\nabla$ satisfies the Leibniz rule in the second variable.

In the case of Riemannian manifolds, we have a natural connection induced from the Riemannian metric, called Levi-Civita connection.

**Definition A.5** (Levi-Civita connection). *For a Riemannian manifold $(M, g)$, let $\mathfrak{X}(M)$ be a set of smooth vector field on $M$. The Levi-Civita connection is the unique affine connection $\nabla \cdot :$ $\mathfrak{X}(M) \times \mathfrak{X}(M) \to \mathfrak{X}(M)$, satisfying the following properties:*

1. $\nabla_X Y - \nabla_Y X = [X, Y]$, *i.e. it is torsion-free. Here, $[\cdot, \cdot]$ denotes a Lie bracket.*

2. $X\left(g(Y, Z)\right) = g(\nabla_X Y, Z) + g(Y, \nabla_X Z)$, *that is, the connection is compatible with the metric $g$.*

The choice of the affine connection determines multiple geometric concepts. One fundamental concept is geodesic curve, which is a constant speed curve on the manifold.

**Definition A.6** (Geodesic). *A smooth curve $\gamma : [0, 1] \to M$ is called a geodesic curve if $\nabla_{\dot\gamma}\dot\gamma = 0$.*

A Riemannian manifold is called *complete* if any two points are connected by some geodesic. We will always assume $M$ is a complete Riemannian manifold.

We say a Riemannian submanifold $\widetilde{M} \subseteq M$ is *totally geodesic* if for every $v \in T\widetilde{M}$, the geodesic with respect to $\widetilde{M}$, $\gamma_v$, lies entirely in $M$.

Equipped with the notion of geodesic, one can define the exponential map and logarithmic map on a Riemannian manifold.

**Definition A.7** (Exponential map, logarithmic map). *Let $p \in M$.*

1. *For any $v \in T_p M$, one can define a geodesic curve $\gamma_v : [0, 1] \to M$ such that $\gamma_v(0) = p$ and $\gamma_v'(0) = v$. Then, one can define a map $\exp_p(v) := \gamma_v(1)$. This map is called the exponential map.*

2. *It is known that the exponential map is a local diffeomorphism on $U$, the open neighborhood of $0 \in T_p M$. Therefore, one can define $\log_p q := \exp_p^{-1}(q)$ for $q \in \exp_p(U)$. This map is called the logarithmic map.*

To understand the notions of the exponential map and logarithmic map, we illustrate these concepts in the Euclidean case. In the Euclidean space, $\exp_p(v) = p + v$ and $\log_p q = q - p$. In other words, the exponential map moves $p$ along the tangent direction $v$, and the logarithmic map returns the tangent direction from $p$ to $q$.

Note the logarithmic map is only defined locally. While our analysis assumed the global existence of the logarithmic map over the geodesically convex subset $N$ (Assumption 3.1), whether there is a global logarithmic map is not always guaranteed.

Another geometric concept induced from the connection is a covariant derivative, a notion of differentiation of the vector field along the curve [Bou23, Definition 5.28, 5.29].

**Definition A.8** (Vector field along the curve). *Let $\gamma : [0, 1] \to M$ be a smooth curve. A map $Z : [0, 1] \to TM$ is called a vector field on $\gamma$ if $Z(t) \in T_{\gamma(t)}M$ for all $t \in [0, 1]$. We write the set of vector fields on $\gamma$ as $\mathfrak{X}(\gamma)$.*

**Definition A.9** (Covariant derivative). *Let $\gamma : [0, 1] \to M$ be a smooth curve and $\nabla$ be an affine connection. Then, the covariant derivative is the unique operator $D_t : \mathfrak{X}(\gamma) \to \mathfrak{X}(\gamma)$ satisfying the following properties for all $Y, Z \in \mathfrak{X}(\gamma), W \in \mathfrak{X}(M), g \in c^\infty([0, 1])$ and $a, b \in \mathbb{R}$:*

1. $D_t(aY + bZ) = aD_t(Y) + bD_t(Z)$.

2. $D_t(gZ) = (\frac{d}{dt}g)Z + gD_t(Z)$.

3. $(D_t(W \circ \gamma))(t) = \nabla_{\gamma'(t)}W$ *for all $t \in [0, 1]$.*

*If $\nabla$ is the Levi-Civita connection, then the covariant derivative also satisfies*

$$\frac{d}{dt}\langle Y, Z\rangle = \langle D_t Y, Z\rangle + \langle Y, D_t Z\rangle.$$

Parallel transport is a notion of transporting vectors between different tangent space parallely. The parallel transport is uniquely determined by the covariant derivative [Bou23, Defintion 10.33, 10.35].

**Definition A.10.** *A vector field $Z \in \mathfrak{X}(\gamma)$ is called parallel if $D_t Z = 0$.*

**Definition A.11** (Parallel transport)**.** *Let $\gamma : [0,1] \to M$ be a smooth curve. The parallel transport of the tangent vector at $T_{\gamma(t_0)}M$ to the tangent vector at $T_{\gamma(t_1)}M$ along the curve $\gamma$ is the map*

$$\Gamma(\gamma)_{t_0}^{t_1} : T_{\gamma(t_0)}M \to T_{\gamma(t_1)}M$$

*defined by $\Gamma(\gamma)_{t_0}^{t_1}(Z(t_0)) = Z(t_1)$ for the parallel vector field $Z \in \mathfrak{X}(\gamma)$.*

We collect some properties of the parallel transport.

**Proposition A.12.** *[Bou23, Proposition 10.36]*

1. *$\Gamma(\gamma)_{t_0}^{t_1}$ is a linear map.*

2. *$\Gamma(\gamma)_{t_1}^{t_2} \circ \Gamma(\gamma)_{t_0}^{t_1} = \Gamma(\gamma)_{t_0}^{t_2}$.*

3. *$\Gamma(\gamma)_{t_0}^{t_1} \circ \Gamma(\gamma)_{t_1}^{t_0} = id$.*

4. *$\langle v, w\rangle_{\gamma(t_0)} = \left\langle \Gamma(\gamma)_{t_0}^{t_1} v, \Gamma(\gamma)_{t_0}^{t_1} w\right\rangle_{\gamma(t_1)}$.*

When $\gamma$ is chosen to be the geodesic curve such that $\gamma(0) = x$ and $\gamma(1) = y$, we denote the parallel transport $\Gamma(\gamma)_0^1$ as $\Gamma_x^y$. When context is clear, we will denote $\Gamma_x^y$ as the (geodesic) parallel transport from $x$ to $y$.

**Remark A.13** (Properties of geodesic parallel transport)**.** *By Proposition A.12, a geodesic parallel transport $\Gamma_x^y$ satisfies the following properties:*

1. *$\Gamma_x^y$ is a linear map.*

2. *$\Gamma_x^y \circ \Gamma_y^x = id$.*

3. *$\langle v, w\rangle_x = \langle \Gamma_x^y v, \Gamma_x^y w\rangle_y$.*

*Note the second property is dropped, as geodesics from $x$ to $y$ and $y$ to $z$ do not necessarily be in the same curve.*

Remark A.13 is the key properties of parallel transport used in our analysis. These properties play a pivotal role when we define the parallel transport in 2-Wasserstein space (Proposition A.30).

The last geometric concept induced from the Levi-Civita connection is curvature.

**Definition A.14** (Riemannian curvature)**.** *The Riemannian curvature tensor $R(\cdot, \cdot)\cdot : \mathfrak{X}(M) \times \mathfrak{X}(M) \times \mathfrak{X}(M) \to \mathfrak{X}(M)$ is defined by the following formula:*

$$R(X, Y)Z = \nabla_X \nabla_Y Z - \nabla_Y \nabla_X Z - \nabla_{[X,Y]} Z$$

*where $[\cdot, \cdot]$ denotes a Lie bracket.*

The key geometric quantity in our analysis is sectional curvature, which generalizes Gaussian curvature in a 2-dimensional surface.

**Definition A.15** (Sectional curvature)**.** *Let $p \in M$, and denote $\Sigma_p$ a set of two-dimensional subspaces in $T_p M$. The sectional curvature $K : \Sigma_p \to \mathbb{R}$ is defined by the following formula:*

$$K(\sigma_p) = \frac{\langle R(u, v)v, u\rangle_p}{\|u\|_p^2 \|v\|_p^2 - \langle u, v\rangle_p^2}$$

*where $\{u, v\}$ is a basis of $\sigma_p$.*

Note that we can write this sectional curvature as a function of two linearly independent vectors in $T_p M$ as well. In particular, if $u, v$ are orthonormal, then $K(u, v) = \langle R(u, v)v, u \rangle_p$.

A Riemannian manifold is called flat if for all $p$ and $\sigma_p$ sectional curvature $K(\sigma_p) = 0$, positively curved if $K(\sigma_p) > 0$, and negatively curved if $K(\sigma_p) < 0$.

### A.1.1  Functional properties of functions on Riemannian manifolds

In this appendix, we introduce additional functional properties of functions on a Riemannian manifold.

We begin with introducing the notion of geodesically convex set.

**Definition A.16.** *[Bou23, Definition 11.2] Let $(M, g)$ be a complete Riemannian manifold. $N \subseteq M$ is called geodesically convex subset of $M$ if for all $x, y \in N$, there exists a geodesic $\gamma : [0, 1] \to M$ such that $\gamma(0) = x$, $\gamma(1) = y$, and $\gamma(t) \in N$ for all $t \in [0, 1]$.*

Next, we introduce the notion of geodesic convexity and smoothness.

**Definition A.17** (Geodesic convexity and smoothness)**.** *Let $f : N \to \mathbb{R}$ be a differentiable function.*

1. *$f$ is called geodesically $\alpha$-strongly convex if for all $x, y \in N$*

$$f(y) \geq f(x) + \langle \operatorname{Grad} f(x), \log_x y \rangle_x + \frac{\alpha}{2} d^2(x, y).$$

   *If $\alpha = 0$, we say $f$ is geodesically convex.*

2. *$f$ is called geodesically $L$-smooth if for all $x, y \in N$*

$$\left\| \Gamma_y^x \operatorname{Grad} f(y) - \operatorname{Grad} f(x) \right\|_x \leq L d(x, y).$$

Now, we show the key inequality induced from the geodesic $L$-smoothness. This is often called descent lemma.

**Lemma A.18** (Descent lemma)**.** *If $f$ is geodesically $L$-smooth, then for all $x, y \in N$*

$$f(y) \leq f(x) + \langle \operatorname{Grad} f(x), \log_x y \rangle + \frac{L}{2} d^2(x, y).$$

*Proof.* Let $\gamma : [0, 1] \to M$ be a geodesic curve such that $\gamma(0) = x, \gamma(1) = y$. By the definition of the Riemannian logarithmic map, we get $\gamma'(0) = \log_x y$. By Fundamental Theorem of Calculus and properties of the parallel transport,

$$f(y) = f(\gamma(1)) = f(\gamma(0)) + \int_0^1 \frac{d}{dt}(f \circ \gamma)(t)dt = f(x) + \int_0^1 \langle \operatorname{Grad} f(\gamma(t)), \gamma'(t) \rangle dt$$

$$= f(x) + \int_0^1 \left\langle \Gamma_{\gamma(t)}^{\gamma(0)} \operatorname{Grad} f(\gamma(t)), \gamma'(0) \right\rangle dt = f(x) + \int_0^1 \left\langle \Gamma_{\gamma(t)}^x \operatorname{Grad} f(\gamma(t)), \log_x y \right\rangle dt.$$

Then, by subtracting $f(x) + \langle \operatorname{Grad} f(x), \log_x y \rangle$ from the both hand sides,

$$f(y) - f(x) - \langle \operatorname{Grad} f(x), \log_x y \rangle = \int_0^1 \left\langle \Gamma_{\gamma(t)}^x \operatorname{Grad} f(\gamma(t)) - \operatorname{Grad} f(x), \log_x y \right\rangle$$

$$\overset{(i)}{\leq} \int_0^1 \left\| \Gamma_{\gamma(t)}^x \operatorname{Grad} f(\gamma(t)) - \operatorname{Grad} f(x) \right\| \|\log_x y\| \, dt$$

$$\overset{(ii)}{\leq} \int_0^1 L d(\gamma(t), x) d(x, y) dt \overset{(iii)}{=} L d^2(x, y) \int_0^1 t \, dt$$

$$= \frac{L}{2} d^2(x, y).$$

For (i) we used Cauchy-Schwartz inequality, and for (ii) we used $L$-smoothness property. For (iii) we used the fact that the geodesic curve satisfies $d(x, \gamma(t)) = t d(x, y)$ due to the constant speed property. ∎

Note we used this descent lemma as the definition of the geodesic $L$-smoothness in the main body. This is because we wanted to make the correspondence with $\mathcal{VT}$-geodesic smoothness. For our $\mathcal{VT}$-geodesic smoothness, Lipschitz gradient type condition no longer implies (3.3) for general $\mathcal{VT}$, unless $\gamma'(t) = \left(\mathcal{VT}_{\gamma(0)}^{\gamma(t)}\right)^* \gamma'(0)$ for a curve $\gamma$, where $*$ denotes the adjoint operator. On the other hand, it turns out that our results only require (3.3) rather than Lipschitz gradient type condition, thus we stayed with Definition 3.5 to pursue the generality.

### A.1.2 Product Riemannain manifold

In Appendix A.2.1, we will encounter a product manifold. To that end, we present some preliminary facts here. We omit the details and simply list a few useful results. For more information on product Riemannian manifolds, we refer the reader to [Lee18].

**Definition A.19** (Product Riemannian manifold). *A product Riemannian manifold is a manifold $M = M_1 \times M_2$ such that each $(M_1, g_1)$ and $(M_2, g_2)$ are Riemannian manifolds, and the Riemannian metric $g$ is defined by the product metric:*

$$g\left((X_1, X_2), (Y_1, Y_2)\right) = g_1(X_1, Y_1) + g_2(X_2, Y_2).$$

Product Riemannians manifold have useful properties that make the computation easier.

**Theorem A.20** (Levi-Civita connection of a product Riemannian manifold). *The Levi-Civita connection of a product Riemannian manifold $(M, g) = (M_1, g_1) \times (M_2, g_2)$ satisfies the following property:*

$$\nabla_{(X_1, X_2)}(Y_1, Y_2) = \nabla_{1, X_1} Y_1 \oplus \nabla_{2, X_2} Y_2.$$

The following corollary is a direct consequence of the definition of Riemannian curvature, Lie bracket, and Theorem A.20.

**Corollary A.21** (Riemannian curvature of a product Riemannian manifold).

$$R\left((X_1, X_2), (Y_1, Y_2)\right)(Z_1, Z_2) = R_1(X_1, Y_1)Z_1 \oplus R_2(X_2, Y_2)Z_2.$$

Lastly, we obtain the following collorary, which will play an important role in our later section.

**Corollary A.22** (Sectional curvature of product Riemannian manifold). *Let $(u_1, u_2), (v_1, v_2)$ be orthonormal vectors in $T_p M$. Write $A_i := \|u_i\|^2 \|v_i\|^2 - g_i(u_i, v_i)^2$. Then,*

$$K\left((u_1, u_2), (v_1, v_1)\right) = A_1 K_1(u_1, v_1) + A_2 K_2(u_2, v_2).$$

*Proof.* From Definition A.15, Definition A.19, and Corollary A.21, we have

$$
\begin{aligned}
K\left((u_1, u_2), (v_1, v_2)\right) &= g\left(R((u_1, u_2), (v_1, v_2))(v_1, v_2), (u_1, u_2)\right) \\
&= g\left((R_1(u_1, v_1)v_1, R_2(u_2, v_2)v_2), (u_1, u_2)\right) \\
&= g_1(R_1(u_1, v_1)v_1, u_1) + g_2(R_2(u_2, v_2)v_2, u_2) \\
&= A_1 K_1(u_1, v_1) + A_2 K_2(u_2, v_2).
\end{aligned}
$$

$\blacksquare$

In particular, if $K_1 = 0$, *i.e.*, one of the spaces is flat, the the curvature behavior of the product manifold is entirely determined by $K_2$. This will be the case in Appendix A.2.1.

## A.2 Wasserstein geometry

In this appendix, we introduce the core concept of Wasserstein geometry, which is one of our key application. We write the space of probability measures with a finite $p$th moment on $\mathbb{R}^d$ by $\mathcal{P}_p(\mathbb{R}^d)$. Again, we mainly introduce the known results without proofs. For interested readers, we refer to [Vil08, AGS08, San14, Che24].

For $\mu, \nu \in \mathcal{P}_p(\mathbb{R}^d)$, let $\Gamma(\mu, \nu)$ be a set of couplings of $\mu$ and $\nu$. Wasserstein distance between $\mu$ and $\nu$ are defined as follows.

**Definition A.23** (Wasserstein metric). *Let $\mu, \nu \in \mathcal{P}_p(\mathbb{R}^d)$. Denote $\Gamma(\mu, \nu)$ to be a set of coupling measures of $\mu$ and $\nu$. $p$-Wasserstein distance between $\mu$ and $\nu$ is defined as follows:*

$$W_p^p(\mu, \nu) := \inf_{\gamma \in \Gamma(\mu, \nu)} \mathbb{E}_{(x,y) \sim \gamma} \left[ \|x - y\|^p \right].$$

This is known to be a well-defined metric. A metric space $(\mathcal{P}_p(\mathbb{R}^d), W_p)$ is called $p$-Wasserstein space.

2-Wasserstein space is typically a more interesting space compared to other p-Wasserstein spaces due to its geometric properties. [Bre91, JKO98, Ott01] found out that if we restrict our attention to the probability measures which are absolutely continuous with respect to Lebesgue measure and have a finite second moment, denoted by $\mathcal{P}_{2,ac}(\mathbb{R}^d)$, then $(\mathcal{P}_{2,ac}(\mathbb{R}^d), W_2)$ endows a richer geometric properties. Specifically, while $(\mathcal{P}_{2,ac}(\mathbb{R}^d), W_2)$ is not precisely a Riemannian manifold, its geometry is almost same to the non-negatively curved Riemannian manifold.

The reason $(\mathcal{P}_{2,ac}(\mathbb{R}^d), W_2)$ endows a Riemannian structure is rooted from the following theorem [Bre91]:

**Theorem A.24** (Brenier Theorem). *If $\mu, \nu \in \mathcal{P}_{2,ac}(\mathbb{R}^d)$, then*

$$W_2^2(\mu, \nu) = \min_{T \in L^2(\mu) \text{ s.t. } T_{\#\mu} = \nu} \mathbb{E}_{x \sim \mu} \left[ \|T(x) - x\|^2 \right] = \min_{T \in L^2(\mu) \text{ s.t. } T_{\#\mu} = \nu} \|T - id\|^2_{L^2(\mu; \mathbb{R}^d)}.$$

*Denote the minima as $T_{\mu,\nu}$. Then $T_{\mu,\nu}$ is a gradient of some convex function $\phi$ on $\mathbb{R}^d$ $\mu$-a.e. Furthermore, $T_{\mu,\nu} \circ T_{\nu,\mu} = id$. The minima $T_{\mu,\nu}$ is called the optimal transport map from $\mu$ to $\nu$.*

Theorem A.24 gives a notion of tangent direction at $\mu$.

**Definition A.25** (Riemannian metric in 2-Wasserstein space). *For $\mu \in W_2(\mathbb{R}^d)$, a tangent space of $\mu$ is $T_\mu \mathcal{P}_{2,ac}(\mathbb{R}^d) = \overline{\{\nabla \psi \mid \psi \in C_c^\infty(\mathbb{R}^d)\}}^{\mathcal{L}^2(\mu)} \subset \mathcal{L}^2(\mu)$. Here, $C_c^\infty(\mathbb{R}^d)$ is a set of compactly supported smooth functions on $\mathbb{R}^d$. The Riemannian metric is defined as a $\mathcal{L}^2(\mu)$-inner product. In other words, $\langle v, w \rangle_\mu = \mathbb{E}_{x \sim \mu}[\langle v(x), w(x)\rangle]$.*

**Remark A.26** (Interpretation of the tangent space). *By Brenier theorem, $T_{\mu,\nu} = \nabla \phi$. For arbitrary $\lambda > 0$, it follows that $\lambda(T_{\mu,\nu} - id) = \nabla(\lambda\phi - \lambda\frac{\|\cdot\|^2}{2}) \in T_\mu \mathcal{P}_{2,ac}(\mathbb{R}^d)$. This implies that the tangent space $T_\mu \mathcal{P}_{2,ac}(\mathbb{R}^d)$ can be interpreted as the set of scaled displacement fields $\lambda(T_{\mu,\nu} - id)$. If $X \sim \mu$ and $Y \sim \nu$, then $\lambda(T_{\mu,\nu} - id)(X) = \lambda(Y - X)$, which corresponds to directions in the usual Euclidean sense. From this perspective, the tangent space is naturally constructed to represent Euclidean directions at the level of individual particles.*

One can naturally define a geodesic curve in $(\mathcal{P}_{2,ac}(\mathbb{R}^d), W_2)$, by pushforwarding the interpolation between particles to the measure space.

**Definition A.27** (Geodesic in Wasserstein space). *A geodesic curve $\gamma : [0, 1] \to \mathcal{P}_{2,ac}(\mathbb{R}^d)$ such that $\gamma(0) = \mu$ and $\gamma(1) = \nu$ can be defined as follows:*

$$\gamma(t) = ((1 - t)id + tT_{\mu,\nu})_{\#\mu}.$$

The exponential map and logarithmic map are then defined accordingly.

**Definition A.28** (Exponential map and Logarithmic map in Wasserstein space). *For $\mu, \nu \in \mathcal{P}_{2,ac}(\mathbb{R}^d)$ and $v \in \mathcal{L}^2(\mu)$, exponential map and logarithmic map of $(\mathcal{P}_{2,ac}(\mathbb{R}^d), W_2)$ are defined as follows:*

$$exp_\mu(v) = (v + id)_{\#\mu},$$
$$\log_\mu(\nu) = T_{\mu,\nu} - id.$$

A favorable property of 2-Wasserstein space is that the exponential map (and accordingly logarithmic map) is globally well-defined on $\mathcal{L}^2(\mu)$, *i.e.*, 2-Wasserstein space satisfies Assumption 3.1.

This Riemannian structure induces 2-Wasserstein metric. Observe the Riemannian distance induced from the above structure coincides with the Wasserstein distance; $d(\mu, \nu)^2 = \|\log_\mu \nu\|^2 = \|T_{\mu,\nu} - id\|^2 = W_2^2(\mu, \nu)$.

One can define a geodesic parallel transport as well.

**Definition A.29.** *[AG08, Parallel transport] For $\mu, \nu \in \mathcal{P}_{2,ac}(\mathbb{R}^d)$ and $v \in T_\mu \mathcal{P}_{2,ac}(\mathbb{R}^d)$,*

$$\Gamma_\mu^\nu v := \Pi_\nu(v \circ T_{\nu,\mu}).$$

*Here, $\Pi_\cdot$ is a projection operator $\mathcal{L}^2(\cdot) \to T.\mathcal{P}_{2,ac}(\mathbb{R}^d)$.*

This definition of parallel transport is not entirely satisfactory, as it involves the operator $\Pi_\cdot$ which lacks an explicit form. However, recall our analysis only requires the properties of parallel transport in Remark A.13. It turns out that even if we drop $\Pi_\cdot$ and consider $\Gamma_\mu^\nu v = v \circ T_{\nu,\mu}$ as a parallel transport onto $\mathcal{L}^2(\mu)$, the corresponding parallel transport still has properties in Remark A.13, which are sufficient for our analyses.

**Proposition A.30** (Transfer lemma). *For $\mu, \nu \in \mathcal{P}_{2,ac}(\mathbb{R}^d)$ and $v \in \mathcal{L}^2(\mu)$, define $\Gamma_\mu^\nu v := v \circ T_{\nu,\mu}$. Then,*

1. $\Gamma_\mu^\nu$ *is linear operator on $\mathcal{L}^2(\mu)$.*

2. $\Gamma_\mu^\nu \circ \Gamma_\nu^\mu = id$.

3. $\langle v, w \rangle_\mu = \langle \Gamma_\mu^\nu v, \Gamma_\mu^\nu w \rangle_\nu$.

*Proof.* Property 1 is direct: for $v, w \in \mathcal{L}^2(\mu)$ and $a, b \in \mathbb{R}$, $\Gamma_\mu^\nu(av + bw) = av \circ T_{\mu,\nu} + bw \circ T_{\mu,\nu} = a\Gamma_\mu^\nu v + b\Gamma_\mu^\nu w$.

Property 2 is from Theorem A.24.

Property 3 is a direct consequence of the change of the measure formula:

$$\langle v, w \rangle_\mu = \int \langle v(x), w(x) \rangle \, d(T_{\nu,\mu})_{\#}\nu(x) = \int \langle v \circ T_{\nu,\mu}(x), w \circ T_{\nu,\mu}(x) \rangle \, d\nu(x) = \langle \Gamma_\mu^\nu v, \Gamma_\mu^\nu w \rangle_\nu.$$

∎

Therefore, by Proposition A.30, we can use the *un-projected* parallel transport $\cdot \circ T_{\nu,\mu}$ as a parallel transport $\Gamma_\mu^\nu \cdot$ and $\mathcal{L}^2(\mu)$ as the tangent space for our analysis. In fact, such parallel transport and tangent space are sufficient for other first-order Wasserstein gradient flow analyses as well (e.g., [AGS08, SKL20]).

Now, we introduce a sectional curvature in 2-Wasserstein space. To establish this, we introduce the continuity equation and the notion of covariant derivative in the 2-Wasserstein space.

**Definition A.31** (Continuity equation). *Let $\mu_t$ be a flow in $\mathcal{P}_{2,ac}(\mathbb{R}^d)$. For given $\mu_t$, there exists a vector field $v_t \in \mathcal{L}^2(\mu_t)$ such that*

$$\partial_t \mu_t = -div(\mu_t v_t).$$

*Such $v_t$ is called a (velocity) vector field of the flow $\mu_t$.*

One can think of $v_t$ as a velocity at $\mu_t$, and plays a similar role as $\gamma'(t)$ in Riemannian manifolds.

**Definition A.32** (Covariant derivative). *A covariant derivative of $w_t \in T_{\mu_t}\mathcal{P}_{2,ac}(\mathbb{R}^d)$ along a curve $\mu_t$ is defined by the following formula:*

$$\nabla_{v_t} w_t = \Pi_{\mu_t} \left( \lim_{h \to 0} \frac{\Gamma_{\mu_t}^{\mu_{t+h}} w_{t+h} - w_t}{h} \right).$$

*Here, $\Gamma$ is a parallel transport defined in Definition A.29, and $v_t$ is a vector field of the flow $\mu_t$.*

We are ready to introduce the result that 2-Wasserstein space is non-negatively curved.

**Lemma A.33.** *Let $v_t, w_t$ be orthonormal elements in $T_{\mu_t}\mathcal{P}_{2,ac}(\mathbb{R}^d)$. Then, the sectional curvature of the subspace spanned by these two tangent vectors is as follows:*

$$K_{\mu_t}(v_t, w_t) = 3\|\nabla v_t \cdot w_t - \nabla_{w_t} v_t\|_{\mathcal{L}^2(\mu_t)}^2$$

*where the first $\nabla$ is Euclidean gradient, and the second $\nabla_{w_t} v_t$ is a covariant derivative.*

We refer to [AG08, Proposition 7.2] or [Lot07, Corollary 5.13] for the derivation.

The last ingredients we need for the analysis of the Wasserstein space are notions of gradient. The concept is defined as an analogous manner to the Riemannian case. Again, we omit the detail and just present the result.

Wasserstein gradient is defined analogously to the formula $d_v f(x) = \langle \mathrm{Grad}\, f(x), v \rangle_x$ in Riemannian manifold.

**Definition A.34** (Wasserstein gradient). *For a functional $\mathcal{F} : \mathcal{P}_{2,ac}(\mathbb{R}^d) \to \mathbb{R}$, the Wasserstein gradient of $\mathcal{F}$ at $\mu_0$ is an element of $\mathcal{L}^2(\mu_0)$ satisfying the following equation:*

$$\partial_t \mathcal{F}(\mu_t)\big|_{t=0} = \langle \mathrm{Grad}_{W_2} \mathcal{F}(\mu_0), v_0 \rangle_{\mu_0}.$$

*Here $v_t$ is a vector field of the flow $\mu_t$.*

*One has the following explicit formula:*

$$\mathrm{Grad}_{W_2} \mathcal{F}(\mu) = \nabla \frac{\delta \mathcal{F}(\mu)}{\delta \mu}.$$

*Here, $\nabla$ is Euclidean gradient and $\frac{\delta \mathcal{F}(\mu)}{\delta \mu}$ is the first variation.*

Here, the role of $\gamma'(0)$ is changed to $v_0$. For the derivation we refer to [Che24, Theorem 1.4.1].

### A.2.1 Bures-Wasserstein geometry

In this appendix, we briefly introduce Bures-Wasserstein space $BW(\mathbb{R}^d)$, a space of Gaussian measures equipped with $W_2$ metric. Main takeaways of this appendix are as follow:

1. $BW(\mathbb{R}^d)$ is a product Riemannian manifold with non-negative sectional curvature.

2. $BW(\mathbb{R}^d)$ is a geodesically convex subset of $(\mathcal{P}_{2,ac}(\mathbb{R}^d), W_2)$ and totally geodesic submanifold. In this regard, we can take $N = BW(\mathbb{R}^d)$ for our algorithm.

3. This example shows how one can parameterize the transport map to make the algorithm implementable as in Equation (5.3).

4. This example confirms that $BW(\mathbb{R}^d)$, and therefore the 2-Wasserstein space, do not admit the curvature upper bound. Consequently, existing acceleration methods requiring the curvature upper bound are not well-suited for solving the optimization problems in Wasserstein space.

Again, we briefly list the results. For detail, we refer to [Tak09, BJL19, ACGS21, LCB$^+$22, DBCS23].

**Definition A.35** (Optimal transport map between Gaussian). *The optimal transport map between $\mu_0 = N(m_0, \Sigma_0)$ and $\mu_1 = N(m_1, \Sigma_1)$ is defined as follows:*

$$T_{\mu_0, \mu_1}(x) = m_1 + \Sigma_0^{-1/2} (\Sigma_0^{1/2} \Sigma_1 \Sigma_0^{1/2})^{1/2} \Sigma_0^{-1/2} (x - m_0).$$

Definition A.35 saids the optimal transport map between Gaussians is an affine map. This fact provides two favorable results.

First, since affine transform of the Gaussian is also a Gaussian, from Definition A.27 every geodesic interpolation between two Gaussians is also Gaussian. This shows $BW(\mathbb{R}^d)$ is a geodesically convex subset of 2-Wasserstein space. In addition it implies $BW(\mathbb{R}^d)$ is totally geodesic submanifold of 2-Wasserstein space [Lee18, Exercise 8.4].

Second, we can identify $\mu = N(m, \Sigma) \cong (m, \Sigma) \in \mathbb{R}^d \times \mathrm{SPD}(d)$ and $T_\mu BW(\mathbb{R}^d) \cong (a, S) \in \mathbb{R}^d \times \mathrm{Sym}(d)$. Here, $\mathrm{SPD}(d)$ is the space of $\mathbb{R}^{d \times d}$ symmetric positive definite matrices, and $\mathrm{Sym}(d)$ is the space of $\mathbb{R}^{d \times d}$ symmetric matrices. By writing an affine map as $T(x) = a + S(x - m)$ for fixed $m$ (which is the mean of $\mu$), any affine map starting at $\mu = N(m, \Sigma)$ can be parameterized by $(a, S)$. Under this identification, we can view $BW(\mathbb{R}^d)$ space as a product Riemannian manifold

of $\mathbb{R}^d \times \mathrm{SPD}(d)$ (Appendix A.1.2). Then one can parameterize every quantity in Appendix A.2 by this product manifold sense. For instance, the vector corresponding to the optimal transport map is $(m_1, \Sigma_0^{-1/2}(\Sigma_0^{1/2}\Sigma_1\Sigma_0^{1/2})^{1/2}\Sigma_0^{-1/2})$.

Then, we can define Riemannian metric, exponential map, logarithmic map, and Bures-Wasserstein gradient in terms of parameters as well.

**Definition A.36** (Riemannian metric of Bures-Wasserstein space). *Let $\mu = N(m, \Sigma)$. The Riemannian metric of $BW(\mathbb{R}^d)$ is define by*

$$\langle (a_0, S_0), (a_1, S_1) \rangle_\mu = \langle a_0, a_1 \rangle_{\mathbb{R}^d} + tr(S_0 \Sigma S_1).$$

**Definition A.37.** *[LCB$^+$22, Appendix B.3] Let $\mu_i = N(m_i, \Sigma_i)$. The exponential map and a logarithm map in $BW(\mathbb{R}^d)$ are defined by*

$$\exp_{\mu_0}((a, S)) = N\left(a + m_0, (S + I)\Sigma_0(S + I)\right),$$
$$\log_{\mu_0}(\mu_1) = (m_1 - m_0, \Sigma_0^{-1/2}(\Sigma_0^{1/2}\Sigma_1\Sigma_0^{1/2})^{1/2}\Sigma_0^{-1/2} - I).$$

Note the exponential map is only defined when $\det(S + I) \neq 0$ (more precisely, for all geodesic interpolations to be well-defined, one needs $-I \prec S$). Hence, Bures-Wasserstein space is *not* geodesically complete.

**Definition A.38.** *[LCB$^+$22, Appendix B.3] Bures-Wasserstein metric of the functional $\mathcal{F}$ can be written as a function on $\mathbb{R}^d \times \mathrm{SPD}(d)$, the space of the mean and covariance. Then, for $m \in \mathbb{R}$ and $\Sigma \in \mathrm{SPD}(d)$,*

$$\mathrm{Grad}_{\mathrm{BW}}\, \mathcal{F}(m, \Sigma) = (\nabla_m \mathcal{F}(m, \Sigma), 2\nabla_\Sigma \mathcal{F}(m, \Sigma)).$$

See [LCB$^+$22, DBCS23] for further discussion.

Using the isometry between the function representation and the vector-matrix representation of $T_p BW(\mathbb{R}^d)$, we can define the following operation, which can be used to construct the (un-projected) parallel transport.

**Definition A.39.** *For $(a, S) \in T_{\mu_1} BW(\mathbb{R}^d)$ and $(b, R) \in T_{\mu_0} BW(\mathbb{R}^d)$, we have the following operation.*

$$(a, S) \circ (b, R) = (a + Sb - Sm_1, SR).$$

*In particular,*

$$\Gamma_{(m_1, \Sigma_1)}^{(m_0, \Sigma_0)}(a, S) = (a, S\Sigma_0^{-1/2}(\Sigma_0^{1/2}\Sigma_1\Sigma_0^{1/2})^{1/2}\Sigma_0^{-1/2}).$$

Some works adopt an alternative definition of the Bures–Wasserstein metric; we make a remark that this definition is equivalent to the one we present here. This remark plays a pivotal role when we conduct actual calculation in $BW(\mathbb{R}^d)$ space (Appendix C).

**Remark A.40** (Equivalent formulation of Bures-Wasserstein metric). *In some works (e.g., [HMJG21]), $BW(\mathbb{R}^d)$ metric is defined as $\langle (a, S), (b, R) \rangle_\mu = \langle a, b \rangle_{\mathbb{R}^d} + \frac{1}{2}\,tr(L_\Sigma(S)R)$, where $L_\Sigma(S)$ is the Lyapunov operator defined via the solution of $L_\Sigma(S)\Sigma + \Sigma L_\Sigma(S) = S$. While it has the different form with what we introduced earlier, these two formulations turned out to be equivalent: our formulation is from Wasserstein perspective, and the other formulation is from Riemannian perspective. In our setup, we define the tangent vector to directly parameterize the optimal transport map. That said, this does not directly fit with the Riemannian framework. For instance, if we consider the curve $\gamma(t) = \exp_\mu(t(a, S))$ defined by our exponential map, then the velocity at $t = 0$ is $\dot{\gamma}(0) = (a, S\Sigma + \Sigma S)$, which does not coincide with the tangent vector $(a, S)$. By contrast, under the Lyapunov operator based definition, the initial velocity is exactly $\dot{\gamma}(0) = (a, S)$. However, since there is a one-to-one correspondence between $S\Sigma + \Sigma S$ and $S$ for a given $\Sigma$, one may regard these two definitions as equivalent by identifying the tangent vector with $v_0 = S$ whenever the velocity $\dot{\gamma}(0) = S\Sigma + \Sigma S$ appears. One can change all corresponding quantities accordingly, and these two definitions turned out to be equivalent. We have chosen our formulation because it leads to a simpler algorithm (5.3) that avoids solving the Lyapunov equation.*

Lastly, we end up with the analysis of the curvature of $BW(\mathbb{R}^d)$. In particular, we show the result that even $BW(\mathbb{R}^d)$ space does not allow the curvature upper bound, indicating that the 2-Wasserstein space does not have the curvature upper bound as well.

By applying Corollary A.22 and the flatness of Euclidean space, we obtain the following result:

**Corollary A.41.** *For any $\mu \in BW(\mathbb{R}^d)$ and $\{(a, S), (b, R)\}$ orthonormal vectors in $T_\mu BW(\mathbb{R}^d) = \mathbb{R}^d \times \text{Sym}(d)$,*

$$K_{BW(\mathbb{R}^d)}\left((a, S), (b, R)\right) = (\text{tr}(S\Sigma S)\,\text{tr}(R\Sigma R) - \text{tr}(S\Sigma R)^2)K_{\text{Sym}_+(\mathbb{R}^{d\times d})}(S, R).$$

Therefore, to analyze the curvature of $BW(\mathbb{R}^d)$, it is sufficient to analyze the space of positive definite matrices, without accounting for the mean component. In this regard, without the loss of generality we consider $\mu = N(0, \Sigma)$. Then, since $\Sigma$ is a symmetric positive definite matrix, it is diagonalizable, and therefore we can write $\Sigma = PD(\lambda_i)P^T$ with $P$ being an orthogonal matrix and all real positive eigenvalues $\lambda_i$. Then, it is known that $\text{Sym}(d)$ is spanned by the following orthonormal basis [Tak09]:

$$\left\{ e_+ = \frac{P(E_{11} + E_{dd})P^T}{\sqrt{\lambda_1 + \lambda_d}}, e_{ij} = \frac{P(E_{ii} - E_{jj})P^T}{\sqrt{\lambda_i + \lambda_j}}, f_{ij} = \frac{P(E_{ij} + E_{ji})P^T}{\sqrt{\lambda_i + \lambda_j}} \right\}_{1 \leq i, j \leq d}$$

where $E_{ij}$ is a matrix with only its $(i, j)$ entry is 1 and 0 otherwise.

Using this orthonormal basis, we can characterize all of the sectional curvature in $\text{SPD}(d)$ as follows:

**Lemma A.42.** *[Tak09][Sectional curvature of Bures-Wasserstein space]*

$$K(e_+, f_{ij}) = \frac{3\lambda_i\lambda_j}{(\lambda_i + \lambda_j)^2(\lambda_1 + \lambda_d)} \qquad (i = 1 \text{ or } j = d),$$

$$K(e_{ik}, f_{ij}) = \frac{3\lambda_i\lambda_j}{(\lambda_i + \lambda_j)^2(\lambda_i + \lambda_k)} \qquad (j \neq k),$$

$$K(e_{ij}, f_{ij}) = \frac{12\lambda_i\lambda_j}{(\lambda_i + \lambda_j)^3},$$

$$K(f_{ij}, f_{ik}) = \frac{3\lambda_j\lambda_k}{(\lambda_i + \lambda_j)(\lambda_j + \lambda_k)(\lambda_i + \lambda_k)} \qquad (j \neq k),$$

$$K(\text{any other combinations}) = 0.$$

This explicit form indicates that the curvature upper bound at $\mu$ depends on the smallest eigenvalue of the covariance matrix $\Sigma$. Since the space of symmetric positive definite matrices does not have the uniform positive eigenvalue lower bound, $BW(\mathbb{R}^d)$ does not have the uniform curvature upper bound. See [Tak09] for more discussions on the sectional curvature of $BW(\mathbb{R}^d)$ space.

In general, the curvature of a submanifold and the curvature of its ambient manifold needs not be the same. However, if the submanifold is totally geodesic, by Gauss formula [Lee18, Theorem 8.2] and the fact that the second fundamental form vanishes [Lee18, Exercise 8.4], the curvature of the submanifold coincides to the curvature of the ambient manifold. Since $BW(\mathbb{R}^d)$ is a totally geodesic submanifold of the 2-Wasserstein space [CL20], Lemma A.42 implies that 2-Wasserstein space also does not have the sectional curvature upper bound.

# B  Deferred proofs

## B.1  Deferred proofs for Section 3

*Proof of Proposition 3.7.* If $f$ is $\mathcal{VT}$-geodesically smooth with base $b \in M$ and $\mathcal{VT}$-geodesically convex, then we take

$$z = \exp_b\left(-\frac{1}{L}\left(\Gamma_y^b \text{Grad}\, f(y) - \Gamma_x^b \text{Grad}\, f(x)\right) + \log_b y\right).$$

$z \in N$ from the condition we assumed. Then,

$$
\begin{aligned}
f(x) - f(y) &= f(x) - f(z) + f(z) - f(y) \\
&\leq -\left\langle \mathcal{VT}_x^b \operatorname{Grad} f(x), \log_b z - \log_b x \right\rangle_b + \left\langle \mathcal{VT}_y^b \operatorname{Grad} f(y), \log_b z - \log_b y \right\rangle_b + \frac{L}{2} \left\| \log_b z - \log_b y \right\|_b^2 \\
&= \left\langle \mathcal{VT}_x^b \operatorname{Grad} f(x), \log_b x - \log_b y \right\rangle_b - \left\langle \mathcal{VT}_x^b \operatorname{Grad} f(x), \log_b z - \log_b y \right\rangle_b \\
&\quad + \left\langle \mathcal{VT}_y^b \operatorname{Grad} f(y), \log_b z - \log_b y \right\rangle_b + \frac{L}{2} \left\| \log_b z - \log_b y \right\|_b^2 \\
&= \left\langle \mathcal{VT}_x^b \operatorname{Grad} f(x), \log_b x - \log_b y \right\rangle_b - \frac{1}{2L} \left\| \mathcal{VT}_y^b \operatorname{Grad} f(y) - \mathcal{VT}_x^b \operatorname{Grad} f(x) \right\|_b^2 .
\end{aligned}
$$

∎

### B.1.1 Additional discussions on Proposition 3.7

In Euclidean space, the important property of convex and $L$-smooth function is that the following inequality holds [Nes14, Theorem 2.1.5]:

$$
f(y) - f(x) - \langle \nabla f(x), y - x \rangle - \frac{1}{2L} \left\| \nabla f(y) - \nabla f(x) \right\|^2 \geq 0. \tag{B.1}
$$

Such inequality is sometimes referred as co-coercivity type inequality or interpolating inequality, and works as the core inequality for some optimization algorithms, e.g. algorithms based on Performance Estimation Problem (PEP) [THG17, AP24c] or some recent adaptive methods [SM25]. A natural Riemannian analogue of (B.1) would be written as follows:

$$
f(y) - f(x) - \langle \operatorname{Grad} f(x), \log_x y \rangle - \frac{1}{2L} \left\| \Gamma_y^x \operatorname{Grad} f(y) - \operatorname{Grad} f(x) \right\|^2 \geq 0. \tag{B.2}
$$

We show the same proof strategy in Appendix B.2 can be applied to obtain the sufficient condition for (B.2) as the special case.

In fact, we provide more general result. We show, if $f$ is generalized geodesically convex and for all $x, y \in N$, and $f$ satisfies $z := \exp_y \left( -\frac{1}{L} \left( \operatorname{Grad} f(y) - \Gamma_x^y \operatorname{Grad} f(x) \right) \right) \in N$, then we show (B.2) is equivalent to geodesic $L$-smoothness. Hence, generalized geodesic convexity and geodesic smoothness, under additional technical assumption, imply (B.2).

For this result, we will use the $L$-Lipchitz gradient for the definition of geodesic $L$-smoothness, as defined in Definition A.17, Recall $L$-Lipchitz gradient implies quadratic upper bound (Definition 2.2) by Lemma A.18.

We need to introduce the notion of *co-coercivity*.

**Definition B.1** (Geodesic co-coercivity). *A differentiable function $f : N \to \mathbb{R}$ is called geodesically co-coercive if for all $x, y \in N$*

$$
\left\langle \Gamma_y^x \operatorname{Grad} f(y) - \operatorname{Grad} f(x), \log_x y \right\rangle \geq \frac{1}{L} \left\| \Gamma_y^x \operatorname{Grad} f(y) - \operatorname{Grad} f(x) \right\|^2 .
$$

The geodesic co-coercivity condition links $L$-smoothness and (B.2). The next lemma is a general version of Proposition 3.7, which shows the relationship between $L$-smoothness, co-coercivity, and (B.2).

**Lemma B.2.** *For a differentiable function $f : N \to \mathbb{R}$, The below relationship holds:*

$$
(\text{B.2}) \overset{(i)}{\Rightarrow} \text{geodesic co-coercivity} \overset{(ii)}{\Rightarrow} \text{geodesic } L\text{-smoothness}.
$$

*In addition, suppose for all $x, y \in N$, $f$ satisfies $z := \exp_y \left( -\frac{1}{L} \left( \operatorname{Grad} f(y) - \Gamma_x^y \operatorname{Grad} f(x) \right) \right) \in N$. Then, if $f$ is generalized geodesically convex,*

$$
\text{geodesic } L\text{-smoothness} \overset{(iii)}{\Rightarrow} (\text{B.2}).
$$

*Proof.* **(i)**: By applying (B.2) for $(x, y)$ and $(y, x)$ and using Lemma C.9, one gets

$$f(y) - f(x) - \langle \text{Grad} \, f(x), \log_x y \rangle - \frac{1}{2L} \left\| \Gamma_y^x \, \text{Grad} \, f(y) - \text{Grad} \, f(x) \right\|^2 \geq 0,$$

$$f(x) - f(y) + \langle \Gamma_y^x \, \text{Grad} \, f(y), \log_x y \rangle - \frac{1}{2L} \left\| \Gamma_y^x \, \text{Grad} \, f(y) - \text{Grad} \, f(x) \right\|^2 \geq 0.$$

Summing up two inequalities, one gets

$$\langle \Gamma_y^x \, \text{Grad} \, f(y) - \text{Grad} \, f(x), \log_x y \rangle \geq \frac{1}{L} \left\| \Gamma_y^x \, \text{Grad} \, f(y) - \text{Grad} \, f(x) \right\|^2.$$

**(ii)**: Using Cauchy-Schwartz inequality on the co-coercivity condition, one gets

$$\frac{1}{L} \left\| \Gamma_y^x \, \text{Grad} \, f(y) - \text{Grad} \, f(x) \right\|^2 \leq \left\| \Gamma_y^x \, \text{Grad} \, f(y) - \text{Grad} \, f(x) \right\| \left\| \log_x y \right\|.$$

Since $\|\log_x y\| = d(x, y)$, one gets the result.

**(iii)**: We follow the same proof strategy as in the proof of Proposition 3.7. Take $z = \exp_y \left( -\frac{1}{L} \left( \text{Grad} \, f(y) - \Gamma_x^y \, \text{Grad} \, f(x) \right) \right)$. Write $f(x) - f(y) = f(x) - f(z) + f(z) - f(y)$. Then, using generalized geodesic convexity with base $y$ and Lemma A.18,

$$\begin{aligned}
f(x) - f(y) &= f(x) - f(z) + f(z) - f(y) \\
&\leq -\langle \Gamma_x^y \, \text{Grad} \, f(x), \log_y z - \log_y x \rangle + \langle \text{Grad} \, f(y), \log_y z \rangle + \frac{L}{2} \left\| \log_y z \right\|^2 \\
&= -\left\langle \Gamma_x^y \, \text{Grad} \, f(x), -\frac{1}{L} (\text{Grad} \, f(y) - \Gamma_x^y \, \text{Grad} \, f(x)) - \log_y x \right\rangle \\
&\quad + \left\langle \text{Grad} \, f(y), -\frac{1}{L} (\text{Grad} \, f(y) - \Gamma_x^y \, \text{Grad} \, f(x)) \right\rangle \\
&\quad + \frac{1}{2L} \left\| \text{Grad} \, f(y) - \Gamma_x^y \, \text{Grad} \, f(x) \right\|^2 \\
&= \langle \Gamma_x^y \, \text{Grad} \, f(x), \log_y x \rangle - \frac{1}{2L} \left\| \text{Grad} \, f(y) - \Gamma_x^y \, \text{Grad} \, f(x) \right\|^2 \\
&= -\langle \text{Grad} \, f(x), \log_x y \rangle - \frac{1}{2L} \left\| \Gamma_y^x \, \text{Grad} \, f(y) - \text{Grad} \, f(x) \right\|^2.
\end{aligned}$$

Here, we again used Lemma C.9 for the last equality. This is equivalent to the desired inequality. ∎

## B.2 Deferred proofs for Section 4

This appendix contains the proofs of Section 4. While the overall proofs follow [AP24c], to clarify that the Riemannian settings are properly taken into accout, we provide the full explicit proofs.

Before we proceed, we introduce simplified formulation of $Q_{ij;b}$. By simply expanding the squared norm term, one can write $Q_{ij;b}$ as follows:

$$\begin{aligned}
Q_{ij;b} = \; & 2f(x_i) - 2f(x_j) - 2 \left\langle \mathcal{VT}_{x_j}^b \, \text{Grad} \, f(x_j), \log_b x_i - \log_b x_j \right\rangle_b \\
& - \left\| \mathcal{VT}_{x_i}^b \, \text{Grad} \, f(x_i) \right\|_{x_i}^2 - \left\| \mathcal{VT}_{x_j}^b \, \text{Grad} \, f(x_j) \right\|_{x_j}^2 + 2 \left\langle \mathcal{VT}_{x_j}^b \, \text{Grad} \, f(x_j), \mathcal{VT}_{x_i}^b \, \text{Grad} \, f(x_i) \right\rangle_b.
\end{aligned}$$

This formulation will be used frequently for the rest of the proof.

In addition, we also use the following formula for the intermeidate calculation.

**Lemma B.3.** *Set* $L = 1$. *The following equality holds.*

$$\begin{aligned}
& \left\| \log_b x_* - \log_b x_n + (2r_k)^{-1} \mathcal{VT}_{x_n}^b \, \text{Grad} \, f(x_n) \right\|_b^2 - \left\| \log_b x_* - \log_b x_0 \right\|_b^2. \\
&= (4r_k^2)^{-1} \left\| \mathcal{VT}_n^b \, \text{Grad} \, f(x_n) \right\|_b^2 + r_k^{-1} \left\langle \mathcal{VT}_{x_n}^b \, \text{Grad} \, f(x_n), \log_b x_* - \log_b x_n \right\rangle_b \\
&\quad + \sum_{i=0}^{n-1} \eta_i^2 \left\| \mathcal{VT}_i^b \, \text{Grad} \, f(x_i) \right\|_b^2 + 2 \sum_{i=0}^{n-1} \eta_i \left\langle \mathcal{VT}_{x_i}^b \, \text{Grad} \, f(x_i), \log_b x_* - \log_b x_i \right\rangle_b
\end{aligned}$$

*Proof of Lemma B.3.*

$$LHS = - \|\log_b x_* - \log_b x_0\|_b^2 + \|\log_b x_* - \log_b x_n\|_b^2 + \frac{1}{4r_k^2} \left\| \mathcal{VT}_n^b \operatorname{Grad} f(x_n) \right\|_b^2$$

$$+ \frac{1}{r_k} \left\langle \log_b x_* - \log_b x_n, \mathcal{VT}_n^b \operatorname{Grad} f(x_n) \right\rangle_b$$

$$= - \|\log_b x_* - \log_b x_0\|_b^2 + \|\log_b x_* - \log_b x_{n-1}\|_b^2 + \|\log_b x_n - \log_b x_{n-1}\|_b^2$$

$$- 2 \left\langle \log_b x_* - \log_b x_{n-1}, \log_b x_n - \log_b x_{n-1} \right\rangle_b$$

$$+ \frac{1}{4r_k^2} \left\| \mathcal{VT}_n^b \operatorname{Grad} f(x_n) \right\|_b^2 + \frac{1}{r_k} \left\langle \log_b x_* - \log_b x_n, \mathcal{VT}_{x_n}^b \operatorname{Grad} f(x_n) \right\rangle_b$$

$$\overset{(i)}{=} - \|\log_b x_* - \log_b x_0\|_b^2 + \|\log_b x_* - \log_b x_{n-1}\|_b^2 + \eta_{n-1}^2 \left\| \mathcal{VT}_{n-1}^b \operatorname{Grad} f(x_{n-1}) \right\|_b^2$$

$$+ 2\eta_{n-1} \left\langle \log_b x_* - \log_b x_{n-1}, \mathcal{VT}_{x_{n-1}}^b \operatorname{Grad} f(x_{n-1}) \right\rangle_b$$

$$+ \frac{1}{4r_k^2} \left\| \mathcal{VT}_n^b \operatorname{Grad} f(x_n) \right\|_b^2 + \frac{1}{r_k} \left\langle \log_b x_* - \log_b x_n, \mathcal{VT}_n^b \operatorname{Grad} f(x_n) \right\rangle_b$$

For (i), we used $\log_b x_i - \log_b x_{i-1} = -\eta_{i-1} \mathcal{VT}_{x_{i-1}}^b \operatorname{Grad} f(x_{i-1})$.

Now, do the same decomposition on $\log_b x_* - \log_b x_{n-1}$ by $(\log_b x_* - \log_b x_{n-2}) - (\log_b x_{n-1} - \log_b x_{n-2})$ and use $\log_b x_i - \log_b x_{i-1} = -\eta_{i-1} \mathcal{VT}_{x_{i-1}}^b \operatorname{Grad} f(x_{i-1})$. Iteratively conducting this procedure until $x_0$ yields the desired claim. ∎

The following lemma is the main component of the proof of Theorem 4.1. Without loss of generality, set $L = 1$. We also set $n = 2^k - 1$ for some $k \in \mathbb{N}$.

**Lemma B.4.** *Let the conditions of Theorem 4.1 be true. Then, for suitably chosen $\lambda_{ij} \geq 0$,*

$$\sum_{i,j=0,\ldots,n,*} \lambda_{ij} Q_{ij;b} = \frac{1}{r_k} (f(x_*) - f(x_n)) + \|\log_b x_* - \log_b x_0\|_b^2 \tag{B.3}$$

$$- \left\| \log_b x_* - \log_b x_n + \frac{1}{2r_k} \mathcal{VT}_{x_n}^b \operatorname{Grad} f(x_n) \right\|_b^2.$$

Once we establish Lemma B.4, the proof of Theorem 4.1 is direct.

*Proof of Theorem 4.1.* First consider the case $L = 1$. By Lemma B.4 and (3.1), one gets

$$f(x_n) - f(x_*) \leq r_k \|\log_b x_* - \log_b x_0\|_b^2.$$

This proves the desired convergence rate for $L = 1$.

For general $L$, let $g = \frac{1}{L} f$. Then, by the linearity of the vector transport (which is assumed) and the Riemannian gradient, $g$ satisfies (3.1) with $L = 1$. By applying $L = 1$ case on $g$, one gets

$$\frac{1}{L} (f(x_n) - f(x_*)) = g(x_n) - g(x_*) \leq r_k \|\log_b x_0 - \log_b x_*\|_b^2.$$

∎

To prove Lemma B.4, as in [AP24c] we will prove it by induction.

We begin with the base step of the induction.

**Base Step** First, we show (B.3) is valid for $n = 1$ ($k = 1$).

**Lemma B.5.** *For any arbitrary initialization $x_0 \in N$, consider the following VTRGD update (4.1).*

$$x_1 = \exp_b \left( \log_b x_0 - \eta_0 \mathcal{VT}_{x_0}^b \operatorname{Grad} f(x_0) \right),$$

*where $\eta_0 = \rho - 1$. Choose $\lambda_{ij}$ the same as in [AP24c, Example 2], i.e.,*

$$\begin{pmatrix} \lambda_{00} & \lambda_{01} & \lambda_{0*} \\ \lambda_{10} & \lambda_{11} & \lambda_{1*} \\ \lambda_{*0} & \lambda_{*1} & \lambda_{**} \end{pmatrix} = \begin{pmatrix} 0 & \rho & 0 \\ 1 & 0 & \rho - 1 \\ \rho - 1 & \frac{1}{2r_1} & 0 \end{pmatrix}. \tag{B.4}$$

*Then, the equality (B.3) holds.*

*Proof of Lemma B.5.* Observe the following calculation:

$$\sum_{i,j} \lambda_{ij} Q_{ij;b} = \rho Q_{01;b} + Q_{10;b} + (\rho - 1)Q_{1*;b} + (\rho - 1)Q_{*0;b} + \frac{1}{2r_1} Q_{*1;b}$$

$$= \frac{f(x_*) - f(x_1)}{r_1} - 2\rho \left\langle \mathcal{VT}_{x_1}^b \operatorname{Grad} f(x_1), \log_b x_0 - \log_b x_1 \right\rangle_b - \rho \left\| \mathcal{VT}_{x_1}^b \operatorname{Grad} f(x_1) \right\|_b^2$$

$$- \rho \left\| \mathcal{VT}_{x_0}^b \operatorname{Grad} f(x_0) \right\|_b^2 + 2\rho \left\langle \mathcal{VT}_{x_1}^b \operatorname{Grad} f(x_1), \mathcal{VT}_{x_0}^b \operatorname{Grad} f(x_0) \right\rangle_b$$

$$- 2\rho \left\langle \mathcal{VT}_{x_0}^b \operatorname{Grad} f(x_0), \log_b x_1 - \log_b x_0 \right\rangle_b - \left\| \mathcal{VT}_{x_1}^b \operatorname{Grad} f(x_1) \right\|_b^2 - \left\| \mathcal{VT}_{x_0}^b \operatorname{Grad} f(x_0) \right\|_b^2$$

$$+ 2 \left\langle \mathcal{VT}_{x_1}^b \operatorname{Grad} f(x_1), \mathcal{VT}_{x_0}^b \operatorname{Grad} f(x_0) \right\rangle_b - (\rho - 1) \left\| \mathcal{VT}_{x_1}^b \operatorname{Grad} f(x_1) \right\|_b^2$$

$$- 2(\rho - 1) \left\langle \mathcal{VT}_{x_0}^b \operatorname{Grad} f(x_0), \log_b x_* - \log_b x_0 \right\rangle_b - (\rho - 1) \left\| \mathcal{VT}_{x_0}^b \operatorname{Grad} f(x_0) \right\|_b^2$$

$$- \frac{1}{r_k} \left\langle \mathcal{VT}_{x_1}^b \operatorname{Grad} f(x_1), \log_b x_* - \log_b x_1 \right\rangle_b - \frac{1}{2r_1} \left\| \mathcal{VT}_{x_1}^b \operatorname{Grad} f(x_1) \right\|_b^2$$

$$\overset{(i)}{=} \frac{f(x_*) - f(x_1)}{r_1} + (2 + 2\rho - 2\eta_0 \rho) \left\langle \mathcal{VT}_{x_1}^b \operatorname{Grad} f(x_1), \mathcal{VT}_{x_0}^b \operatorname{Grad} f(x_0) \right\rangle_b$$

$$- \left( \frac{1}{2r_1} + 2\rho \right) \left\| \mathcal{VT}_{x_1}^b \operatorname{Grad} f(x_1) \right\|_b^2 - (2\rho - 2\eta_0) \left\| \mathcal{VT}_{x_0}^b \operatorname{Grad} f(x_0) \right\|_b^2$$

$$- 2(\rho - 1) \left\langle \mathcal{VT}_{x_0}^b \operatorname{Grad} f(x_0), \log_b x_* - \log_b x_0 \right\rangle_b - \frac{1}{r_1} \left\langle \mathcal{VT}_{x_1}^b \operatorname{Grad} f(x_1), \log_b x_* - \log_b x_1 \right\rangle_b$$

$$\overset{(ii)}{=} \frac{f(x_*) - f(x_1)}{r_1} - \eta_0^2 \left\| \mathcal{VT}_{x_0}^b \operatorname{Grad} f(x_0) \right\|_b^2 - \frac{1}{4r_1^2} \left\| \mathcal{VT}_{x_1}^b \operatorname{Grad} f(x_1) \right\|_b^2$$

$$- 2\eta_0 \left\langle \mathcal{VT}_{x_0}^b \operatorname{Grad} f(x_0), \log_b x_* - \log_b x_0 \right\rangle_b - \frac{1}{r_1} \left\langle \mathcal{VT}_{x_1}^b \operatorname{Grad} f(x_1), \log_b x_* - \log_b x_1 \right\rangle_b$$

$$\overset{(iii)}{=} RHS.$$

Here, for (i) we used again $\log_b x_{i+1} - \log_b x_i = -(\log_b x_i - \log_b x_{i+1}) = -\eta_i \Gamma_{x_i}^b \operatorname{Grad} f(x_i)$, and for (ii) we used the explicit quantity of $\eta_0, \rho,$ and $r_1$. (iii) holds from Lemma B.3. ∎

**Induction step** Lemma B.5 validates that (B.3) holds for the base case $n = 1$. In this section, given we have the inequality (B.3) for $n = 2^k - 1$ number of iterates, we show by merging two silver stepsizes, one can get (B.3) for $2n + 1 = 2^{k+1} - 1$ number of iterates.

**Lemma B.6.** *Fix* $n = 2^k - 1$. *Take* $\{x_i\}_{i=0,\ldots,n} \subset N$ *a sequence induced from the silver stepsize VTRGD. Suppose there exist* $\lambda_{ij}^{(k)} \geq 0$ *such that* (B.3) *holds. Write*

$$\sigma_{ij} = \lambda_{ij}^{(k)} \mathbb{1}_{\{i,j=0,\ldots,n,*\}} + (1 + 2\rho)\lambda_{i-n-1,j-n-1}^{(k)} \mathbb{1}_{\{i,j=n+1,\ldots,2n+1,*\}}$$

*where* $* - n - 1$ *is understood to mean* $*$. *Define*

$$\lambda_{ij}^{(k+1)} := \sigma_{ij} + \rho \eta_j \mathbb{1}_{\{i=n,2n+1,j=n+1,\ldots,2n\}} - 2\rho \eta_j \mathbb{1}_{\{i=*,j=n+1,\ldots,2n\}}$$

$$+ \left( 1 + \rho^{k-1} - \frac{1}{2r_k} \right) \mathbb{1}_{\{i=*,j=n\}} + \left( \frac{1}{2r_{k+1}} - \frac{1+2\rho}{2r_k} \right) \mathbb{1}_{\{i=*,j=2n+1\}}$$

$$+ \rho \mathbb{1}_{\{i=n,j=2n+1\}} + \rho^k \mathbb{1}_{\{i=2n+1,j=n\}}$$

$$+ (1 - \rho^k) \mathbb{1}_{\{i=n,j=*\}} + (2\rho - \sqrt{2}\rho^{k+1}) \mathbb{1}_{\{i=2n+1,j=*\}}.$$

*Then,* $\lambda_{ij}^{(k+1)}$ *satisfies*

$$\sum_{i,j=0,\ldots,2n+1,*} \lambda_{ij}^{(k+1)} Q_{ij;b} = \frac{f(x_*) - f(x_{2n+1})}{r_{k+1}} + \|\log_b x_* - \log_b x_0\|_b^2$$

$$- \left\| \log_b x_* - \log_b x_{2n+1} + \frac{1}{2r_{k+1}} \mathcal{VT}^b_{x_{2n+1}} \operatorname{Grad} f(x_{2n+1}) \right\|_b^2.$$

*In particular, if $\lambda_{ij}^{(1)}$ is chosen as in Lemma B.5, then $\lambda_{ij}^{(k)} \geq 0$ for all $k \in \mathbb{N}$ and $i, j = 0, \ldots, 2^k - 1, *$.*

*Proof of Lemma B.6.* For the simplicity of the notation, we write $g_i := \operatorname{Grad} f(x_i)$.

From the construction of $\sigma_{ij}$, we have

$$\sum_{i,j=0,\ldots,2n+1,*} \sigma_{ij} Q_{ij;b} = \sum_{i,j=0,\ldots,n,*} \lambda_{ij}^{(k)} Q_{ij;b} + (1+2\rho) \sum_{i,j=n+1,\ldots,2n+1,*} \lambda_{i-n-1,j-n-1}^{(k)} Q_{ij;b}.$$

Since we assumed (B.3) in the induction, we have

$$\sum_{i,j=0,\ldots,2n+1,*} \sigma_{ij} Q_{ij;b} = \frac{1}{r_k}(f(x_*) - f(x_n)) + \|\log_b x_* - \log_b x_0\|_b^2$$

$$- \left\| \log_b x_* - \log_b x_n + \frac{1}{2r_k} \mathcal{VT}^b_{x_n} \operatorname{Grad} f(x_n) \right\|_b^2$$

$$+ \frac{1+2\rho}{r_k}(f(x_*) - f(x_{2n+1})) - (1+2\rho)\|\log_b x_* - \log_b x_{n+1}\|^2$$

$$- (1+2\rho)\left\| \log_b x_* - \log_b x_{2n+1} + \frac{1}{2r_k} \mathcal{VT}^b_{x_{2n+1}} \operatorname{Grad} f(x_{2n+1}) \right\|_b^2.$$

We subtract $\sum_{ij} \sigma_{ij} Q_{ij;b}$ from the RHS. Using Lemma B.3, one gets

$$RHS - \sum_{ij} \sigma_{ij} Q_{ij;b}$$

$$= \left( \frac{1}{r_{k+1}} - \frac{2+2\rho}{r_k} \right) f(x_*) + \frac{1}{r_k} f(x_n) + \left( \frac{1+2\rho}{r_k} - \frac{1}{r_{k+1}} \right) f(x_{2n+1})$$

$$+ 2\rho \sum_{i=n+1}^{2n} \eta_i^2 \left\| \mathcal{VT}^b_{x_i} \operatorname{Grad} f(x_i) \right\|_b^2 + 4\rho \sum_{i=n+1}^{2n} \eta_i \left\langle \log_b x_* - \log_b x_i, \mathcal{VT}^b_{x_i} \operatorname{Grad} f(x_i) \right\rangle_b$$

$$- \left( \eta_n^2 - \frac{1}{4r_k^2} \right) \left\| \mathcal{VT}^b_{x_n} \operatorname{Grad} f(x_n) \right\|_b^2 - \left( 2\eta_n - \frac{1}{r_k} \right) \left\langle \log_b x_* - \log_b x_n, \mathcal{VT}^b_{x_n} \operatorname{Grad} f(x_n) \right\rangle_b$$

$$- \left( \frac{1}{4r_{k+1}^2} - \frac{1+2\rho}{4r_k^2} \right) \left\| \mathcal{VT}^b_{x_{2n+1}} \operatorname{Grad} f(x_{2n+1}) \right\|_b^2$$

$$- \left( \frac{1}{r_{k+1}} - \frac{1+2\rho}{r_k} \right) \left\langle \log_b x_* - \log_b x_{2n+1}, \mathcal{VT}^b_{x_{2n+1}} \operatorname{Grad} f(x_{2n+1}) \right\rangle_b.$$

We now subtract the rest of terms in LHS. After careful calculations, one gets

$$RHS - \sum_{i,j} \lambda_{ij}^{(k+1)} Q_{ij;b} = 2\rho^k(1 + \rho^{k-1}) \left\| \mathcal{VT}^b_{x_n} \operatorname{Grad} f(x_n) \right\|_b^2 + 2\rho \sum_{i=n+1}^{2n} \eta_i^2 \left\| \mathcal{VT}^b_{x_i} \operatorname{Grad} f(x_i) \right\|_b^2$$

$$+ 2\rho \sum_{i=n+1}^{2n} \eta_i \left\langle \mathcal{VT}^b_{x_i} \operatorname{Grad} f(x_i), \log_b x_n - \log_b x_i \right\rangle_b$$

$$+ 2\rho \sum_{i=n+1}^{2n} \eta_i \left\langle \mathcal{VT}^b_{x_i} \operatorname{Grad} f(x_i), \log_b x_{2n+1} - \log_b x_i \right\rangle_b$$

$$- 2\rho \sum_{i=n+1}^{2n} \eta_i \left\langle \mathcal{VT}^b_{x_i} \operatorname{Grad} f(x_i), \mathcal{VT}^b_{x_n} \operatorname{Grad} f(x_n) \right\rangle_b$$

$$- 2\rho \sum_{i=n+1}^{2n} \eta_i \left\langle \mathcal{VT}^b_{x_i} \operatorname{Grad} f(x_i), \mathcal{VT}^b_{x_{2n+1}} \operatorname{Grad} f(x_{2n+1}) \right\rangle_b$$

$$+ 2\rho \left\langle \mathcal{VT}^b_{x_{2n+1}} \operatorname{Grad} f(x_{2n+1}), \log_b x_n - \log_b x_{2n+1} \right\rangle_b$$

$$+ 2\rho^k \left\langle \mathcal{VT}^b_{x_n} \operatorname{Grad} f(x_n), \log_b x_{2n+1} - \log_b x_n \right\rangle_b$$

$$- 2\rho(\rho^{k-1} + 1) \left\langle \mathcal{VT}^b_{x_n} \operatorname{Grad} f(x_n), \mathcal{VT}^b_{x_{2n+1}} \operatorname{Grad} f(x_{2n+1}) \right\rangle_b =: A.$$

We show $A = 0$, which implies the desired equality. To this end, note $\mathcal{VT}^b_{x_i} \operatorname{Grad} f(x_i) = -\frac{1}{\eta_i}\left(\log_b x_{i+1} - \log_b x_i\right)$. Plug-in this quantity to $A$. Then, the second, third, and fourth terms can be rewritten as

$$2\rho \sum_{i=n+1}^{2n} \eta_i^2 \left\| \mathcal{VT}^b_{x_i} \operatorname{Grad} f(x_i) \right\|_b^2 + 2\rho \sum_{i=n+1}^{2n} \eta_i \left\langle \mathcal{VT}^b_{x_i} \operatorname{Grad} f(x_i), \log_b x_n - \log_b x_i \right\rangle_b$$

$$+ 2\rho \sum_{i=n+1}^{2n} \eta_i \left\langle \mathcal{VT}^b_{x_i} \operatorname{Grad} f(x_i), \log_b x_{2n+1} - \log_b x_i \right\rangle_b$$

$$= \rho \sum_{i=n+1}^{2n} \left( 2 \left\| \log_b x_{i+1} - \log_b x_i \right\|_b^2 - 2 \left\langle \log_b x_{i+1} - \log_b x_i, \log_b x_n - \log_b x_i \right\rangle_b \right.$$

$$\left. - 2 \left\langle \log_b x_{i+1} - \log_b x_i, \log_b x_{2n+1} - \log_b x_i \right\rangle_b \right)$$

(B.5)

$$= \rho \sum_{i=n+1}^{2n} \left( \left\| \log_b x_n - \log_b x_{i+1} \right\|_b^2 - \left\| \log_b x_n - \log_b x_i \right\|_b^2 \right.$$

$$\left. + \left\| \log_b x_{2n+1} - \log_b x_{i+1} \right\|_b^2 - \left\| \log_b x_{2n+1} - \log_b x_i \right\|_b^2 \right)$$

$$\overset{\text{(i)}}{=} \rho \left( \left\| \log_b x_n - \log_b x_{2n+1} \right\|_b^2 - \left\| \log_b x_n - \log_b x_{n+1} \right\|_b^2 - \left\| \log_b x_{2n+1} - \log_b x_{n+1} \right\|_b^2 \right)$$

$$\overset{\text{(ii)}}{=} -2\rho \left\langle \log_b x_n - \log_b x_{n+1}, \log_b x_{2n+1} - \log_b x_{n+1} \right\rangle_b$$

$$\overset{\text{(iii)}}{=} 2\rho\eta_n \sum_{i=n+1}^{2n} \eta_i \left\langle \mathcal{VT}^b_{x_n} \operatorname{Grad} f(x_n), \mathcal{VT}^b_{x_i} \operatorname{Grad} f(x_i) \right\rangle_b.$$

For (i) we used the telescoping sum, for (ii) we used the fact

$$\|a - b\|^2 - \|a - c\|^2 - \|b - c\|^2 = -2\left\langle a - c, b - c \right\rangle,$$

and for (iii) we used the fact $\log_b x_{2n+1} - \log_b x_{n+1} = -\sum_{i=n+1}^{2n} \eta_i \mathcal{VT}^b_{x_i} \operatorname{Grad} f(x_i)$.

Likewise, using the same fact,

$$2\rho \left\langle \mathcal{VT}^b_{x_{2n+1}} \operatorname{Grad} f(x_{2n+1}), \log_b x_n - \log_b x_{2n+1} \right\rangle_b = 2\rho \sum_{i=n}^{2n} \eta_i \left\langle \mathcal{VT}^b_{x_{2n+1}} \operatorname{Grad} f(x_{2n+1}), \mathcal{VT}^b_{x_i} \operatorname{Grad} f(x_i) \right\rangle_b,$$

$$2\rho^k \left\langle \mathcal{VT}^b_{x_n} \operatorname{Grad} f(x_n), \log_b x_{2n+1} - \log_b x_n \right\rangle_b = -2\rho^k \sum_{i=n}^{2n} \eta_i \left\langle \mathcal{VT}^b_{x_n} \operatorname{Grad} f(x_n), \mathcal{VT}^b_{x_i} \operatorname{Grad} f(x_i) \right\rangle_b.$$

(B.6)

Pluggin-in Equation (B.5) and (B.6) into $A$ and using $\eta_n = 1 + \rho^{k-1}$ yields

$$A = 2\rho^k(1 + \rho^{k-1}) \left\| \mathcal{VT}^b_{x_n} \operatorname{Grad} f(x_n) \right\|_b^2 + 2\rho\eta_n \left\langle \mathcal{VT}^b_{x_{2n+1}} \operatorname{Grad} f(x_{2n+1}), \mathcal{VT}^b_{x_n} \operatorname{Grad} f(x_n) \right\rangle_b$$

$$- 2\rho^k\eta_n \left\langle \mathcal{VT}^b_{x_n} \operatorname{Grad} f(x_n), \mathcal{VT}^b_{x_n} \operatorname{Grad} f(x_n) \right\rangle_b$$

$$- 2\rho(1 + \rho^{k-1}) \left\langle \mathcal{VT}_{x_n}^b \operatorname{Grad} f(x_n), \mathcal{VT}_{x_{2n+1}}^b \operatorname{Grad} f(x_{2n+1}) \right\rangle_b$$

$$= 0$$

by using the facts that $\eta_n = 1 + \rho^{k-1}$. This completes the induction argument.

For non-negativeness of the coefficients, if the initialization is the same, then by [AP24c, Section 3.2] the non-negativeness is guaranteed as we selected the same coefficients. ∎

Equipped with Lemma B.5 B.6, Lemma B.4 is direct.

*Proof of Lemma B.4.* Start with the coefficients in Lemma B.5. The coefficients are clearly non-negative, and (B.3) holds for $n = k = 1$. Then, applying Lemma B.6 gives (B.3) for all $k \in \mathbb{N}$ and $n = 2^k - 1$. ∎

### B.2.1 Moving to strongly convex smooth functional: Restarting method

We now turn our attention to the geodesically strongly convex case. Although an alternative silver stepsize scheme has been proposed for strongly convex, smooth problems in the Euclidean setting [AP24b], the co-coercivity condition it relies on does not carry over to geodesically strongly convex, smooth problems on Riemannian manifolds. In contrast, for convex, smooth functions the co-coercivity condition admits a natural Riemannian interpretation via generalized geodesic convexity and geodesic smoothness (see Proposition 3.7.

Nevertheless, as noted in the main text, one can still employ the silver stepsize in the convex, smooth setting by combining it with the restarting technique of [OC15]. Theorem 4.2 shows that applying the restarting method [OC15] to our silver stepsize RGD yields an algorithm that also applies to geodesically strongly convex problems.

*Proof of Theorem 4.2.* Since $f$ is $\mathcal{VT}$-geodesically $\alpha$-strongly convex with base $b$ and $x_*$ is a minimizer,

$$f(x_m) - f(x_*) \geq \frac{\alpha}{2} \left\| \log_b x_m - \log_b x_* \right\|_b^2.$$

Therefore, for $m = 2^k - 1$, one gets

$$\left\| \log_b x_m - \log_b x_* \right\|_b^2 \leq \frac{2}{\alpha} \left( f(x_m) - f(x_*) \right) \leq 2\kappa r_k \left\| \log_b x_0 - \log_b x_* \right\|_b^2.$$

We first consider the case when $r = 0$, *i.e.*, we exactly iterate $m = 2^k - 1$ silver stepsize gradient descent $\ell$ times, by restarting the algorithm from the very last update of the previous runs. The total number of iterations becomes $n = m\ell = (2^k - 1)\ell$. Then, one gets the following bound for $n$ number of iterations:

$$\left\| \log_b x_n - \log_b x_* \right\|_b^2 \leq (2\kappa r_k)^\ell \left\| \log_b x_0 - \log_b x_* \right\|_b^2.$$

The term $(2\kappa r_k)^\ell$ is the rate we obtain for this algorithm. Now, one can optimize the choice of $k, \ell$ to get the tightest convergence rate, by solving

$$\min_{\ell, k} (2\kappa r_k)^\ell \quad \text{given} \quad (2^k - 1)\ell = n.$$

Specifically, we plug-in $k^* = \lceil \log_\rho \kappa \rceil + 1$. Observe $\rho^{k^*} + 1 \geq 1 + \rho^{\log_\rho \kappa} = 1 + \rho\kappa \geq \rho\kappa$. Then,

$$2\kappa r_{k^*} = \frac{2\kappa}{1 + \sqrt{4\rho^{2k^*} - 3}} \leq \frac{2\kappa}{\rho^{k^*} + 1} \leq \frac{2}{\rho} < 1$$

Now, since $\ell = \frac{n}{2^{k^*} - 1}$,

$$(2\kappa r_{k^*})^\ell = \exp\left( \ell \log (2\kappa r_{k^*}) \right) \leq \exp\left( \left( \log \frac{2}{\rho} \right) \frac{n}{2^{k^*} - 1} \right) \leq \exp\left( -\left( \log \frac{\rho}{2} \right) \frac{n}{\kappa^{\log_\rho 2}} \right)$$

which is the claimed rate.

For the $\epsilon$-approximate error, $\|\log_b x_m - \log_b x_*\|_b^2 \leq \epsilon$ holds whenever

$$\exp\left(-\left(\log\frac{\rho}{2}\right)\frac{n}{\kappa^{\log_\rho 2}}\right)\|\log_b x_0 - \log_b x_*\|_b^2 \leq \epsilon.$$

This is equivalent to

$$n \geq \frac{\kappa^{\log_\rho 2}}{\log(\rho/2)}\log\frac{\|\log_b x_0 - \log_b x_*\|_b^2}{\epsilon} = \Theta(\kappa^{\log_\rho 2}\log(1/\epsilon)).$$

∎

## B.3    Deferred proofs for Section 5

This appendix contains the proofs for the results in Section 5.

*Proof of Corollary 5.1.* First, choose the base $b \in \mathcal{P}_{2,ac}(\mathbb{R}^d)$ from the points that makes $\mathcal{F}$ being generalized geodesically $L$-smooth with base $b$. Then, if Proposition 3.7 holds, then the proof goes exactly same as in Theorem 4.1 and 4.2, once one substitutes the following quantities in the proof of Theorem 4.1 and 4.2 accordingly:

- Set $M = N = \mathcal{P}_{2,ac}(\mathbb{R}^d)$.

- Change the Riemannian metric by $\langle\cdot,\cdot\rangle_\mu = \langle\cdot,\cdot\rangle_{\mathcal{L}^2(\mu)} = \mathbb{E}_{x\sim\mu}[\langle\cdot(x),\cdot(x)\rangle]$.

- Set $\mathcal{VT}_{\mu_n}^b = T_{b,\mu_n}$, where $T_{b,\mu_n} = T_{b,\mu_{n-1}} - \frac{\eta_{n-1}}{L}\operatorname{Grad}_{W_2}\mathcal{F}(\mu_{n-1}) \circ T_{b,\mu_{n-1}}$ and $T_{b,\mu_0}$ is an arbitrary transport map (e.g., optimal transport map).

- Use $\exp_b(v) = (id + v)_{\#b}$.

- Take $\log_b \nu = T_{b,\nu} - id$.

- Set $\operatorname{Grad} f(x)$ to $\operatorname{Grad}_{W_2}\mathcal{F}(\mu)$, introduced in Definition A.34.

- To substitute the left hand side of Theorem 4.2 by $W_2^2(\mu_n, \mu_*)$, one uses the fact that $W_2^2(\mu,\nu) \leq \|T_{\mu,\nu} - id\|_\mu^2 \leq \|T_{b,\mu} - T_{b,\nu}\|_b^2$ for any transport map $T_{b,\mu}, T_{b,\nu}$ from the optimality of the optimal transport map and the Kantorovich coupling.

One thing to clarify is that 2-Wasserstein space is *not geodesically complete* [PZ20], so Assumptions 3.1 and 3.2 are not direct. However, we claim that the regularity conditions we imposed ensure that all the proof ingredients in Theorem 4.1, 4.2, and Proposition 3.7 to be valid.

To check the claim, first Proposition 3.7 will remain valid if the condition on $z$ in Proposition 3.7 is satisfied. Under this specific geometry, the condition on $z$ can be written as for all $\mu,\nu \in \mathcal{P}_{2,ac}(\mathbb{R}^d)$,

$$\sigma := \left(-\frac{1}{L}(\operatorname{Grad}_{W_2}\mathcal{F}(\nu) \circ T_{b,\nu} - \operatorname{Grad}_{W_2}\mathcal{F}(\mu) \circ T_{b,\mu}) + T_{b,\nu}\right)_{\#b}$$

$$= \left(id - \frac{1}{L}(\operatorname{Grad}_{W_2}\mathcal{F}(\nu) - \operatorname{Grad}_{W_2}\mathcal{F}(\mu) \circ T_{\nu,\mu})\right)_{\#\nu} \in \mathcal{P}_{2,ac}(\mathbb{R}^d).$$

The above equality holds since $T_{\nu,\mu} \circ T_{b,\nu} = T_{b,\mu}$ when $\mu,\nu$ are the iterates from (5.2) (this is because $T_{\cdot,\cdot}$ is *not* the optimal transport map). Precisely, for $j > i$,

$$T_{b,\mu_j} = \underbrace{(id - \operatorname{Grad}_{W_2}\mathcal{F}(\mu_{j-1})) \circ (id - \operatorname{Grad}_{W_2}\mathcal{F}(\mu_{j-2})) \circ \cdots \circ (id - \operatorname{Grad}_{W_2}\mathcal{F}(\mu_i))}_{=T_{\mu_i,\mu_j}} \circ T_{b,\mu_i}.$$

Since $\operatorname{Grad}_{W_2}\mathcal{F}(\mu), \operatorname{Grad}_{W_2}\mathcal{F}(\mu) \circ T_{\nu,\mu} \in \mathcal{L}^2(\nu)$, the second moment condition is satisfied, and the regularity condition we imposed ensures the absolute continuity.

Remaining part is whether the gradient iterates are well-defined in $\mathcal{P}_{2,ac}(\mathbb{R}^d)$, which is not guaranteed anymore as the space is not geodesically complete. Again, the second moment condition is direct; since $\mathrm{Grad}_{W_2}\,\mathcal{F}(\mu_n) \in L^2(\mu_n)$, $(id - \frac{\eta_n}{L}\,\mathrm{Grad}_{W_2}\,\mathcal{F}(\mu_n))_{\#\mu_n}$ also has the second moment. In addition, we imposed $d(id - \frac{\eta_n}{L}\,\mathrm{Grad}_{W_2}\,\mathcal{F}(\mu_n)_{\#\mu_n} \ll dm$ where $dm$ is Lebesugue measure, so it certifies that the gradient iterates are staying in $\mathcal{P}_{2,ac}(\mathbb{R}^d)$. However, there is still one more thing to clarify: Definition A.28 is well defined only when all geodesic interpolations exist, which is stronger than merely requiring invertibility at the endpoints. Such a condition guarantees the well definedness of geodesic interpolations between two points. Nevertheless, note that our proofs of Theorems 4.1 and 4.2 rely only on inequalities evaluated at the iterates themselves; no interpolating points are involved (e.g., the proofs do not invoke time integrals). Hence, we do not need the the all geodesic interpolations between $\mu_n$ and $(id - \frac{\eta_n}{L}\,\mathrm{Grad}\,\mathcal{F}(\mu_n))_{\#\mu_n}$ to exist. Under this perspective, we can conclude that as long as the gradient iterates stay in $\mathcal{P}_{2,ac}(\mathbb{R}^d)$, all our proof ingredients remain valid.

In sum, as long as the imposed regularity conditions hold, all our proof ingredients remain valid even without the geodesic completeness. This completes the proof. ∎

**Remark B.7** (Proof for Bures-Wasserstein space). *The proof of Corollary 5.1 holds the same if we replace $N$ to be $BW(\mathbb{R}^d)$, as $BW(\mathbb{R}^d)$ is a totally geodesic submanifold of $\mathcal{P}_{2,ac}(\mathbb{R}^d)$. This justifies our choice of $N$ in Section 5.1.*

Now, we prove the results in Remark 5.3.

*Proofs on the statements in Remark 5.3.* First, we show if $d\mu \ll dm$, $I - s\nabla^2 h$ being invertible implies $d(id - s\nabla h)_{\#\mu} \ll dm$ for any fixed $s > 0$, if $h \in C^{1,1}_{loc}(\mathbb{R}^d)$. Since $h \in C^{1,1}_{loc}(\mathbb{R}^d)$, the map $T(x) = x - s\nabla h(x)$ is locally Lipschitz and differentiable a.e.

Write $B_R := \overline{B}(0, R)$, and for any nonnegative measurable function $g$ consider $g_k(y) = \min\{g(y), k\}\mathbb{1}_{\{|y|\leq k\}}$. Then, by the change of variable formula and area formula,

$$\int_{B_R} g_k(y) d(T_{\#\mu})(y) = \int_{T^{-1}(B_R)} g_k \circ T(x) d\mu(x) = \int_{T^{-1}(B_R)} g_k \circ T(x)\mu(x)dx$$
$$= \int_{B_R} g_k(y) \sum_{x \in B_R \cap T^{-1}(y)} \frac{\mu(x)}{|\det(I - s\nabla^2 h(x))|}dy.$$

We used the invertibility for the division by the Jacobian. Next, using monotone convergence theorem with $R \to \infty$, one gets

$$\int_{\mathbb{R}^d} g_k(y) d(T_{\#\mu})(y) = \int_{\mathbb{R}^d} g_k(y) \sum_{x \in T^{-1}(y)} \frac{\mu(x)}{|\det(I - s\nabla^2 h(x))|}dy.$$

Apply MCT one more time with $k \to \infty$ to get

$$\int_{\mathbb{R}^d} g(y) d(T_{\#\mu})(y) = \int_{\mathbb{R}^d} g(y) \sum_{x \in T^{-1}(y)} \frac{\mu(x)}{|\det(I - s\nabla^2 h(x))|}dy.$$

Now, for any measurable function $g$, one can use the standard method in measure theory (spliiting $g = g^+ - g^-$) to get

$$\int_{\mathbb{R}^d} g(y) d(T_{\#\mu})(y) = \int_{\mathbb{R}^d} g(y) \sum_{x \in T^{-1}(y)} \frac{\mu(x)}{|\det(I - s\nabla^2 h(x))|}dy$$

for any measurable function $g$. This shows that $(id - s\nabla h)_{\#\mu}$ admits the density $\sum_{x \in T^{-1}(y)} \frac{\mu(x)}{|\det(I-s\nabla^2 h(x))|}$.

For $(id - \frac{1}{L}(\nabla h - \nabla h \circ T_{\nu,\mu})_{\#\nu} \ll dm$, the same argument as in the above holds as long as $T_{\nu,\mu} \in C^{1,1}_{loc}(\mathbb{R}^d)$. ∎

In many applications, $\mathrm{Grad}_{W_2}\,\mathcal{F}(\mu) \in C^{1,1}_{loc}(\mathbb{R}^d)$ is weak condition. For example, our potential energy functional application satisfies this as long as the potential function is convex and $L$-smooth,

which is typical case. In addition, the negative entropy functional $\mathcal{H}(\mu) = \int \mu \log \mu$ also satisfies this condition as long as the log density is $C^{1,1}_{loc}(\mathbb{R}^d)$. The regularity condition on the optimal transport map $T_{\nu,\mu} \in C^{1,1}_{loc}(\mathbb{R}^d)$ have been studied intensively in the Monge-Ampere Equation literature, and depends on the relationship between densities of $\mu$ and $\nu$, but it holds in many practical applications. For instance, in our Gaussian application, since the transport map is linear, it satisfies the desired regularity.

Next, we present a complete proof of Proposition 5.4.

*Proof of Proposition 5.4.* Since the argument is identical for both the 2-Wasserstein and Bures–Wasserstein geometries, we only present the proof in the 2-Wasserstein case.

Fix an aribtrary base $b \in \mathcal{P}_{2,ac}(\mathbb{R}^d)$. Let $T_{b,\mu}$ and $T_{b,\nu}$ be any transport maps from $b$ to $\mu$ and $\nu$ respectively. From the condition that $V$ is convex and $L$-smooth on $\mathbb{R}^d$, we have for any $z \sim b$,

$$V(T_{b,\nu}(z)) - V(T_{b,\mu}(z)) - \langle \nabla V(T_{b,\mu}(z)), T_{b,\nu}(z) - T_{b,\mu}(z) \rangle - \frac{1}{2L} \left\| \nabla V(T_{b,\nu}(z)) - \nabla V(T_{b,\mu}(z)) \right\|^2 \geq 0$$

which is the standard inequality for convex $L$-smooth function on $\mathbb{R}^d$. Take an expectation over $z \sim b$ on the above inequality. The result follows from the fact $\mathrm{Grad}_{W_2} \mathcal{V}(\mu)(\cdot) = \nabla V(\cdot)$, which is from [San14, Remark 7.13] and Definition A.34. Since the result holds regardless of the choice of base $b$, it holds with any base $b$.

To substitute $W_2^2(\mu_0, \mu_*)$ for $\|T_{b,\mu_0} - T_{b,\mu_*}\|_b^2$, we notice the above result holds for any $b$. In addition, notice the gradient update $\mu_n$ itself does not depend on $b$. Therefore, one can rewrite the result of Corollary 5.1 as follows:

$$\mathcal{V}(\mu_n) - \mathcal{V}(\mu_*) \leq r_k L \inf_{b \in \mathcal{P}_{2,ac}(\mathbb{R}^d)} \|T_{b,\mu_0} - T_{b,\mu_*}\|_b^2 = r_k L \inf_{b \in \mathcal{P}_{2,ac}(\mathbb{R}^d)} \|T_{b,\mu_*} \circ T_{\mu_0,b} - id\|_{\mu_0}^2 .$$

By the optimality of the transport plan, one has $\inf_{b \in \mathcal{P}_{2,ac}(\mathbb{R}^d)} \|T_{b,\mu_*} \circ T_{\mu_0,b} - id\|_{\mu_0}^2 = W_2^2(\mu_0, \mu_*)$, which completes the proof.

Lastly, for strongly convex result, we claim $\mathcal{V}$ is geodesically $\alpha$-strongly convex if $V$ is $\alpha$-strongly convex. Under the claim, the result is direct from Corollary 5.1. To show the claim, for any $\mu, \nu \in \mathcal{P}_{2,ac}(\mathbb{R}^d)$, write $T_{\mu,\nu}$ the optimal transport map from $\mu$ to $\nu$. From the strong convexity of $V$, for any $x \sim \mu$ we get

$$V(T_{\mu,\nu}(x)) \geq V(x) + \langle \nabla V(x), T_{\mu,\nu}(x) - x \rangle + \frac{\alpha}{2} \|T_{\mu,\nu}(x) - x\|^2 .$$

Take the expectation over $x \sim \mu$ on the above inequality. Using the facts that $T_{\mu,\nu}$ is the optimal transport map and $\mathrm{Grad}_{W_2} \mathcal{V}(\mu)(\cdot) = \nabla V(\cdot)$ lead to the claim.

∎

**Remark B.8.** *Note for the above proof we did not use the* optimal *transport map, and considered arbitrary transport map. Hence, our algorithm is readily applicable to this setting.*

## C  Generalized geodesic convexity and smoothness

The notion of generalized geodesic convexity was originally introduced in optimal transport and has found various usages in Wasserstein geometry, including the theoretical analysis of the proximal operator in the 2-Wasserstein space [AGS08, Lemma 9.2.7], [SKL20, DBCS23], and its connection to $\Gamma$-convergence [AGS08, Lemma 9.2.9]. To the best of our knowledge, this notion has not yet been explored in the Riemannian geometry literature. We therefore expect that introducing it in this context could provide new tools for analyzing proximal operators and $\Gamma$-convergence on Riemannian manifolds, as it has in the 2-Wasserstein setting-areas that, to date, remain underdeveloped.

In this appendix, we provide some examples of generalized geodesically convex functionals for readers who are not familiar with the concept.

First, recall the notion of generalized geodesic convexity, which is $\mathcal{VT}$-geodesic convexity with $\mathcal{VT} = \Gamma$. Generalized geodesic smoothness can be understood in analogous manner.

**Definition C.1** (Generalized geodesic convexity). *A differentiable function $f : N \to \mathbb{R}$ is called generalized geodesically $\alpha$-strongly convex with base $b \in M$ if for all $x, y \in N$*

$$f(y) \geq f(x) + \left\langle \Gamma_x^b \operatorname{Grad} f(x), \log_b y - \log_b x \right\rangle_b + \frac{\alpha}{2} \left\| \log_b y - \log_b x \right\|_b^2.$$

*If $\alpha = 0$, we say $f$ is generalized geodesically convex with base $b$. If $f$ is generalized geodesically $\alpha$-strongly convex for all $b \in M$, then $f$ is called generalized geodesically $\alpha$-strongly convex.*

We start with the trivial example: Euclidean space.

**Example C.2.** *A differentiable, $\alpha$-strongly convex function $f : \mathbb{R}^d \to \mathbb{R}$ is generalized geodesically $\alpha$-strongly convex.*

*Proof.* In Euclidean space, $\exp_x(v) = x + v$ and $\log_x y = y - x$. Since $f$ is differentiable and $\alpha$-strongly convex, for all $x, y, b \in \mathbb{R}^d$

$$\begin{aligned} f(y) &\geq f(x) + \langle \nabla f(x), y - x \rangle + \frac{\alpha}{2} \left\| y - x \right\|^2 \\ &= f(x) + \langle \nabla f(x), (y - b) - (x - b) \rangle + \frac{\alpha}{2} \left\| (y - b) - (x - b) \right\|^2. \end{aligned}$$

∎

Now, we move to nontrivial examples: non-Euclidean manifolds. As mentioned in the main body, this concept has already been widely discussed in the Wasserstein space. Therefore, there are some known examples in 2-Wasserstein space. We first introduce some generalized geodesically convex functionals in Wasserstein space: potential energy functional and internal energy functional.

**Example C.3** (Potential energy). *Consider a function $V : \mathbb{R}^d \to \mathbb{R}$. A functional $\mathcal{V}(\mu) := \mathbb{E}_{X \sim \mu}[V(X)]$ is called a potential functional. If $V$ is $\alpha$-strongly convex (L-smooth) in $\mathbb{R}^d$, then $\mathcal{V}$ geodesically $\alpha$-strongly convex (resp. L-smooth).*

This is duplicate of Proposition 5.4.

**Example C.4** (Internal energy). *Let $F : [0, \infty) \to (-\infty, \infty]$ be a proper, lower semi-continuous convex function such that*

$$F(0) = 0, \quad \liminf_{s \downarrow 0} \frac{F(s)}{s^\alpha} > -\infty \text{ for some } \alpha > \frac{d}{d+2}.$$

*Consider a functional $\mathcal{H}_F : \mathcal{P}_{2,ac}(\mathbb{R}^d) \to \mathbb{R}$ defined by*

$$\mathcal{H}_F(\mu) := \int_{\mathbb{R}^d} F(\mu(x)) dx.$$

*If the map $s \mapsto s^d F(s^{-d})$ is convex and non-increasing in $(0, \infty)$, then the functional $\mathcal{H}_F$ is generalized geodesically convex.*

We refer to [AGS08, Proposition 9.3.9] for the proof.

**Remark C.5.** *Some widely used choice of $F$ satisfying the conditions are as follows:*

1. *$F(s) = s \log s$. This choice leads to $\mathcal{H}_F$ being the differential entropy functional.*

2. *For any $q > 1$, $F(s) = s^q$.*

3. *For $m \geq 1 - 1/d$, $F(s) = \frac{1}{m-1} s^m$.*

Now, we present examples on Riemannian manifolds. We begin by providing sufficient conditions for generalized geodesic convexity, which turns out to be useful in verifying the generalized geodesic convexity for a given functional.

**Lemma C.6** (Criteria for generalized geodesic convexity). *Fix $b \in N$. For any $x, y \in N$, let $\gamma(t)$ be any curve such that $\gamma(0) = x$, $\gamma(1) = y$, and $\dot{\gamma}(0) = \Gamma_b^x(\log_b y - \log_b x)$. If a differentiable function $f : N \to \mathbb{R}$ satisfies either one of the following conditions, then $f$ is generalized geodesically convex with base $b \in N$.*

1. *Zeroth-order criterion: $(1 - t)f(x) + tf(y) \geq (f \circ \gamma)(t)$ for all $t \in [0, 1]$.*

2. *Second-order criterion: $\frac{d^2}{dt^2}(f \circ \gamma)(t) \geq 0$ for all $t \in (0, 1)$.*

*Proof.* **1. Zeroth-order criterion:** Let $\phi(t) := (1 - t)f(x) + tf(y) - (f \circ \gamma)(t)$. Then, $\phi(t) \geq 0$ and $\phi(0) = 0$ is the global minimizer. Since $f$ is differentiable,

$$0 \leq \phi'(0^+) = \lim_{t \to 0+} \frac{\phi(t)}{t} = f(y) - f(x) - \frac{d}{dt}\Big|_{t=0}(f \circ \gamma)(t)$$
$$= f(y) - f(x) - \left\langle \text{Grad}\, f(x), \Gamma_b^x(\log_y b - \log_b x) \right\rangle_x$$
$$= f(y) - f(x) - \left\langle \Gamma_x^b \, \text{Grad}\, f(x), \log_b y - \log_b x \right\rangle_b.$$

**2. Second-order criterion:** By Taylor's theorem,

$$f(y) = f(x) + \frac{d}{dt}\Big|_{t=0}(f \circ \gamma)(t) + \int_0^1 (1 - t)\frac{d^2}{dt^2}(f \circ \gamma)(t)dt$$
$$\geq f(x) + \left\langle \text{Grad}\, f(x), \Gamma_b^x(\log_b y - \log_b x) \right\rangle = f(x) + \left\langle \Gamma_x^b \, \text{Grad}\, f(x), \log_b y - \log_b x \right\rangle.$$

$\blacksquare$

**Remark C.7** (Existence of $\gamma$). *It is natural to ask whether such curve $\gamma(t)$ exists. In fact, as long as the exponential map is defined for sufficiently large neighborhood of $x$, there always exists a curve satisfying the conditions. For example, in a complete manifold, such curve always exists. Let $v(t) := t\Gamma_b^x(\log_b y - \log_b x) + t^2(\log_x y - \Gamma_b^x(\log_b y - \log_b x))$, and define $\gamma(t) = \exp_x(v(t))$. Observe $\gamma(0) = x$ and $\gamma(1) = y$. Furthermore, since the differential of the exponential map is the identity at the origin, by the chain rule*

$$\dot{\gamma}(0) = d\exp_x(v(0))[v'(0)] = \Gamma_b^x(\log_b y - \log_b x).$$

*In certain Riemannian manifolds with a particularly well-behaving exponential map, simpler curves can be used. For instance, in the 2-Wasserstein space, a more natural choice of curve is available. Fix a base $\pi \in \mathcal{P}_{2,ac}(\mathbb{R}^d)$. For any $\mu, \nu \in \mathcal{P}_{2,ac}(\mathbb{R}^d)$, let $\gamma(t) := \exp_\pi((1 - t)\log_\pi \mu + t\log_\pi \nu) = ((1 - t)T_{\pi,\mu} + tT_{\pi,\nu})_{\#}\pi$ be a curve. Then, $\gamma(0) = \mu, \gamma(1) = \nu$, and the velocity vector field corresponding to $\gamma(t)$ is $v_t = (T_{\pi,\nu} - T_{\pi,\mu}) \circ T_{\gamma(t),\pi}$ [DBCS23, Appendix B.2].*

As a specific example, we consider the entropy functional on $SPD(d)$ space. This example will show how one can verify the generalized geodesic convexity using Lemma C.6.

**Example C.8** (Entropy of Gaussian). *Consider a functional $\mathcal{H} : SPD(d) \to \mathbb{R}$ defined by $\mathcal{H}(A) = -\frac{1}{2}\log \det A$. This functional is in fact the entropy functional of the multivariate Gaussian distribution $N(0, A)$ (up to an affine transformation). There are two natural Riemannian metrics in $SPD(d)$ space [FAP+05, PFA05, BH06, HMJG21, Ngu22, TP22, KPB25].*

1. *Affine invariant metric: $d_{AI}(A, B) := \left\| \log A^{-1/2}BA^{-1/2} \right\|_F$, and $\langle S, R \rangle_A = \text{tr}(A^{-1}SA^{-1}R)$ for $S, R \in Sym(d)$. This metric induces non-positively curved geometry on SPD(d).*

2. *Bures-Wasserstein metric: $d_{BW}^2(A, B) := \text{tr}(A) + \text{tr}(B) - 2\text{tr}(A^{1/2}BA^{1/2})^{1/2}$, and $\langle S, R \rangle_A = \text{tr}(SAR)$ for $S, R \in Sym(d)$. This metric induces non-negatively curved geometry on SPD(d).*

*Both geometries originate from the geometry of zero-mean Gaussian distributions. The metric $d_{AI}$ arises from the Fisher information metric associated with zero-mean Gaussians [Nie23], while the metric $d_{BW}$ corresponds to the Wasserstein geometry of zero-mean Gaussians, as described in Appendix A.2.1. Under both geometries, $\mathcal{H}(A)$ is generalized geodesically convex.*

Note that $d_{BW}$ corresponds to the 2-Wasserstein distance between Gaussians, so the result for $d_{BW}$ is a special case of Example C.4. Nonetheless, we present the proof entirely in the language of Riemannian geometry to demonstrate that the notion of generalized geodesic convexity remains valid purely within the Riemannian framework.

*Proof of Example C.8.* In both cases, we apply the second-order criterion from Lemma C.6. The general strategy is to construct a curve that satisfies the required conditions with respect to a fixed starting point, endpoint, and base point. The specific choice of curve should reflect the underlying geometry. Once the curve is chosen, we compute the time derivative of the functional along the curve; this can be carried out entirely using matrix calculus, without explicitly invoking the Riemannian structure.

We will use $N$ to denote the arbitrary base point, and $M_0, M_1$ to denote the starting point and the endpoint of the curve.

**1. Affine invariant metric**: We first construct a curve satisfying the desired property. For simplicity, write $X_i := N^{-1/2} M_i N^{-1/2}$. Define $c(t) := At + Bt^2$ where

$$A = \log X_1 - \log X_0,$$
$$B = \log(X_0^{-1/2} X_1 X_0^{-1/2}) - (\log X_1 - \log X_0).$$

We now consider a curve on $\mathrm{SPD}(d)$ defined by

$$M(t) := N^{1/2} X_0^{1/2} \exp(c(t)) X_0^{1/2} N^{1/2}.$$

Here, the $\exp$ is usual matrix exponential, not the exponential map. We claim this is the desired curve in Lemma C.6.[1] First, $M(0) = N^{1/2} X_0 N^{1/2} = M_0$, $M(1) = N^{1/2} X_1 N^{1/2} = M_1$. Now, we check $M'(0)$. Note

$$M'(0) = N^{1/2} X_0^{1/2} \exp(c(0)) c'(0) X_0^{1/2} N^{1/2}$$
$$= N^{1/2} X_0^{1/2} A X_0^{1/2} N^{1/2}.$$

Now, since $M_0 = N^{1/2} X_0 N^{1/2}$, $M_0 N^{-1} = N^{1/2} X_0 N^{-1/2}$. Hence, $(M_0 N^{-1})^{1/2} = N^{1/2} X_0^{1/2} N^{-1/2}$. This leads to

$$M'(0) = N^{1/2} X_0^{1/2} N^{-1/2} [N^{1/2} A N^{1/2}] N^{-1/2} X_0^{1/2} N^{1/2}$$
$$= (M_0 N^{-1})^{1/2} N^{1/2} A N^{1/2} ((M_0 N^{-1})^{1/2})^T.$$

This exactly coincides to $\Gamma_N^{M_0}(\log_N M_1 - \log_N M_0)$ on $(\mathrm{SPD}(d), d_{AI})$.[2]

Now, since we obtained the desired curve, we compute $\frac{d^2}{dt^2} \mathcal{H}(M_t)$. First, observe

$$\mathcal{H}(M_t) = -\frac{1}{2} \left( \log \det \exp(c(t)) + \log \det N + \log \det X_0 \right)$$
$$= -\frac{1}{2} \left( \log \exp \mathrm{tr}(c(t)) + \log \det N + \log \det X_0 \right) = -\frac{1}{2} \left( \mathrm{tr}(c(t)) + \log \det N + \log \det X_0 \right)$$

where we used the well-known matrix identity $\det \exp(Y) = \exp \mathrm{tr}(Y)$.

Thus,

$$\frac{d^2}{dt^2} \mathcal{H}(M_t) = -\frac{1}{2} \mathrm{tr}(c''(t)) = -\mathrm{tr}(B)$$
$$= -\mathrm{tr}(\log(X_0^{-1/2} X_1 X_0^{-1/2})) + \mathrm{tr}(\log X_1) - \mathrm{tr}(\log X_0)$$
$$= -\log \det X_0 - \log \det X_1 + \log \det X_1 - \log \det X_0 = 0$$

---

[1] In fact, this curve is constructed as in Remark C.7.

[2] For the formula of the parallel transport and Riemannian logarithmic map on $(SPD, d_{AI})$, see [Ngu22, Supplement 1.1].

from the well-known matrix identity $\operatorname{tr}\log(Y) = \log\det(Y)$. Hence, by the second order criterion of Lemma C.6, $\mathcal{H}$ is generalized geodesically convex with base $N$.

Since the above result holds for arbitrary choice of $N \in \mathrm{SPD}(d)$, we get the generalized geodesic convexity of $\mathcal{H}$.[3]

**2. Bures-Wasserstein metric**: We again start with constructing a curve satisfying the desired properties. As noted in Remark A.40, in this setting we must match the *tangent vector corresponding to* $M'(0)$ with $\Gamma_N^{M_0}(\log_N M_1 - \log_N M_0)$, rather than matching $M'(0)$ directly. We consider $\nu = N(0, N), \mu_0 = N(0, M_0)$, and $\mu_1 = N(0, M_1)$. From Appendix A.2.1, the optimal transport map between 0-mean Gaussians is a linear map. Therefore, for any $\pi_0, \pi_1$, we denote $B_{L_0,L_1}$ to be the matrix corresponding to the optimal transport map between $\pi_0 = N(0, L_0), \pi_1 = N(0, L_1)$, *i.e.*, $T_{\pi_0,\pi_1}(x) = B_{L_0,L_1}x$. Now, consider a curve on $\mathrm{SPD}(d)$ defined by

$$M(t) := ((1-t)I + tB_{N,M_1}B_{M_0,N}) M_0 ((1-t)I + tB_{N,M_1}B_{M_0,N})^T \,^{4}.$$

Then, $M(0) = M_0$ trivially and $M(1) = M_1$; for any $X \sim N(0, M_0)$, on the one hand $B_{N,M_1}B_{M_0,N}X = T_{\nu,\mu_1} \circ T_{\mu_0,\nu}(X) \sim N(0, M_1)$, and on the other hand $B_{N,M_1}B_{M_0,N}X \sim N(0, (B_{N,M_1}B_{M_0,N})M_0(B_{N,M_1}B_{M_0,N})^T)$, meaning $(B_{N,M_1}B_{M_0,N})M_0(B_{N,M_1}B_{M_0,N})^T = M_1$. In addition, since $M'(0) = B_{N,M_1}B_{M_0,N}M_0 + M_0B_{N,M_1}B_{M_0,N}$, from the identification in Remark A.40 the tangent vector corresponding to $M'(0)$ is $V_0 = B_{N,M_1}B_{M_0,N} - I = \Gamma_N^{M_0}(B_{N,M_1} - B_{N,M_0})$. Therefore, the curve $M(t)$ satisfies the conditions in Lemma C.6.

Now, we compute $\frac{d^2}{dt^2}\mathcal{H}(M_t)$. First, since $M_t = A_t M_0 A_t^T$, $\mathcal{H}(M_t) = -\log\det(A_t) - \frac{1}{2}\log\det M_0$. Then, for all $t \in (0,1)$,

$$\begin{aligned}
\frac{d^2}{dt^2}\mathcal{H}(M_t) &= -\frac{d^2}{dt^2}\log\det(A_t) = -\frac{d}{dt}\operatorname{tr}\left(A_t^{-1}\dot{A}_t\right) = -\frac{d}{dt}\operatorname{tr}\left(A_t^{-1}(B_{N,M_1}B_{M_0,N} - I)\right) \\
&= -\operatorname{tr}\left(\frac{d}{dt}A_t^{-1}(B_{N,M_1}B_{M_0,N} - I)\right) = \operatorname{tr}\left(A_t^{-1}\dot{A}_t A_t^{-1}(B_{N,M_1}B_{M_0,N} - I)\right) \\
&= \operatorname{tr}\left(A_t^{-1}(B_{N,M_1}B_{M_0,N} - I)A_t^{-1}(B_{N,M_1}B_{M_0,N} - I)\right) \\
&\overset{(i)}{=} \operatorname{tr}\left(\left[A_t^{-1/2}(B_{N,M_1}B_{M_0,N} - I)A_t^{-1/2}\right]^2\right) \geq 0
\end{aligned}$$

which is the desired inequality. For (i), we claim that $A_t^{-1/2}$ is well-defined as the principal square root for all $t \in (0,1)$. This follows from the fact that both $B_{N,M_1}, B_{M_0,N}$ are optimal transport maps and thus, by Brenier's Theorem A.24, they are non-negative definite. Consequently, the product $B_{N,M_1}B_{M_0N}$ also has non-negative eigenvalues. Since $A_t$ is a convex combination of the identity matrix $I$ and a matrix with non-negative eigenvalues, it follows that all eigenvalues of $A_t$ are strictly positive on $t \in (0,1)$. Hence, all eigenvalues of $A_t^{-1}$ are positive for $t \in (0,1)$, and then $A_t^{-1/2}$ is well-defined as the principal square root.

Again, since the inequality holds for arbitrary base $N$, we obtain the generalized geodesic convexity of $\mathcal{H}$. ∎

Lastly, we show the generalized geodesic smoothness with base $b$ is not strictly stronger than geodesic smoothness. In particular, as mentioned in Section 3, we show the function $f(x) = \frac{1}{2}d^2(x, p)$ for fixed $p$ on Hadamard manifold $M$ is generalized geodesically smooth with base $p$, while it is not geodesically smooth [CK25].

Before we show the result, we need the following lemma, which shows how logarithmic map changes under the parallel transport.

**Lemma C.9.** *For all $x, y \in N$, let $\Gamma_x^y$ be a parallel transport from $x$ to $y$ induced from the geodesic connecting $x$ and $y$. Then,*

$$\Gamma_x^y \log_x y = -\log_y x.$$

---

[3]More precisely, since the second derivative is zero, functional $\mathcal{H}$ is generalized geodesically linear.

[4]While $B_{N,M_1}B_{M_0,N} - I$ may not be symmetric, the formula on the right hand side is still well-defined. Consequently, there is no harm in defining the curve via this formula.

This result is analogous result of $y - x = -(x - y)$ in Euclidean case.

*Proof.* Let $\gamma : [0, 1] \to M$ be a geodesic curve such that $\gamma(0) = x$ and $\gamma(1) = y$. Then, by definition of logarithmic map, one gets $\gamma'(0) = \log_x y$.

Now, consider the reversed geodesic $\sigma(t) := \gamma(1 - t)$. Then, $\sigma'(0) = -\gamma'(1) = \log_y x$. By the property of the geodesic and the parallel transport,

$$\Gamma_x^y \log_x y = \Gamma_x^y \gamma'(0) = \gamma'(1) = -\sigma'(0) = -\log_y x.$$

∎

Now we are ready to prove the following example.

**Example C.10** (Generalized geodesic smoothness with base is not restrictive)**.** *The function $f(x) = \frac{1}{2} d^2(x, p)$ on Hadamard manifold $M$ is generalized geodesically $1$-smooth with base $p$.*

*Proof.* On Hadamard manifold, the exponential map is global diffeomorphism. Hence, we have $f(x) = \frac{1}{2} \left\| \log_p x \right\|^2$. In addition, the standard fact on Riemannian manifold is that $\text{Grad} f(x) = -\log_x p$ [AOBL20, Section 4]. Now, we show the inequality (3.3) with $L = 1$. In fact, in this case it holds with equality. This can be verified by the below calculation:

$$
\begin{aligned}
&f(y) - f(x) - \left\langle \Gamma_x^p \text{Grad} f(x), \log_p y - \log_p x \right\rangle \\
&= \frac{1}{2} \left\| \log_p y \right\|^2 - \frac{1}{2} \left\| \log_p x \right\|^2 + \left\langle \Gamma_x^p \log_x p, \log_p y - \log_p x \right\rangle \\
&\overset{\text{(i)}}{=} \frac{1}{2} \left\| \log_p y \right\|^2 - \frac{1}{2} \left\| \log_p x \right\|^2 - \left\langle \log_p x, \log_p y - \log_p x \right\rangle \\
&= \frac{1}{2} \left\| \log_p y \right\|^2 + \frac{1}{2} \left\| \log_p x \right\|^2 - \left\langle \log_p x, \log_p y \right\rangle \\
&= \frac{1}{2} \left\| \log_p y - \log_p x \right\|^2.
\end{aligned}
$$

∎

It is known that the above $f(x)$ is not geodesically smooth [CK25]. Thus, Example C.10 shows that generalized geodesic smoothness with single base is not strictly stronger than standard geodesic smoothness.

# D    Implementation detail and additional experiments

This section includes implementation detail and more experiments of our algorithm under different settings. We conduct additional experiments on the problems in Section 5, to show the robustness of our algorithm. In particular, in this appendix we elaborate the following points that were briefly mentioned in the main body.

1. Because the silver stepsize schedule sometimes uses very large stepsizes, one might ask whether simply increasing RGD's constant stepsize could match its performance. We show this is not the case: using a constant stepsize above the critical threshold $2/L$ causes RGD to diverge, while silver stepsize shows the improved performance.

2. We conducted experiments using multiple random seeds and demonstrate that our algorithm's performances are statistically significant.

Furthermore, to demonstrate our method's versatility, we include experiments on an additional optimization problem in the Wasserstein space: the mean-field training of a two-layer neural network. This problem showcases the applicability of our algorithm, and of Wasserstein-based optimization more broadly, to neural network training. In addition, we provide additional experiments on SPD matrix space.

### D.1 Implementation detail

All experiments in our paper were conducted on the free version of Google Colab using a T4 GPU. Each task took no more than 5 minutes.

**Wasserstein potential functional optimization** For the potential functional optimization problem in Section 5.1, we used Python packages `numpy, scipy` for the implementation. We generated $m_*$ from the uniform distribution on the unit cube $[0,1]^d$. For $\Sigma_*$, since we conducted experiments with fixed $L = 1$ and $\alpha = 10^{-1}, 10^{-3}, 10^{-7}, 10^{-13}$, we have $\lambda_{\min} = 1/L = 1$ and $\lambda_{\max} = 1/\alpha$. We placed $d$ points evenly on a log-scale over the interval $[1/L, 1/\alpha]$ and used those values as the eigenvalues to construct a diagonal matrix $\Lambda$. Then, we uniformly sampled an orthogonal matrix $P$ from the uniform distribution on the orthogonal group $O(d)$ (using Haar measure), and set $\Sigma_* = P\Lambda P^T$. We used $m_0 = 0$ and $\Sigma_0 = I$ as the initialization for all experiments.

**SPD space optimization** As in Wasserstein potential optimization, for $C$ we placed $d$ points evenly on a log-scale over the interval $[1/\alpha, 1]$ and used those values as the diagonal matrix $\Lambda$. Then, we again uniformly sampled the orthogonal matrix $P$ and set $C = P\Lambda P^T$. We used $\alpha = 10$ and $\alpha = 10^5$ to make the desired condition number.

### D.2 Additional experiments

#### D.2.1 Potential functional optimization

We conduct numerial experiments on two tasks: the same task as in Section 5.1, and logistic regression. To verify that our algorithm remains effective with a general choice of iteration count unless $n$ is close to the spikes (e.g. $n = 2^k$), we set the number of iterations $n = 1500$, which is neither of the form $2^k - 1$ nor close to $2^{10} - 1$ or $2^{11} - 1$. For the inner-iterations in the strongly convex setting for the restarting, we chose $m = 20$ for $\alpha = 10^{-1}$ and $m = 500$ for $\alpha = 10^{-3}$, selecting values near the $2^{k^*} - 1$ in Theorem 4.2 while ensuring divisibility by 1,500. We compared our silver stepsize RGD with constant stepsize RGD using $\eta = 1/L$ (the standard choice), $\eta = 1.99/L$ (just below the theoretical threshold), and $\eta = 2.01/L$ (just above it). The experiment was repeated over 100 random seeds, and we report the mean error curves along with 95% confidence intervals. Here, using different seeds can be understood as solving instances of a stochastic optimization problem. In this regard, comparing the errors across different seeds is a reasonable evaluation.

The results are displayed in Figure 4. Figure 4 provides evidence supporting our claims:

1. The algorithm performs well even when the number of iterations is not of the form $2^k - 1$, as long as it is not close to $2^k$.

2. Our method is not equivalent to simply increasing the constant stepsize in RGD; it consistently outperforms all tested stepsize choices. In particular, the large stepsize RGD, unlike silver stepsize RGD, diverges.

3. The performances of our algorithm are statistically significant.

In addition, we conduct the same experiments when gradient oracles are stochastic. The results are coherent with the previous observations at the cost of extra oscillations, given sufficient number of gradient samples. See Figure 5.

Lastly, we provide the Bures-Wasserstein gradient descent experiments on logistic regression potential. For response $Y_i \in \{0, 1\}$ and predictor $X_i \in \mathbb{R}^d$ $(i = 1, \ldots, k)$, a logistic regression model assumes $Y_i \mid X_i \sim \text{Bernoulli}(\text{logistic}(X_i^T \theta))$ independently across $i$. Assuming an improper prior on $\theta$, the potential $V$ takes the form

$$V(\theta) = \sum_{i=1}^{k} \log(1 + \exp(X_i^T \theta)) - Y_i X_i^T \theta.$$

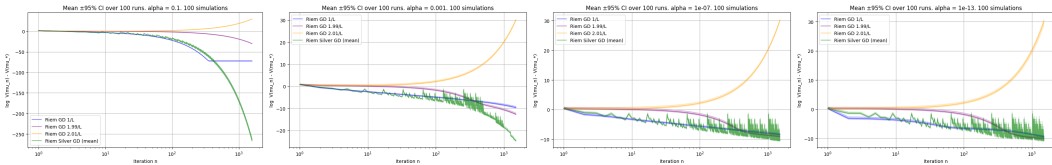

Figure 4: Comparison between silver stepsize method and RGD for potential functional optimization in $BW(\mathbb{R}^d)$ with different convexity parameters. For each task, we conduct 100 simulations with different seeds and plot the mean and 95% confidence interval of the error over the iterates. **Columns**: From left to right, each column corresponds to $\kappa = 10^1, 10^3, 10^7, 10^{13}$.

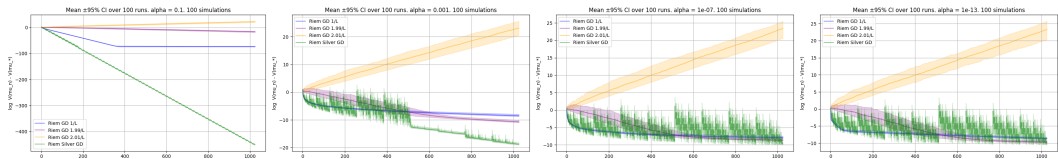

Figure 5: Same experiments as in Figure 4, but with stochastic gradients. We set the number of gradient samples to be 100. **Columns**: From left to right, each column corresponds to $\kappa = 10^1, 10^3, 10^7, 10^{13}$.

It is well-known that $V$ is convex and $\frac{1}{4}\|X\|_{op}^2$-smooth. Thus, proposition 5.4 applies with $L = \frac{1}{4}\|X\|_{op}^2$.

We conduct the experiments on this logistic regression potential minimization problem. Note for this problem, the Wasserstein gradient $(\mathbb{E}_\mu[\nabla V(\theta)], \mathbb{E}_\mu[\nabla^2 V(\theta)])$ does not allow the closed form solution. Hence, we used the Monte Carlo approximation for these quantities. The result is provided in Figure 6. This experiment not only verifies the extendability of our method beyond the quadratic case, but also to the stochastic gradient given the sufficient number of gradient samples.

### D.2.2  Mean-Field Two-Layer Network Training via Wasserstein gradient

Next, we numerically demonstrate the effectiveness of our algorithms for two-layer neural network training. We first introduce the mean-field training formulation for a two-layer neural network, which enables us to view neural network training as a Wasserstein optimization problem, and then present our experimental results. For further details, we refer the interested reader to [CB18, MMN18, Woj20, FRF22].

**Problem formulation**  One way to interpret two-layer neural networks is to view their function space as a space of probability measures. In particular, we adopt the Barron space formulation studied in [Bar93, WE20, Woj20]. In Barron space formulation, a (possibly infinitely wide) two-layer neural network is represented as

$$f_\pi(x) := \mathbb{E}_{(a,w,b)\sim\pi}\left[a\sigma(w^T x + b)\right]$$

where $\sigma$ denoting a fixed activation function (e.g., ReLU). For instance, a $m$-width two-layer neural network corresponds to $f_{\pi_m}$, where $\pi_m = \frac{1}{m}\sum_{i=1}^m \delta_{(a_i,w_i,b_i)}$.

This formulation enables us to view neural network training as an optimization over probability measures. In particular, it becomes the following risk-functional minimization problem:

$$\pi_* := \underset{\pi\in\mathcal{P}_{2,ac}(\mathbb{R}^d)}{\operatorname{argmin}} R(\pi) := \mathbb{E}_{x\sim\mathbb{P}}\left[\ell(f_\pi(x), f^*(x))\right] \tag{D.1}$$

where $f^*$ is the target function, $f_\pi$ is the two-layer neural network, and $\ell$ is a loss function (e.g., squared loss). The neural network $f_{\pi_*}$ is the risk-functional minimizer and thus the desired solution. Since (D.1) is now just the optimization problem on the Wasserstein space, it is possible to consider Wasserstein gradient descent algorithms (5.2) to solve (D.1):

$$\pi_{n+1} = (id - \eta_n \operatorname{Grad}_{W_2} R(\pi_n))_{\#\pi_n} . \tag{D.2}$$

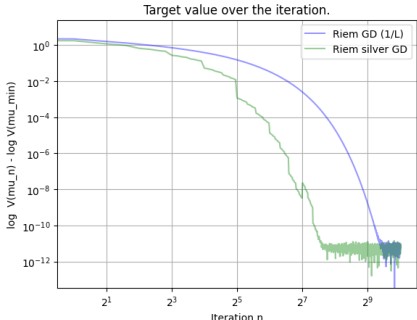

Figure 6: Comparison between silver stepsize method and RGD for potential functional optimization in $BW(\mathbb{R}^d)$ for logistic regression potential. We plotted $\log \mathcal{V}(\mu_n) - \log \mathcal{V}_{\min}$ for the $y$-axis, where $\mathcal{V}_{\min}$ is the minimum value among all experiments. We set $k = 20, d = 10$, and the number of gradient samples to be 100.

In practice, this update operates over the space of functions and is thus not directly implementable. Instead, one typically uses a particle approximation of the probability measure, *i.e.*,

$$\pi_n = \frac{1}{m} \sum_{i=1}^{m} \delta_{(a_i^{(n)}, w_i^{(n)}, b_i^{(n)})},$$

where $m$ is the number of particles chosen by the user [SKL20, WL22]. Under this approximation, the Wasserstein gradient update becomes

$$\pi_{n+1} = (id - \eta_n \operatorname{Grad}_{W_2} R(\pi_n))_{\#\pi_n}$$
$$= \frac{1}{m} \sum_{i=1}^{m} \delta_{(a_i^{(n)}, w_i^{(n)}, b_i^{(n)}) - \eta_n \operatorname{Grad}_{W_2} R(\pi_n)(a_i^{(n)}, w_i^{(n)}, b_i^{(n)})}.$$

Using Definition A.34, it is known from [Woj20] that

$$\operatorname{Grad}_{W_2} R(\pi)(a, w, b) = \mathbb{E}_{x \sim \mathbb{P}} \left[ \nabla_{(a,w,b)} \ell(f_\pi(x), f^*(x)) \right].$$

Therefore, the particle approximation of the Wasserstein gradient update for a two-layer neural network takes the form

$$(a_i^{(n+1)}, w_i^{(n+1)}, b_i^{(n+1)}) = (a_i^{(n)}, w_i^{(n)}, b_i^{(n)}) - \eta_n \mathbb{E}_{x \sim \mathbb{P}} \left[ \nabla_{(a_i^{(n)}, w_i^{(n)}, b_i^{(n)})} \ell(f_{\pi_n}(x), f^*(x)) \right]$$
$$\text{(D.3)}$$

for $i = 1, \ldots, m$. Observe (D.3) exactly coincides with the standard gradient descent update of the parameters.

In conclusion, the silver stepsize (and, respectively, constant stepsize) parameter updates in two-layer neural networks (D.3) can be interpreted as the particle approximation of silver stepsize (*resp.* constant stepsize) Wasserstein gradient descent (D.2) applied to the risk minimization problem (D.1). Hence, we consider applying silver stepsize WGD (5.2) on this problem.

**Numerical experiments** To evaluate the effectiveness of the silver stepsize for this task, we conduct experiments on learning a target function using a two-layer neural network with ReLU activation. Specifically, we consider the simple task of learning a univariate function $f^* : [-1, 1] \to \mathbb{R}$. We consider two target functions:

1. $f^*(x) = \frac{1}{30} \sum_{i=1}^{30} a_i^* \sigma(w_i^* x + b_i^*)$, *i.e.*, a 30-width two-layer neural network with fixed parameters $a_i^*, w_i^*, b_i^*$. Here, $\sigma$ is the ReLU activation.

2. $f^*(x) = \sin(2\pi x)$.

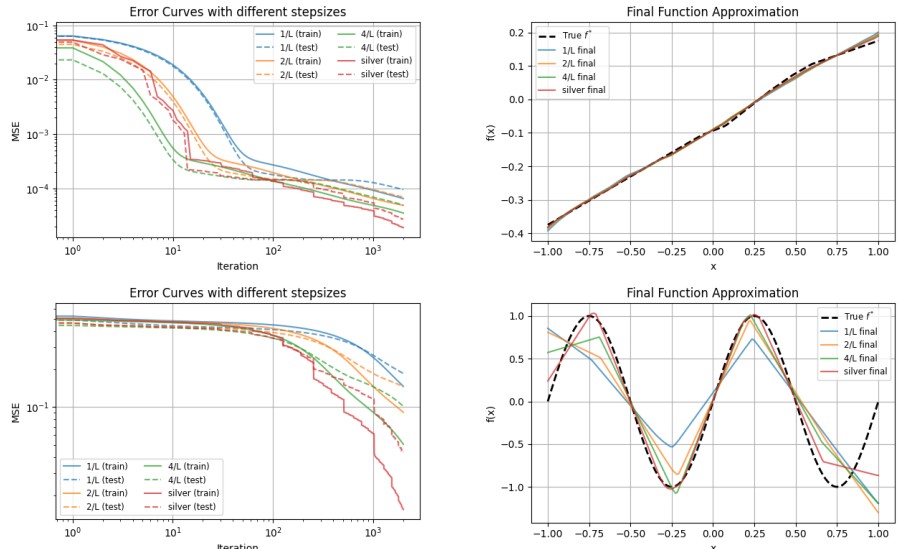

Figure 7: Mean-field training (D.3) of two-layer neural networks. **Rows**: The first row is the results from $f^*(x) = \frac{1}{30}\sum_{i=1}^{30} a_i^* \sigma(w_i^* x + b^*)$, and the second row is the results from $f^*(x) = \sin(2\pi x)$. **Columns**: The first column is the training and test error curve, and the second column is the function graph of the learned function.

We use $N = 200$ samples, with 70% of the data used for training and the remaining 30% for testing. The model is a two-layer neural network with width $m = 100$, trained using mean squared loss. We set the smoothness parameter to $L = 100$, and the number of training iterations to $n = 2000$.

Figure 7 shows the results of our experiments for solving (D.3) using different stepsize schedules. Consistent with previous findings, the silver stepsize algorithm outperforms constant stepsize RGDs with various stepsizes in solving (D.1). While the figure displays results for a specific random seed, we observed similar trends across multiple seeds.

### D.2.3   Additional experiments on SPD space

As briefly mentioned in the main body, there is a non-trivial family of functions that is favorable to use VTRGD algorithm (4.1), which are the functions of the form $f_\varphi(X) = \varphi(\log \det X)$ for some $\varphi : \mathbb{R} \to \mathbb{R}$. The first favorable fact is that $f_\varphi(X)$ is generalized geodesically convex (*resp. L-smooth*) on $(\mathrm{SPD}(d), d_{AI})$ when $\varphi$ is convex (*resp. L-smooth*).

**Proposition D.1.** *For any convex function $\phi : \mathbb{R} \to \mathbb{R}$, the functional $f_\phi(X) = \phi(\log \det X)$ defined on $\mathrm{SPD}(d)$ is generalized geodesically convex. Also, for any L-smooth function $\psi : \mathbb{R} \to \mathbb{R}$, the function $f_\psi(X) = \psi(\log \det X)$ is generalized geodesically Ld-smooth.*

*Proof.* We will show for arbitrary base $B \in \mathrm{SPD}(d)$, Definition 3.4 and 3.5 holds with $\mathcal{VT} = \Gamma$.

First, we consider $g(X) = \log \det X$. We first show for any $X, Y, B \in \mathrm{SPD}(d)$

$$g(Y) - G(X) = \left\langle \Gamma_X^B \operatorname{Grad} g(X), \log_B Y - \log_B X \right\rangle. \tag{D.4}$$

To verify this, first observe that $\operatorname{Grad} g(X) = X$. This can be verified by the definition of Riemannian gradient. Riemannain gradient is defined by the operator satisfying for all $H \in \mathrm{Sym}(d)$

$$\operatorname{tr}(X^{-1}\operatorname{Grad} g(X)Y^{-1}H) = \langle \operatorname{Grad} g(X), H \rangle_X = dg_X(H) = \operatorname{tr}(X^{-1}H).$$

For the last inequality we used the well-known formula for the derivative of log-determinant function. Since $X \in \mathrm{SPD}(d)$, this implies $\operatorname{Grad} g(X) = X$.

Next, we show $\Gamma_X^B X = X$. To check this, from the definition of the parallel transport in $(\mathrm{SPD}(d), d_{AI})$ [Ngu22, Supplement 1.1],

$$\Gamma_X^B X = (BX^{-1})^{1/2} X ((BX^{-1})^{1/2})^T = B.$$

Hence,

$$\left\langle \Gamma_X^B X, \log_B Y - \log_B X \right\rangle = \langle B, \log_B Y \rangle - \langle B, \log_B X \rangle.$$

Now, for any matrix $M$, from the definition of the Riemannian metric on $d_{AI}$ and logarithmic map [Ngu22, Supplement 1.1],

$$\langle B, \log_B M \rangle = \mathrm{tr}(B^{-1} \log_B M) = \mathrm{tr}(B^{-1/2} \log(B^{-1/2} M B^{-1/2}) B^{1/2}) = \mathrm{tr}(\log(B^{-1/2} M B^{-1/2}))$$

$$= \log \det(B^{-1/2} M B^{-1/2}) = 2 \log \det B^{-1/2} + \log \det M.$$

Therefore,

$$\left\langle \Gamma_B^C B, \log_C A - \log_C B \right\rangle = \log \det A - \log \det B = g(A) - g(B)$$

which shows (D.4).

Now we are left with the $\phi$ and $\psi$ part. For $\phi$, by the convexity of $\phi$,

$$\phi \circ g(Y) - \phi \circ g(X) \geq \phi'(g(X))(g(Y) - g(X)) = \phi'(g(X)) \left\langle \Gamma_X^B \mathrm{Grad}\, g(X), \log_B Y - \log_B X \right\rangle$$

$$= \left\langle \Gamma_X^B \mathrm{Grad}(\phi \circ g)(X), \log_B Y - \log_B X \right\rangle$$

where the last equality is from the chain rule and linearity of the parallel transport and gradient. This shows $\phi \circ g$ is $\Gamma$-geodesically convex with arbitrary base $B$.

For $\psi$, first observe

$$g(Y) - g(X) = \left\langle \Gamma_X^B \mathrm{Grad}\, g(X), \log_B Y - \log_B X \right\rangle \leq \left\| \Gamma_X^B \mathrm{Grad}\, g(X) \right\| \left\| \log_B Y - \log_B X \right\|$$

$$\leq \|X\| \left\| \log_B Y - \log_B X \right\| = \mathrm{tr}(X^{-1} X X^{-1} X) \left\| \log_B Y - \log_B X \right\| = d \left\| \log_B Y - \log_B X \right\|.$$

Then, using the $L$-smoothness of $\psi$,

$$\psi \circ g(Y) - \psi \circ g(X) \leq \psi'(g(X))(g(Y) - g(X)) + \frac{L}{2} \|g(Y) - g(X)\|^2$$

$$\leq \left\langle \Gamma_X^B \mathrm{Grad}(\psi \circ g)(X), \log_B Y - \log_B X \right\rangle + \frac{Ld}{2} \left\| \log_B Y - \log_B X \right\|^2$$

which shows $\psi \circ g$ is $\Gamma$-geodesically $Ld$-smooth with arbitrary base $B$.

∎

Note affine invariant metric yields the complete Riemannian manifolds on $\mathrm{SPD}(d)$ [PFA05], so Proposition D.1 combined with Proposition 3.7 yields (3.1) with any base $b \in \mathrm{SPD}(d)$, hence guarantee Theorem 4.1. On the other hand, since $X \mapsto \log \det X$ is not $L$-smooth in matrix norm sense, Proposition D.1 highlights the strength of VTRGD algorithms for optimization problems involving functions of the form $f_\varphi(X) = \varphi(\log \det X)$.

Moreover, as in Wasserstein space, for this certain function VTRGD coincides with standard RGD.

**Proposition D.2.** *For the function of the form $f_\varphi(X) = \varphi(\log \det X)$, VTRGD algorithm coincides with standard RGD, i.e., for any $X, B \in \mathrm{SPD}(d)$,*

$$\exp_B(\log_B X - \Gamma_X^B \mathrm{Grad}\, f_\varphi(X)) = \exp_X(\mathrm{Grad}\, f_\varphi(X)).$$

*Proof.* We directly compute (4.1) and compare. We write $B$ as the base point. First, from the calculations in the proof of Proposition D.1,

$$\Gamma_{X_n}^B \mathrm{Grad}\, f_\varphi(X_n) = \Gamma_{X_n}^B \varphi'(\log \det X_n) \mathrm{Grad}(\log \det X_n) = \varphi'(\log \det X_n) \Gamma_{X_n}^B X = \varphi'(\log \det X_n) B.$$

Hence,

$$\exp_B\left(\log_B X_n - \eta_n \Gamma_{X_n}^B \mathrm{Grad}\, \log \det X_n\right)$$

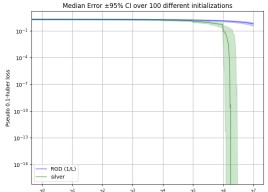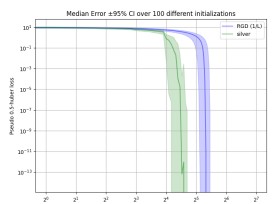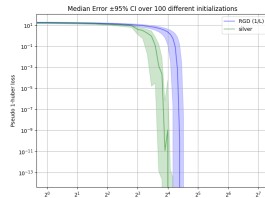

Figure 8: Comparison between silver stepsize method and RGD for entropy matching under $\delta$-pseudo Huber loss on SPD(50). We set $n = 2^7 - 1$ as the total iteration number, and conducted 100 simulations under different seeds and initializations. **Left:** $\delta = 0.1$. **Middle:** $\delta = 0.5$. **Right:** $\delta = 1$.

$$= B^{1/2}\exp\left(B^{-1/2}[\log_B X_n - \eta_n\varphi'(\log\det X_n)B]B^{-1/2}\right)B^{1/2}$$

$$= e^{-\eta_n\varphi'(\log\det X_n)}B^{1/2}\exp\left(B^{-1/2}\log_B X_n B^{-1/2}\right)B^{1/2}$$

$$= e^{-\eta_n\varphi'(\log\det X_n)}X_n = \exp_{X_n}(-\eta_n\varphi'(\log\det X_n)X_n) = \exp_{X_n}(-\eta_n\,\mathrm{Grad}\,f_\varphi(X_n)).$$

∎

The function $X \mapsto \log\det X$ can be considered as the entropy of the zero-mean Gaussian distributions (up to affine transform). Hence, functionals of the form $\varphi(\log\det\cdot)$ appears in many practical applications, such as entropy regularization.

**Entropy matching**    For the numerical experiment we consider the problem

$$f_\delta(X) = \delta^2\left(\sqrt{1 + \frac{(\log\det X - \tau)^2}{\delta^2}} - 1\right)$$

which corresponds to the entropy-matching problem with target value $\tau$ under the pseudo-Huber loss. We adopt the pseudo-Huber loss to examine how the algorithm's behavior depends on the curvature of the objective function (small $\delta$ yields a flatter function). Since pseudo-Huber loss is convex and 1-smooth, by Proposition D.1 $f(A)$ is generalized geodesically convex and generalized geodesically $d$-smooth. In addition, by Proposition D.2 we can apply silver stepsize directly to standard RGD. We conduced 100 experiments with $\tau = 1$ under different initializations, to verify our algorithm's stability. The results are summarized in Figure 8.

We additional provide two more benchmark problems on SPD($d$) space. The Fréchet mean estimation problem and Gaussian mixture model problem. Note for these general problem we do not have Proposition D.2, so we again need to choose the base for VTRGD (4.1). We again chose $b = I$ for both experiments.

**Fréchet mean estimation**    Fréchet mean estimation problem on the SPD space is widely studied problem for both theoretically interesting properties and practical application. Practically, under Affine invariant metric, Fréchet mean becomes geometric mean [FAP+05, PFA05] and therefore has many applications. Theoretically the Fréchet mean over SPD matrices is again SPD, so unlike typical matrix norm, this geometry is favorable when SPD constraint has to be involved when obtaining the means over SPD matrices; such constraint is common in covariance estimation problem [KPB25]. Hence, this problem is one of the standard benchmark for RGD methods [AOBL21, KY22]. The Fréchet mean estimation problem can be written for general manifold $M$ as follows: for given $\{p_1, \ldots, p_n\} \subset M$, the empirical Fréchet mean over $p_i$ is

$$p_* := \operatorname*{argmin}_{x \in M} f(x) := \frac{1}{2}\sum_{i=1}^{n} d^2(x, p_i).$$

This problem is known to be geodesically $n$-strongly convex, but not geodesically $L$-smooth. While the squared distance function $x \mapsto d^2(x, p)/2$ is generalized geodesically 1-smooth with base $p$, as we discussed in Section 3, by summing these together there is no single base that makes $f$ to

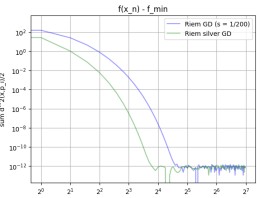 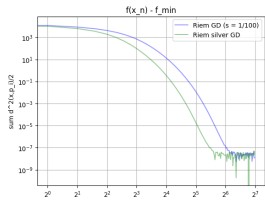

Figure 9: Optimization of Barycenter problem. We plot $f(x) - f_{\min}$, where $f_{\min}$ is the minimum value over all experiments. We set $b = I$ for VTRGD (4.1). **Left:** $n = 100$ and $d = 10$. We set $L = 200$ which guaranteed the stability. **Right:** $n = 10$ and $d = 100$. We set $L = 100$ which again guaranteed the stability.

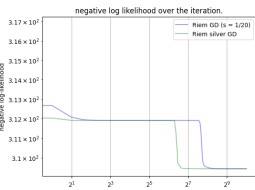 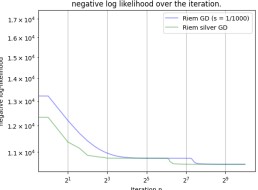

Figure 10: Optimization of Gaussian mixture model fitting. We set $b = I$ for VTRGD (4.1). **Left:** Simple setting, where $n = 100, K = 3, d = 2, L = 20$. **Right:** Complicated setting, where $n = 1000, K = 5, d = 5, L = 1000$. The $L$ is chosen to guarantee the numerical stability.

be generalized geodesically $L$-smooth with base $b$. Hence, neither RGD or VTRGD guarantee the theoretical convergence for this problem. Still, we found out that our algorithm numerically outperforms the standard RGD for this problem.

With $M = \mathrm{SPD}(d)$, We considered two settings: $(n, d) = (100, 10)$, and reversely $(n, d) = (10, 100)$, to observe whether dimension or sample size matters. Again we chose $b = I$ for VTRGD (4.1). We observed consistent strength of our method. The results are summarized in Figure 9.

**Gaussian mixture model**    Another practical application of Riemannian optimization on $\mathrm{SPD}(d)$ is minimization of the negative likelihood in Gaussian mixture model, which is a classical problem in Statistics. [HS20] proposed the reformulated objective function for solving Gaussian mixture model fitting, which coincides to classical negative log likelihood minimization at the minima, and allows the use of Riemannian optimization algorithm [HS20, HMJG21, SBS21]. For given observations $y_i = (x_i, 1)^T \in \mathbb{R}^d$ for $i = 1, \ldots, n$, the problem is formulated as follows:

$$\operatorname*{argmin}_{\theta = (\{w_j\}_{j=1,\ldots,K}, \{S_j\}_{j=1,\ldots,K})} \ell(\theta) = -\sum_{i=1}^{n} \log \left( \sum_{j=1}^{K} \frac{e^{w_j}}{\sum_{l=1}^{K} e^{w_l}} q_N(y_i; S_j) \right)$$

where

$$q_N(y_i; S_j) = \frac{\exp\left(\frac{1}{2}(1 - y_i^T S_j^{-1} y_i)\right)}{\sqrt{(2\pi)^d \det S_j}}.$$

This problem can be considered as the product manifold $(\mathbb{R}^d)^K \times (\mathrm{SPD}(d))^K$. One can conduct usual Euclidean gradient descent for vectors $w_j$'s, and conduct Riemannian methods in $S_j$'s. Again, this problem is not geodesically $L$-smooth, so neither our algorithm nor standard RGD do not allow the theoretical guarantee. Still, we found out that numerically our algorithm turned out to be useful for this problem.

We considered two setting: simple setting ($n = 100, d = 2, K = 3, L = 20$) and complicated setting ($n = 1000, d = 5, K = 5, L = 1000$). The quantities of $L$ are chosen again to guarantee the stability of the algorithms. Again, we set $b = I$ for VTRGD (4.1). The results are aggregated in Figure 10.

# E    Changes from the Submitted Version

In the submitted and reviewed version of our paper, we subsequently identified two errors. Below we document the resulting revisions as transparently as possible.

**Error I: Algebraic misstep**    We originally claimed that standard RGD (1.2) with silver stepsize achieves the $O(n^{-\log_2 \rho})$ rate of convergence, when the manifold is non-negatively curved and the function satisfies (3.1) with $b = x_j$. However, we identified an algebraic misstep when proving the induction step (Lemma 5.4 in the reviewed version)[5]. To address this issue, we made the following changes:

1. **Algorithm modification**. We replace standard RGD (1.2) by VTRGD (4.1). In contrast to standard RGD, VTRGD is defined relative to a specified vector transport, to which the algorithm and assumptions are tied.

2. **Scope**. The revised analysis no longer requires nonnegative curvature; it applies under vector transport-aware assumptions on the objective (3.1). Heuristically, the curvature requirement can be seen as transferred to vector transport-indexed inequality conditions.

3. **Function class and assumptions**. The admissible function class is now tied to the chosen vector transport via (3.1); a sufficient condition appears in Proposition 3.7. For the canonical choice $\mathcal{VT} = \Gamma$ (parallel transport), this replaces *geodesic L-smoothness* by *generalized geodesic L-smoothness with a single base $b$*. These conditions are incomparable; for instance, as discussed in Example C.10, the function $x \mapsto d^2(x, b)$ on a Hadamard manifold satisfies the new but not the previous condition.

4. **Numerical experiments**: We replace the experiments on Rayleigh quotient maximization problem by experiments on benchmarks over symmetric positive definite (SPD) matrices, to include a nonpositively curved setting where the revised analysis applies (Section 5, Appendix D.2.3). In this setting we also identify an additional function class satisfying our assumptions (Propositions D.1, D.2), which did not appear in the previous setup.

**Error II: regularity condition**    In Section 5.1, we noticed that we overlooked the *geodesic incompleteness* of 2-Wasserstein space: along discrete updates, the update may leave the class of absolutely continuous densities, (e.g., for large step sizes the update map may not be the gradient of a convex potential). Consequently, Corollary 5.1 requires a regularity condition ensuring that the iterates admit densities. In addition, the gradient update may not be the optimal transport map. We clarified these points in Remark 5.2, 5.3 and provide sufficient conditions relevant to our applications.

**What remains unchanged or strengthened**    Despite these changes, some core points remain intact or are further reinforced by the revision.

1. Our central application, potential function optimization on the 2-Wasserstein space, remains intact. In this setting, VTRGD with canonical vector transport $\mathcal{VT}_b^\mu = T_{b,\mu}$ recovers the standard Wasserstein gradient descent. Particularly, the potential energy satisfies the modified assumptions (Proposition 5.4). Thus, our main claim, the first acceleration result for Wasserstein gradient descent in Wasserstein space, still holds.

2. The Riemannian co-coercivity inequality, identified during the rebuttal as a key technical component, is recovered as the special case of Proposition 3.7 when $b = y$; see Appendix B.1.1.

3. Allowing negative curvature broadens applicability (e.g., the log-determinant objective on SPD manifolds verified in Proposition D.1, D.2).

---

[5]The error is in the last paragraph on p. 33 of the reviewed version, when subtracting $A$. We include the reviewed version in the supplementary materials.

**Summary**    The revision corrects the errors via an algorithmic modification (RGD (1.2) $\rightarrow$ VTRGD (4.1)) and additional regularity conditions (Corollary 5.1). The revised algorithm VTRGD is *not* the standard RGD, and the updated assumptions are neither uniformly stronger nor weaker than those in the original version. However, in some central applications VTRGD *coincides* with RGD, and those applications satisfy the newly introduced assumptions. Crucially, VTRGD coincides with standard RGD in our main application, the Wasserstein space. Hence, one of our main claims, the first acceleration result on Wasserstein gradient descent, remains valid, and the revision additionally includes a log-determinant example on the SPD space, which was not covered before.
