# OpenReview forum: "Acceleration via silver step-size on Riemannian manifolds with applications to Wasserstein space"
_NeurIPS.cc/2025/Conference — NeurIPS 2025 poster_

### Official Review · Reviewer_zbh5 · 2025-06-18

**Clarity:** 3
**Significance:** 3
**Originality:** 3
**Rating:** 5
**Confidence:** 3

**Summary:**

This paper studies the silver step-sizes for Riemannian manifold optimization. Silver step-sizes are a recently proposed technique that can accelerate gradient descent without explicitly incorporating momentum terms. The authors extend this idea to the Riemannian setting and show that silver step-sizes can lead to improved worst-case complexity bounds for optimization on manifolds. The paper also discusses two applications of the proposed method: optimization on the 2-Wasserstein space and Rayleigh quotient maximization.

**Questions:**

The silver step-size was originally inspired by recent advances in the Performance Estimation Problem (PEP). In this paper, it is not clear whether PEP plays any role in the analysis or proof techniques. Could the authors clarify whether any part of the theoretical results, was informed or assisted by PEP? Is it possible to apply the PEP to the Riemannian setting considered in this paper, in order to derive a lower bound result?

**Ethical Concerns:**

["NO or VERY MINOR ethics concerns only"]

**Final Justification:**

The response is clear and helpful. The strengths and weaknesses of the paper are still clear and have been described. I maintain the previous assessment.

**Limitations:**

Yes. This paper raises some open questions.

**Paper Formatting Concerns:**

I noticed some equations are too long, such as (3.1)

In addition, some hyperlinks in this paper do not work.

**Quality:**

4

**Strengths And Weaknesses:**

This paper addresses an important and fundamental problem: how to accelerate gradient descent methods on Riemannian manifolds. The authors propose using the recently introduced silver step-size to achieve acceleration in this setting. Both Riemannian manifold optimization and gradient-based methods are important areas of research.

In terms of presentation, I appreciate the clarity with which the paper defines the necessary preliminaries, as well as the additional preliminaries included in the appendix. The exposition is generally clear and accessible. However, as someone not deeply familiar with Riemannian optimization, I am not in a good position to verify the technical correctness of the mathematical arguments.

Assuming the theoretical results are correct, I believe the contribution is significant. While the proposed step-size rule is directly borrowed from the Euclidean silver step-size literature, extending it to the Riemannian setting looks nontrivial. This generalization is meaningful, especially given that optimization on manifolds is less mature. From this perspective, the results presented in the paper are valuable.

One limitation is that, unlike in the Euclidean setting, the paper does not provide a corresponding lower bound to demonstrate the optimality of the obtained complexity result. Although this is unfortunate, it is understandable given the increased technical complexity of the manifold setting.

Another strength of the paper is the inclusion of two applications, in which the proposed method demonstrates improved empirical performance. Given that the practical value of the silver step-size in Euclidean settings is still valuable than its theoretical value, these positive experimental results are encouraging.

---

> ### Author Rebuttal · Authors · 2025-07-30
>
> We are grateful to the reviewer for their detailed feedback, which we will take into account when preparing the final version, if accepted.
>
> > Unlike in the Euclidean setting, the paper does not provide a corresponding lower bound to demonstrate the optimality of the obtained complexity result. Is it possible to apply the PEP to the Riemannian setting considered in this paper, in order to derive a lower bound result?
>
> First, to the best of our knowledge, the optimality of silver step-size is not verified even in Euclidean space yet [1] (although it is conjectured to be the optimal [2]).
>
> Second, the worst-case analysis in the Riemannian setting must account for both the function and the geometry of the manifold, unlike in the Euclidean case, where the analysis is function-centric. In particular, certain functions, such as the squared-distance function, are intrinsically tied to the geometric structure of the manifold [3]. These complexities make lower bound analysis substantially more challenging in the Riemannian context, even when using tools such as PEP. In particular, to use PEP, the key ingredient is the interpolating inequality. However, to the best of our knowledge, the only Riemannian version of the interpolating inequality we are aware of is Proposition 38 of [4], which requires to verify whether the condition holds for all $x \in M$ instead of $x = x_j$ for $j = 1, \dots, n$. Since such condition is hard to implement numerically, applying PEP to Riemannian setting does not seem direct at this stage (see also Introduction of [5]). We leave this as an open direction for future work.
>
> > Could the authors clarify whether any part of the theoretical results, was informed or assisted by PEP?
>
> In this work, we did not directly utilize the PEP, as we were already aware of the target inequality (Lemma 5.2) from the Euclidean studies. However, certainly our analysis is based on the original silver step-size paper [1] which was heavily informed by PEP.
>
> References
>
> [1] Altschuler et al. Acceleration by stepsize hedging II: Silver stepsize schedule for smooth convex optimization. Mathematical Programming, 2024.
>
> [2] Grimmer et al. Composing Optimized Stepsize Schedules for Gradient Descent. Arxiv 2024.
>
> [3] Alimisis et al. A Continuous-time Perspective for Modeling Acceleration in Riemannian Optimization. AISTATS 2020.
>
> [4] Criscitiello et al. Curvature and complexity: Better lower bounds for geodesically convex optimization. COLT 2023.
>
> [5] Hu et al. Extragradient Type Methods for Riemannian Variational Inequality Problems. AISTATS 2024.

---

> ### Comment · Reviewer_zbh5 · 2025-08-05
>
> I would like to thank the authors for the response. It was helpful. I have clearly described the strengths and weaknesses of the paper in the previous comments.

---

### Official Review · Reviewer_sSNV · 2025-06-30

**Clarity:** 3
**Significance:** 4
**Originality:** 4
**Rating:** 4
**Confidence:** 4

**Summary:**

This paper analyzes the Riemannian gradient descent method under a varying silver stepsize schedule~\cite{altschuler2024acceleration}, a strategy recently proposed and successfully applied in Euclidean spaces. The authors prove that, under a newly introduced notion of generalized geodesic convexity, the Riemannian gradient descent algorithm achieves an accelerated convergence rate on manifolds with non-negative curvature—consistent with the Euclidean case, i,e, $O(1/k^{\alpha})$, $\alpha \approx 1.2716$.

Moreover, by incorporating a restarting scheme, the authors extend the silver stepsize analysis to the setting of geodesically strongly convex functions, without requiring any modification to the stepsize itself.

As an application, the paper develops a theoretically accelerated gradient optimization method on Wasserstein spaces. Numerical experiments further demonstrate the effectiveness of the variable stepsize strategy on the Rayleigh quotient maximization problem over the sphere manifold.

**Questions:**

* Can the assumption of generalized geodesic convexity be relaxed to standard geodesic convexity?
* What challenges arise in establishing convergence rate estimates when $n\neq 2^{k}-1$?
* Is it possible to achieve acceleration with silver step-size for geodesically $\alpha$-strongly convex functions without employing a restarting strategy?

**Ethical Concerns:**

["NO or VERY MINOR ethics concerns only"]

**Final Justification:**

I will keep my positive score, and my questions are all answered.

**Limitations:**

The present analysis is confined to manifolds with non-negative curvature and has not been extended to general manifolds.
Additionally, the method relies on the exponential and logarithmic maps, which may limit its applicability to some manifold-structured problems.

**Paper Formatting Concerns:**

* The notation is occasionally inconsistent (e.g., switching between $\sigma_p$ and $\Sigma_{p}$ in Definition A.15).
* Some references (e.g., [AP24a, AP24b, AP24c]) lack publication details.
* The manuscript contains grammatical errors that affect the overall readability and clarity. For example, on page 6, line 195, the phrase “all gradient iterates...” should be corrected to “for all gradient iterates...”, and “Lemma” should be changed to “Lemmas.” Additionally, some other expressions are ambiguous or ungrammatical, which may cause confusion.
* On page 9, line 281, the dimension of the sphere $S^{d-1}$ should be $d - 1$, not $d$.
* I believe the conclusion of Lemma D.6 is correct; however, the proof of the zeroth-order criterion is not sufficiently rigorous. Specifically, differentiating both sides of the inequality $(1-t) f(x)+t f(y) \geq(f \circ \gamma)(t)$ with respect to $t$ does not guarantee that the inequality relation remains valid.
* The proof of Lemma 5.4 is not clear to me.
* Throughout the proofs in the appendix, there are multiple instances of missing or incorrect symbols. For example,

-- on page 24, the first formula below line 925 is missing the symbol $d t$;

-- on page 32,  the formula $\Gamma_{x_{*}}^{x_{1}}$ in the third line below line 1173 should be $\Gamma_{x_{1}}^{x_{*}}$;

-- in the last line of page 32, $\left\|\operatorname{Grad} f\left(x_{1}\right)\right\|_{1}^{2}$ should be $\left\|\operatorname{Grad} f\left(x_{1}\right)\right\|_{x_{1}}^{2}$;

-- in Lemma C.3, $+2\left\langle T_{\mu, \nu}-i d, T_{\mu, \pi}-i d\right\rangle_{\mu}$ should be $-2\left\langle T_{\mu, \nu}-i d, T_{\mu, \pi}-i d\right\rangle_{\mu}$;

-- on page 42, in line 1423, $1+\rho^{\log _{\rho} \kappa}$ should be $1+\rho^{\log _{\rho} \kappa +1}$;

-- Several subscript errors appear in the equations between lines 1443 and 1444.

**Quality:**

3

**Strengths And Weaknesses:**

Strengths:
* The paper introduces a new notion of generalized geodesic convexity and gives some examples.
* The proposed Riemannian gradient descent method with silver stepsize is theoretically accelerated under non-negatively curved Riemannian manifolds.
* The paper includes applications on the Wasserstein space and numerical experiments of Rayleigh quotient maximization on the sphere.

Weaknesses:
* The silver step-size schedule is not clearly explained.
* Several key theoretical arguments, including Lemma 5.4 and its proof, are not clearly presented.
* The study focuses on the exponential and logarithmic maps on the manifold, while omitting the consideration of the retraction map.

---

> ### Author Rebuttal · Authors · 2025-07-30
>
> We are grateful to the reviewer for their detailed feedback, which we will take into account when preparing the final version, if accepted.
>
> > The silver step-size schedule is not clearly explained.
>
> Silver step-size is constructed recursively, starting from $n = 1$ where $\eta_0 = \rho - 1$ for $\rho = 1+ \sqrt{2}$. Then, given the sequence of the step-sizes at $n$ number of iterations, $\eta_i$ for $i = 0, \dots, n-1$, one constructs the new sequence of the length $2n + 1$ by
>
> \begin{align*}
>     \eta_i^{new} &=\eta_i, \qquad i = 0, \dots, n-1,\\\\
>     \eta_n^{new} &= 1 + \rho^{k-1},\\\\
>     \eta_i^{new} &= \eta_{i-n-1}, \qquad i = n+1, \dots, 2n.
> \end{align*}
>
> Then, one conducts the gradient descent by
> $x_{i+1} = \exp_{x_i}(-\eta_i \nabla f(x_i))$
> We will add more details about the silver step-size in the revised version.
>
> > Several key theoretical arguments, including Lemma 5.4 and its proof, are not clearly presented.
>
> We will make our best effort to clarify the arguments in the revision. If the reviewer finds a specific part unclear, we would greatly appreciate it if the reviewer could point it out so that we can elaborate on it more thoroughly.
>
> > The study focuses on the exponential and logarithmic maps on the manifold, while omitting the consideration of the retraction map.
>
> It is true our analysis is performed on exponential map and logarithmic map, and this is due to the use of metric distortion lemma which require the exact exponential map and logarithmic map. For the retraction map, one can consider the similar error analysis as in [1], while we did not pursue this generality.
>
> > Can the assumption of generalized geodesic convexity be relaxed to standard geodesic convexity?
>
> The need of generalized geodesic convexity comes from deriving the key Equation (3.1). Therefore, avoiding the use of generalized geodesic convexity would depend on whether (a) one can derive Equation (3.1) without invoking the generalized geodesic convexity, or (b) one can prove the silver step-size acceleration without bringing Equation (3.1) type inequality.
>
> Regarding (a), there has been an attempt to derive a similar inequality under standard geodesic convexity and $L_0$-smoothness (Lemma 2.3. of [2]). However, this approach does not recover the exact inequality we need: First, a constant $L$ is of the form $L_0d^2(x_i,x_j)$, which depends on the specific points involved. Second, the structure of the subtractive term differs from that in Equation (3.1).
>
> Regarding (b), the current silver step-size analysis relies critically on the precise form of Equation (3.1) even in the Euclidean case. So generalized geodesic convexity appears necessary at this moment.
>
> Moreover, even in the Euclidean case, the proof of Equation (3.1) implicitly uses the notion of generalized geodesic convexity (which coincides with standard convexity in the Euclidean setting) by introducing an auxiliary third point $z$. In this regard, we believe that deriving (3.1) without invoking generalized geodesic convexity, if possible, would require fundamentally different techniques from those used in existing proofs.
>
> > What challenges arise in establishing convergence rate estimates when $n \neq 2^k - 1$? Is it possible to achieve acceleration with silver step-size for geodesically strongly convex functions without employing a restarting strategy?
>
> 1. *Under additionally geodesically strongly convex functional, one can establish the same convergence rate with respect to $n$ for general $n \neq \ell (2^{k^{\ast}} - 1)$.* One can proceed as follows: Consider $m = 2^{k^{\ast}} -  1$ number of inner-iterations and $\ell$ number of outer iteration, as in our restarting algorithm. Now, for general $ 0 < r < m$, observe the following bound:
>
> \begin{align*}
> d^2(x_{m\ell + r, x_*)} &\leq 2d^2(x_{m \ell + r - 1}, x_*) + 2 d^2(x_{m \ell + r -1}, x_{m \ell + r})\\\\
> &= 2d^2(x_{m \ell + r - 1}, x_*) + 2 \|\log_{x_{m\ell + r - 1}} x_{m\ell + r}\|^2\\\\
> &= 2d^2(x_{m \ell + r - 1}, x_*) +2 \eta_{r-1}^2 \|\nabla f(x_{m \ell + r - 1})\|^2\\\\
> &\leq 2(1 + L^2 \eta_{r-1}^2) d^2(x_{m \ell + r - 1}, x_*).
> \end{align*}
>
> Here, the first inequality is coming from triangular inequality, the second equality is coming from the definition of the  logarithmic map, and the last inequality comes from geodesic $L$-smoothness condition applied on $x_{m \ell + r - 1}$ and $x_*$.
>
> Inductively applying the above result, one gets
> $$d^2(x_{m\ell + r}, x_*) \leq 2 \prod_{i=0}^{r-1} (1 + L^2 \eta_{i}^2)d^2(x_{m \ell}, x_*)$$
>
> Now, observe $\prod_{i=0}^{r-1} (1 + L^2 \eta_{i}^2) \leq (1 + L^2(1 + \rho^{\lceil \log_{\rho} \kappa \rceil}))^{m}$, by the definition of the silver step-size, restarting scheme in Theorem 4.2, and $r < m$. This is just a constant with respect to $n := m\ell + r$, since $m = 2^{\lceil \log_{\rho} \kappa \rceil + 1} - 1$ is a fixed constant. And by Theorem 4.2 of our work, $d^2(x_{m \ell}, x_*) \leq \exp(-Cm\ell/\kappa^{\log_{\rho} 2})d^2(x_0, x_*)$. Thus,
> $$d^2(x_{m\ell + r}, x_*) \leq C_{\kappa} \exp(-Cm\ell/\kappa^{\log_{\rho} 2})d^2(x_0, x_*).$$
> This implies one still achieves the same $\exp(-Cn/\kappa^{\log_{\rho}2})$ convergence rate for any $m\ell < n < m(\ell + 1)$ for any $\ell \in \mathbb{N}$, and therefore for general $n$. We will elaborate this in detail in the revision.
>
> 2. On the other hand, for generalized geodesically convex case, we are currently unaware how to avoid such constraint. In the Euclidean setting, theoretical guarantees for general $n \neq 2^k-1$ are derived from the analysis of the silver step-size under strong convexity [3] without the use of restarting method. However, interpretation of the co-coercivity condition used in [3] does not readily extend to the Riemannian setting, unlike the role played by Proposition 3.8. As a result, we cannot directly apply the strong convexity-based analysis to generalize the theory for arbitrary $n \neq 2^k -1$ in Riemannian setup. That said, our numerical experiments show that the algorithm performs well even when $n \neq 2^k - 1$. This suggests that a generalization of the proof may be possible, though we do not currently have a complete answer. However, even in this case, we do not view the restriction $n = 2^k - 1$ as a critical drawback, as the number of iterations can be chosen freely in practice.
>
> > Paper Formatting Concerns
>
> We appreciate the reviewer for carefully reviewing the paper and catching the typos, notation errors, and grammar errors. If accepted, we will fix these issues when we prepare the camera-ready version.
>
> > I believe the conclusion of Lemma D.6 is correct but the proof of the zeroth-order criterion is not sufficiently rigorous.
>
> We appreciate the reviewer for noticing the missing detail. Your argument is correct, and the result holds due to the additional equality condition at $t = 0$, which was inadvertently omitted. Precisely, let $\phi(t) = (1-t) f(x) + tf(y) - (f \circ \gamma)(t)$. Then, $\phi(t) \geq 0$ and $\phi(0) = 0$ is the global minimizer. Therefore,
>
> $$
> 0 \leq \phi'(0^{+}) = \lim_{t \to 0+}\frac{\phi(t)}{t} = f(y) - f(x) - \lim_{t \to 0+}\frac{(f \circ \gamma)(t) - f(x)}{t} = f(y) - f(x) - \frac{d}{dt}\bigg\vert_{t=0}(f \circ \gamma)(t).
> $$
>
> From here, one can proceed the same as in Lemma D.6. We will revise the statement to clarify this point if the paper is accepted.
>
> References
>
> [1] Han et al. Riemannian Accelerated Gradient Methods via Extrapolation. AISTATS 2023.
>
> [2] Ansari-Önnestam et al. Adaptive Gradient Descent on Riemannian Manifolds with Nonnegative Curvature. Arxiv 2025.
>
> [3] Altschuler et al. Acceleration by Stepsize Hedging I: Multi-Step Descent and the Silver Stepsize Schedule. ACM, 2024.

---

> > ### Comment · Reviewer_sSNV · 2025-08-04
> >
> > Thank the authors for the responses. I would prefer to keep the current score.

---

### Official Review · Reviewer_Y9D4 · 2025-06-30

**Clarity:** 3
**Significance:** 3
**Originality:** 3
**Rating:** 4
**Confidence:** 4

**Summary:**

This paper investigates Riemannian Gradient Descent (RGD) with silver step sizes. It demonstrates that, for optimization problems defined on non-negatively curved Riemannian manifolds, where the objective function is generalized geodesically convex and geodesically smooth, vanilla RGD can achieve accelerated convergence when the step size sequence is carefully chosen—specifically, using silver step sizes. The derived convergence rate is approximately $\mathcal{O}(1/n^{1.27})$, improving upon the known $\mathcal{O}(1/n)$ rate for RGD with constant step sizes.

In the strongly convex setting, the paper shows that RGD with silver step sizes and restarts converges exponentially fast, similar to RGD with constant step sizes. However, it offers an improved dependence on the condition number: the convergence rate involves a coefficient scaling as $\kappa^{0.786}$, compared to $\kappa$ for constant step sizes.

Numerical experiments include minimizing potential energy in the Bures--Wasserstein space and Rayleigh quotient maximization on the sphere. The results confirm that RGD with silver step sizes achieves acceleration over its constant-step-size counterpart.

**Questions:**

# Some questions and suggestions:
- The Wasserstein space is not a bona fide Riemannian manifold. I wondered if the theorems 4.1 and 4.2 still applied.
- The first experiment in the Wasserstein space: maybe also consider a more challenging form of $V$ instead of a quadratic function.
- The second experiment on a sphere: why did not you choose another function that is generalized geodesically convex instead?

# Minor points:
- The second condition of Prop. 3.8: should add "it holds" before $z$, otherwise it looks a bit confusing.
- Line 85: A word should be added at the beginning of the sentence starting with $\exp_{x}(v)$.
- Line 90: a sentence should be added to introduce/characterize parallel transport.
- Equation (2.1): should say explicitly the square bracket represents concatenation.
- Line 104: $\eta_0$ should be $\eta^{(0)}$. In equations after line 104, all indices are off by one unit.

**Ethical Concerns:**

["NO or VERY MINOR ethics concerns only"]

**Final Justification:**

The authors have addressed my concerns, I therefore keep my score.

**Limitations:**

yes

**Quality:**

2

**Strengths And Weaknesses:**

# Strengths:

- The theoretical results are noteworthy. To the best of my knowledge, achieving acceleration on Riemannian manifolds remains an active area of research--primarily focused on extensions of Nesterov-style acceleration--where progress has been limited due to challenges posed by the metric distortion of curved spaces. In contrast, this work explores a complementary direction: demonstrating that acceleration can be attained solely through the careful design of step sizes, similar to recent developments in the Euclidean setting.


- The writing is of high quality.

# Weaknesses:

- The theory only guarantees for iteration counts of the form $n=2^k-1$.

- One of the main motivations of the paper is to relax the assumption on upper bound and lower bound of the space curvature -- where the authors argue that the Wasserstein space is such an example. However, when working with the Wasserstein space, **deterministic** RGD usually cannot apply (except for very special and rare examples like the pontential energy), even for the practically important case of  KL divergence. Therefore, the motivation is somewhat limited.

- The assumption of generalized geodesic convexity -- while being mild and standard in the Wasserstein space -- is strong and less standard in general Riemannian manifolds. Maybe this is the reason why this definition has not been introduced in the context of Riemannian optimization in the literature.

- The numerical experiments, while illustrative, are somewhat limited in scope, focusing only on the Bures--Wasserstein space and the sphere. Moreover, the paper does not include comparisons with Nesterov-style Riemannian Gradient Descent, which would provide a valuable benchmark. Finally, it would be informative to include the theoretical convergence curve $y = 1/x^{1.27}$ in the plots as a reference, to better contextualize the empirical performance.

---

> ### Author Rebuttal · Authors · 2025-07-30
>
> We are grateful to the reviewer for their detailed feedback, which we will take into account when preparing the final version, if accepted.
>
> > The theory only guarantees for iteration counts of the form $n=2^k-1$.
>
> 1. *In fact, under additional geodesic strong convexity, one can establish the same convergence rate with respect to $n$ for general $n \neq \ell (2^{k^{\ast}} - 1)$*. One can proceed as follows: Consider $m = 2^{k^{\ast}} -  1$ number of inner-iterations and $\ell$ number of outer iteration, as in our restarting algorithm. Now, for general $ 0 < r < m$, observe the following bound:
>
> \begin{align*}
> d^2(x_{m\ell + r, x_*)} &\leq 2d^2(x_{m \ell + r - 1}, x_*) + 2 d^2(x_{m \ell + r -1}, x_{m \ell + r})\\\\
> &= 2d^2(x_{m \ell + r - 1}, x_*) + 2 ||\log_{x_{m\ell + r - 1}} x_{m\ell + r}||^2\\\\
> &= 2d^2(x_{m \ell + r - 1}, x_*) +2 \eta_{r-1}^2 ||\nabla f(x_{m \ell + r - 1})||^2\\\\
> &\leq 2(1 + L^2 \eta_{r-1}^2) d^2(x_{m \ell + r - 1}, x_*).
> \end{align*}
>
> Here, the first inequality is coming from triangular inequality, the second equality is coming from the definition of the  logarithmic map, and the last inequality comes from geodesic $L$-smoothness condition applied on $x_{m \ell + r - 1}$ and $x_*$.
>
> Inductively applying the above result, one gets
> $$d^2(x_{m\ell + r}, x_*) \leq 2 \prod_{i=0}^{r-1} (1 + L^2 \eta_{i}^2)d^2(x_{m \ell}, x_*)$$
>
> Now, observe $\prod_{i=0}^{r-1} (1 + L^2 \eta_{i}^2) \leq (1 + L^2(1 + \rho^{\lceil \log_{\rho} \kappa \rceil}))^{m}$, by the definition of the silver step-size, restarting scheme in Theorem 4.2, and $r < m$. This is just a constant with respect to $n := m\ell + r$, since $m = 2^{\lceil \log_{\rho} \kappa \rceil + 1} - 1$ is a fixed constant. And by Theorem 4.2 of our work, $d^2(x_{m \ell}, x_*) \leq \exp(-Cm\ell/\kappa^{\log_{\rho} 2})d^2(x_0, x_*)$. Thus,
> $$d^2(x_{m\ell + r}, x_*) \leq C_{\kappa} \exp(-Cm\ell/\kappa^{\log_{\rho} 2})d^2(x_0, x_*).$$
> This implies one still achieves the same $\exp(-Cn/\kappa^{\log_{\rho}2})$ convergence rate for any $m\ell < n < m(\ell + 1)$ for any $\ell \in \mathbb{N}$, and therefore for general $n$. We will elaborate this in detail in the revision.
>
> 2. On the other hand, as the reviewer mentioned, for generalized geodesically convex case the theory is only valid for $n = 2^k -1$. In the Euclidean setting, theoretical guarantees for general $n \neq 2^k-1$ are derived from the analysis of the silver step-size under strong convexity [1]. However, interpretation of the co-coercivity condition used in [1] does not readily extend to the Riemannian setting, unlike the role played by Proposition 3.8. As a result, we cannot directly apply the strong convexity-based analysis to generalize the theory for arbitrary $n \neq 2^k -1$ in Riemannian setup. That said, our numerical experiments show that the algorithm performs well even when $n \neq 2^k - 1$. This suggests there is a possibility of a generalizing the proof for arbitrary $n$, though we do not currently have a complete answer. However, even in this case, the restriction $n = 2^k - 1$ is not severe, as the number of iteration is something one can choose freely, rather than some fixed parameter.
>
> > When working with the Wasserstein space, deterministic RGD usually cannot apply.
>
> We agree that in Wasserstein space, a Monte Carlo approximation of the updates are often used rather than a closed form expression [8,9]. That said, we would like to make three remarks:
>
> First, in the literature of stochastic optimization, the analysis of deterministic algorithm is typically a prerequisite. Since a theoretical analysis of the deterministic silver step-size method on Riemannian manifolds was previously lacking, we considered this an essential first step.
>
> Second, as the reviewer pointed out, in our main application, potential energy minimization, the deterministic form of the gradient is typically available, making our current formulation applicable.
>
> Third, the main challenge in the stochastic setting lies in controlling the variance. To do so, the common practice in Wasserstein gradient-based methods is to use Monte-Carlo approximation of the true gradient [8,9]. While omitted for clarity, our experiments show the algorithm remains effective with stochastic gradients given enough samples (e.g., 100 in our tests). If needed, we’re happy to include these results in the appendix for the camera-ready version. The interaction between large silver step-sizes and noisy/inexact gradients poses challenges for variance control (while maintaining the acceleration), even in Euclidean settings. We leave it to future work.
>
> > The assumption of generalized geodesic convexity is strong and less standard in general Riemannian manifolds. Maybe this is the reason why this definition has not been introduced in the context of Riemannian optimization in the literature.
>
> We acknowledge that generalized geodesic convexity may indeed be a strong and potentially restrictive condition in general Riemannian manifolds. However, we would like to make two points:
>
> First, Example D.8 shows the concept of generalized geodesic convexity is not vacuous on general Riemannian manifolds beyond Wasserstein space. The Bures–Wasserstein (BW) space, although rooted in Wasserstein geometry, is a genuine Riemannian manifold in its own right. Moreover, the space of symmetric positive definite (SPD) matrices is a Riemannian manifold, without a direct connection to Wasserstein geometry. These examples illustrate that the notion of generalized geodesic convexity is not limited to Wasserstein space and may be broadly applicable across general Riemannian manifolds.
>
> Second, the generalized geodesic convexity condition stems from the well-known Euclidean inequality (3.1), which has guided algorithm design in prior work [2,3,4]. In this regard, we thought extending this concept to Riemannian manifolds may offer a promising direction for developing analogous algorithms in the non-Euclidean setting.
>
> > The numerical experiments, while illustrative, are somewhat limited in scope, focusing only on the Bures--Wasserstein space and the sphere.
>
> *We also included the experiments on mean-field training of the two-layer neural network in Appendix E.2.3.*
>
> > The paper does not include comparisons with Nesterov-style Riemannian Gradient Descent, which would provide a valuable benchmark. Also, include theoretical convergence curve.
>
> Our primary objective in this work was to demonstrate acceleration relative to standard RGD, and for that reason, we thought including the comparison between Nesterov-type results might dilute the message. That said, if needed, we can include the comparison between Nesterov-type RGD, and the theoretical convergence curve in the final version.
>
> > The Wasserstein space is not a bona fide Riemannian manifold. I wondered if the theorems 4.1 and 4.2 still applied.
>
> Yes. As discussed in the proof of Corollary 6.1, the same argument applies by directly substituting the corresponding quantities for the Wasserstein space. These substitutions are based on Otto calculus and do not require the Wasserstein space to be a Riemannian manifold in the strict sense.
>
> > The first experiment in the Wasserstein space: maybe also consider a more challenging form of $V$ instead of a quadratic function.
>
> We used the quadratic potential energy since the target function $\mathbb{E}[V(X)]$ admits the closed-form expression. Note that even solving the quadratic potential problem is nontrivial due to the underlying Riemannian structure. Instead, *to demonstrate the broader applicability of our method, we also included an additional experiment in mean-field neural network training in Wasserstein space in Appendix E.2.3.* If the reviewer has any specific suggestion for $V$, we are open to include additional experiments in the final version.
>
> > The second experiment on a sphere: why did not you choose another function that is generalized geodesically convex instead?
>
> The main reason we chose this problem is because the Rayleigh quotient maximization problem is a widely recognized benchmark in the Riemannian optimization literature [5, 6]. In fact, even functions that just satisfy both geodesic convexity and geodesic smoothness are rare. For instance, on compact manifolds, globally geodesically convex functionals do not exist [7].
>
> > Points regarding the notations.
>
> We appreciate the reviewer for carefully noticing these typos. We will fix these typos when we prepare the camera-ready version.
>
> References
>
> [1] Altschuler et al. Acceleration by Stepsize Hedging I: Multi-Step Descent and the Silver Stepsize Schedule. ACM 2024.
>
> [2] Altschuler et al. Acceleration by stepsize hedging II: Silver stepsize schedule for smooth convex optimization. Mathematical Programming 2024.
>
> [3] Li et al. A simple uniformly optimal method without line search for convex optimization. ArXiv 2023.
>
> [4] Shugart et al. Negative Stepsizes Make Gradient-Descent-Ascent Converge. Arxiv 2025.
>
> [5] Alimisis et al. Momentum improves optimization on riemannian manifolds. AISTATS 2021.
>
> [6] Kim et al. Accelerated gradient methods for geodesically convex optimization: Tractable algorithms and convergence analysis. ICML 2022.
>
> [7] Jonathan W. Siegel. Accelerated Optimization with Orthogonality Constraints.
> Journal of Computational Mathematics, 2020.
>
> [8] Salim et al. The Wasserstein proximal gradient algorithm. NeurIPS 2020.
>
> [9] Bonet et al. Mirror and preconditioned gradient descent in wasserstein space. NeurIPS 2024.

---

> > ### Comment · Reviewer_Y9D4 · 2025-08-04
> >
> > Thanks for addressing my comments. I believe with these added points, the paper is in a good shape. I therefore maintain my score.

---

### Official Review · Reviewer_WD9n · 2025-07-03

**Clarity:** 4
**Significance:** 3
**Originality:** 2
**Rating:** 4
**Confidence:** 5

**Summary:**

This work adapts the recent "silver stepsize" analysis for Euclidean convex optimization to the Riemannian setting. In particular, the authors focus on optimization over geodesically convex subsets of non-negatively curved manifolds, and develop some sufficient machinery to be able to extend the analysis to the setting of curved geometry. The authors then discuss the specific case of optimization over the Bures-Wasserstein space as a an example (since it carries a Riemannian structure, modulo some details).

One of the claimed key technical contributions relies on a crucial assumption (on $Q_{ij}$) made by the authors.

**Questions:**

* One of the key technical assumptions is 3.1. The need for precisely the same inequality has come up several times over the years, and to my knowledge, previous authors left it as an open problem (I feel if one does not assume it, then by paying an extra factor of "L" one should be able to get around this assumption, but then the constants might be wrong; however, this feeling is not a theorem, so please don't view it as such), while the current authors put it in as an assumption. That said, it is useful to have identified sufficient conditions (such as generalized geodesic convexity) for this assumption to hold. The authors should discuss the implications and limitations of this assumption more thoroughly, ultimately, because this seems to be one of the few "truly Riemannian barriers" that are in the paper.

In fact, I think if the authors could prove inequality 3.1 without invoking their sufficient condition for some nontrivial settings, that would be a valuable contribution on its own, and would add to the depth and maturity of the current work.

* The numerical instability of the silver step size suggests that there may be a stronger rate for the "lim inf" hiding in there? Moreover, is this a problem that also plagues the Euclidean case? How to ameliorate that? Possibly, there is a Lyapunov function that the authors could exhibit, that is actually monotonically decreasing (for instance this Lyapunov function in the authors case might easily come from combining function values with a stepsize x tangent-space distance (because their generalized geodesic convexity relies on an object that relates to such a distance)

*  The hypersphere example, while not geodesically convex, is known to satisfy a Riemannian PL inequality. From Euclidean results, it is known that PL functions cannot be solved at an accelerated rate, so that applies to the Riemannian case too. However, it is still possible that for the sphere, one can bypass that type of lower bound. Might the authors have some thoughts about that?

**Ethical Concerns:**

["NO or VERY MINOR ethics concerns only"]

**Final Justification:**

The authors have worked hard to answer all the questions and concerns. I would have upgraded my score, had the paper had sufficient novelty. The positive score that I have for the paper is thanks to its technical achievements, but ultimately, for a higher score, I would be pushed more by strong novelty (and to the authors: novelty is a subjective aspect, so I admit that you may not necessarily agree with my assessment). Nevertheless, the paper does deserve to be accepted, and the authors have been very responsive during the rebuttal process to tackle all other concerns that were raised.

**Limitations:**

Yes.

**Paper Formatting Concerns:**

-

**Quality:**

3

**Strengths And Weaknesses:**

# Strengths
1. The topic studied seems relevant
2. The analysis is largely clean, and the authors point out clearly where and how they deal with the metric distortion.
3. The application presented is relevant  (though, to be a bit critical, one must admit that at  this point using barycenter problems to highlight the performance of a Riemannian method is like continuing to use the  "MNIST" dataset as a benchmark).
4. Localization of the metric distortion challenge to a condition like (3.1) -- more on this in the review below.

# Weaknesses

1. The adaptation from the Euclidean to the Riemannian case seems to not require any true heavy lifting (but this is not that big a weakness, as this view applies to a large body of Riemannian methods anyways).

2. The functions that may satisfy the "generalized geodesic convexity/smoothness" inequality seems small (though appendix D does provide some interesting examples)

3. The analysis is (strangely) limited to the nonnegative curvature case, and it would be valuable to see what happens for negative curvature.

4. The empirical results suggest intense oscillations / numerical instability of the Riemannian silver step method, without further control or discussion or amelioration, this behavior can easily render the current contribution to be primarily of theoretical value.

---

> ### Author Rebuttal · Authors · 2025-07-31
>
> We are grateful to the reviewer for their detailed feedback, which we will take into account when preparing the final version, if accepted.
>
> > Using barycenter problems to highlight the performance of a Riemannian method is like continuing to use the "MNIST" dataset as a benchmark.
>
> Our experiment addresses the Rayleigh quotient maximization problem, not the barycenter problem. It is a standard benchmark in Riemannian optimization [1, 2]. *We also included more complex problems, Bures-Wasserstein optmization (Section 6.1) and mean-field training of the neural network (Appendix E.2.3).*
>
> > The adaptation from the Euclidean to the Riemannian case seems to not require any true heavy lifting.
>
> We faced two major challenges:
>
> 1. Meaningful geometric interpretation of (3.1) is nontrivial. This required proving Proposition 3.8, which was possible by introducing the notion of generalized geodesic convexity.
>
> 2. A key challenge in extending gradient methods to Riemannian manifolds is the lack of closed-form expressions for displacements between non-consecutive iterates. In Euclidean space, one has $x_l - x_k = -\sum_{i=k}^{l-1}\eta_i \nabla f(x_i)$ for all $k < l$. Unlike constant/monotonically decreasing step-size gradient analyses (including the momentum), Euclidean silver step-size analyses employ this non-consecutive relationship heavily [4]. However, RGD only allows the accesses to consecutive terms like $\log_{x_k} x_{k+1}$, making it difficult to adapt Euclidean silver stepsize analyses directly. Our main contribution was showing that, perhaps surprisingly, pairwise relationships alone suffice to recover the desired guarantees, which we elaborated in Remark 5.5.
>
> > The functions that may satisfy the generalized geodesic convexity/smoothness inequality seems small.
>
> While this condition is new and few functions are known to satisfy it, as the reviewer mentioned *we show in Appendix D.1 that it's non-vacuous with examples in popular spaces like Wasserstein, Bures–Wasserstein, and SPD matrices, along with verifiable criteria (Lemma D.6) and concrete examples (e.g., Example D.8).* The fact that it is not vacuous suggests that it is worth further investigation. We believe that continued study will help clarify the scope of this condition.
>
> > The analysis is (strangely) limited to the nonnegative curvature case, and it would be valuable to see what happens for negative curvature.
>
> Our proof does not readily extend to negative curvature case. In this case, the metric distortion inequality takes the form
> $$
>    \\|\log_{x_{n+1}}x_*\\|^2 \leq  \\|\log_{x_n}x_*\\|^2 + \zeta \\|\log_{x_n}x_{n+1}\\|^2 - 2\langle\log_{x_n}x_{n+1}, \log_{x_n}x_* \rangle.
> $$
> for some curvature-dependent constant $\zeta > 1$ (Lemma 5.2, [2]). When the curvature is non-negative, $\zeta = 1$.
>
> Naive application of our technique boils down to showing:
>
> $$ \frac{f(x_*) - f(x_k)}{r_{k}} - \frac{1}{4r_{k}^2}\\|\nabla f(x_n)\\|^2 - \frac{1}{r_{k}}\langle\log_{x_n}x_*,\nabla f(x_n)\rangle - \zeta \sum_{i=0}^{n-1}\eta_{i}^{2} \\|\nabla f(x_i)\\|^2 - 2\sum_{i=0}^{n-1}\eta_{i}\langle \nabla f(x_i), \log_{x_i}x_*\rangle  \geq \sum_{i,j = 0,\dots, *}\lambda_{ij}Q_{ij} \geq 0. $$
>
> For this to hold, $\eta_i$ or/and $r_k$ and $\lambda_{ij}$ should depend appropriately on $\zeta$, which poses two challenges if $\zeta > 1$.
>
> 1. Due to dependencies between $r_k, \eta_i, \lambda_{ij}$, and $Q_{ij}$, adjusting these parameters jointly is a much harder problem.
>
> 2. More importantly, since (5.1) is an inequality (unlike the equality in Euclidean case), coefficient matching technique used in Euclidean case fails, and proving $\lambda_{ij} \geq 0$ becomes harder. Our proof detoured this problem by showing that the nonzero coefficients actually coincide to the Euclidean ones. But when extending to negative curvature, since $\lambda_{ij}$ changes, such technique no longer works.
>
> Since there are interesting non-negatively curved manifolds with broad applications, e.g., Wasserstein and Bures-Wasserstein space, we stayed with the non-negatively curved manifolds for this work. We defer the extension to negatively curved space to future work.
>
> > The empirical results suggest intense oscillations of the Riemannian silver step method ... render the current contribution to be primarily of theoretical value. ... is this a problem that also plagues the Euclidean case? How to ameliorate that?
>
> *The oscillation is expected under silver step-size scheme that involves occasional large step-sizes, as this step-size is designed to guarantee multi-step descent, rather than monotonic decrease in function value that one observes in gradient descent with constant/monotonically decreasing step-sizes (see [8]).* Similar behavior is observed in neural network training with large step-sizes as well [10].
>
> On the other hand, *our experiments consistently show that the last iterates remain stable across multiple runs (see Figure 4) as the theory suggested*, illustrating statistical robustness of the algorithm.
>
> Lastly, while [4] does not provide the experimental result for $n \neq 2^k -1$, we observe similar oscillations in our experiments in Euclidean case too.
>
> > The need for the same inequality (3.1) has come up several times ... previous authors left it as an open problem.
>
> In Proposition 3.8, we provided the sufficient condition for the inequality (3.1). Other than Proposition 3.8, we were unaware of study of this inequality in Riemannian setting (except the similar one in Lemma 2.3 of [3], but it is not the same) in any other works. We would be grateful if the reviewer can provide the reference.
>
> > By paying an extra factor of $L$ one should be able to get around generalized geodesic convexity assumption.
>
> With only geodesic convexity and $L_0$-smoothness, a similar inequality can be obtained as shown in Lemma 2.3 of [3], but this is not precisely the same inequality needed. First, the constant $L = L_0d^2(x_i,x_j)$, now is not global and depends on the iterates. Second, the subtractive term is different from that in Equation (3.1). Since the precise form of Equation (3.1) is crucial for our proof, generalized geodesic convexity appears necessary at this moment.
>
> > The authors should discuss the implications and limitations of Equation (3.1) more thoroughly.
>
> We interpret (3.1) via Proposition 3.8, as generalized geodesic convexity and geodesic smoothness. Geodesic smoothness is a standard concept, and we provide interpretations with examples in Appendix D.1 to illustrate generalized geodesic convexity.
>
> > If the authors could prove inequality 3.1 without invoking their sufficient condition for some nontrivial settings, that would be a valuable contribution on its own.
>
> Even in the Euclidean case, the proof of Equation (3.1) implicitly uses the notion of generalized geodesic convexity by introducing an auxiliary third point $z$. In this regard, we believe that deriving (3.1) without invoking generalized geodesic convexity would require fundamentally different techniques from those used in existing proofs.
>
> > Is there a possibility of a Lyapunov function type analysis?
>
> We did not pursue a Lyapunov functional-based argument in this paper. While it will be interesting to explore Lyapunov functions that capture multi-step function-value descent, we defer it to future work.
>
> > The hypersphere example is known to satisfy a Riemannian PL inequality.
>
> Rayleigh quotient maximization problem does not satisfy PL. Here the minimizer is $x_* = v_{\max}$, the eigenvector corresponding to the largest eigenvalue of $H$. The Riemannian gradient is $\nabla f(x) = (x^T Hx)x - Hx$. When  $x = v_k \neq v_{\max}$ is any eigenvector of $H$,  $\| \nabla f(x) \|_{x}^2=0 < 2\mu(f(x) - f(x_*))$, hence contradicting Riemannian PL inequality.
>
> > PL functions cannot be solved at an accelerated rate, so that applies to the Riemannian case too. However, it is still possible that for the sphere, one can bypass that type of lower bound. Might the authors have some thoughts about that?
>
> Firstly, since Rayleigh quotient problem does not satisfy Riemannian PL condition, the lower bound does not apply.
>
> Secondly, typically the lower bound analysis is obtained by the worst-case constructions. However, many functions do not exhibit worst-case behavior and may admit faster convergence rates than the lower bound suggests (e.g., Introduction of [9] comes up with one example).
>
> Thirdly, we are cautious about extending Euclidean lower bounds to specific Riemannian manifolds (e.g., the sphere). The metric distortion inequality (Lemma C.1), which is an equality in Euclidean space, can be a strict inequality on some non-negatively curved Riemannian manifolds. Thus, at least in principle, some Riemannian manifolds may allow potentially sharper convergence rates.
>
> References
>
> [1] Alimisis et.al. Momentum improves optimization on riemannian manifolds. AISTATS 2021.
>
> [2] Kim et.al. Accelerated gradient methods for geodesically convex optimization: Tractable algorithms and convergence analysis. ICML 2022.
>
> [3] Ansari-Önnestam et.al. Adaptive Gradient Descent on Riemannian Manifolds with Nonnegative Curvature. Arxiv  2025.
>
> [4] Altschuler et.al. Acceleration by stepsize hedging II: Silver stepsize schedule for smooth convex optimization. Mathematical Programming. 2024.
>
> [5] Li et.al. A simple uniformly optimal method without line search for convex optimization. ArXiv 2023.
>
> [6] Shugart et.al. Negative Stepsizes Make Gradient-Descent-Ascent Converge. Arxiv 2025.
>
> [7] Zhang et.al. Riemannian SVRG: Fast Stochastic Optimization on Riemannian Manifolds. NIPS 2016.
>
> [8] Altschuler et.al. Acceleration by Stepsize Hedging I: Multi-Step Descent and the Silver Stepsize Schedule. Journal of the ACM 2024.
>
> [9] Grimmer et.al. Beyond Minimax Optimality: A Subgame Perfect Gradient Method. Arxiv 2025.
>
> [10] Cohen et.al. Gradient descent on neural networks typically occurs at the edge of stability. ICLR 2021.

---

> > ### Comment · Reviewer_WD9n · 2025-08-05
> > **Update**
> >
> > The responses do address many of the questions that arose, but do not yet provide a decisive / clear discussion of Assumption 3.1. To my knowledge, the authors of [7] had raised this type of condition (though it is possible they don't have it written in a paper, but might have had in a presentation).
> >
> > Also, to be _pedantic_ -- if there are oscillations, one cannot claim that the last iterate remains stable, because that depends on when one chose to stop (and if one ends up choosing to stop at an upswing of the loss). Hence, more precise statements would be better appreciated.
> >
> > The Riemannian PL inequality for the sphere does not hold everywhere, but either with some randomness or with some other qualifications -- see the line of work by Alimisis et al where such properties and variants such as weak strong convexity or quadratic growth type of conditions are studied.

---

> > > ### Author Response · Authors · 2025-08-06
> > >
> > > We truly appreciate the reviewer's detailed feedback.
> > >
> > > > The responses do not yet provide a decisive / clear discussion of Assumption 3.1. To my knowledge, the authors of [7] had raised this type of condition.
> > >
> > > First of all, we were unable to find a similar formula in [7], at least in the paper.
> > >
> > > Second, we would like to clarify whether the reviewer is referring to Assumption 3.1 or Equation (3.1). If the question concerns Assumption 3.1, we believe that these assumptions are standard in the Riemannian optimization literature [1,2]. If the question concerns Equation (3.1), then as discussed in our earlier response, the best interpretation we can provide is through Proposition 3.8.
> > >
> > > We believe that the non-standard components in Proposition 3.8 are (i) the generalized geodesic convexity condition, and (ii) the condition on $z$. The latter simply requires that $N$ is sufficiently large; for instance, it is trivially satisfied when  $M = N$ (which is the case of our (Bures)-Wasserstein example).
> > >
> > > Thus, the main conceptual difficulty lies in the generalized geodesic convexity. We acknowledge that its interpretation is not straightforward, which is why we included an overview in Figure 1 (a visual representation of generalized geodesic convexity), Remark 3.5 (Geometric interpretation of generalized geodesic convexity), and Appendix D.1 (contains specific examples and Lemma D.6).
> > >
> > > In our view, **Lemma D.6, which states the zeroth-order and second-order condition for generalized geodesic convexity, provides the clearest interpretation of this notion**. We are open to moving Lemma D.6 into the main body of the paper in order to offer a more geometric understanding of generalized geodesic convexity for the reader.
> > >
> > > We believe we can answer your question better if you could specifically mention which part of Assumption 3.1 is unclear to you.
> > >
> > > > If there are oscillations, one cannot claim that the last iterate remains stable, because that depends on when one chose to stop.
> > >
> > > We sincerely appreciate your valuable feedback. Your observation is indeed correct, and our initial response was loosely stated (However, Theorem 4.1 is precise). Our intended message was that **if one chooses the number of iterations (which we did in our experiments) of the form $n=2^k-1$, the last iteration is stable**. The key here is that due to the fractal-like pattern of the silver step-size, we know apriori where such spikes will appear. So the number of iterations can easily be chosen by the practitioner of the form $n=2^k-1$ where the last iterate will always be stable and will satisfy the convergence rate stated in Theorem 1.
> > >
> > > Moreover, as discussed in our replies to Reviewers Y9D4 and sSNV, in the geodesically strongly convex case our theoretical guarantee holds for all $n \geq 2^{k_*} - 1$, not just for $n = \ell (2^k - 1)$ (as evidence by the leftmost plot of Figure 2).
> > >
> > > Lastly, this oscillation also appears in the Euclidean silver step-size algorithms as well. Notably, in the Euclidean case, this has been addressed by redesigning the long step-size for specific $n \neq 2^k - 1$ [11]. Similar adaptations may apply to our setting, though technical challenges remain. We leave this extension for future work.
> > >
> > > > The Riemannian PL inequality for the sphere does not hold everywhere, but either with some randomness or with some other qualifications -- see the line of work by Alimisis et al where such properties and variants such as weak strong convexity or quadratic growth type of conditions are studied.
> > >
> > > We truly appreciate the reviewer for bringing [12] to our attention, which we will cite in our final version. As the reviewer noted, the Rayleigh quotient maximization problem indeed satisfies the Riemannian PL condition in a probabilistic (Theorem 4 of [7]) or local sense (Proposition 4 of [12]).
> > >
> > > However, this does not affect our previous argument regarding the lower bound, which we elaborate in more detail below.
> > >
> > > As noted earlier in our response, a lower bound only applies to worst-case, pathological instances. Specific cases may still exhibit significantly faster convergence. For instance, [13] shows impossibility of acceleration of first-order methods in Euclidean space using a specific PL function (Eq. 18). However, this lower bound **does not rule out the possibility of acceleration for all PL functions**. Moreover, as discussed in our previous response, some Riemannian manifolds could, in principle, permit sharper rates than Euclidean space.
> > >
> > > We also want to mention that Alimisis et al. [1] themselves use the same experiment to illustrate the acceleration of RGD. This was in fact the reason we chose this experiment.
> > >
> > > References
> > >
> > > [11] Grimmer et al. Composing Optimized Stepsize Schedules for Gradient Descent. Arxiv 2024.
> > >
> > > [12] Alimisis et al. Distributed Principal Component Analysis with Limited Communication. NeurIPS 2021.
> > >
> > > [13] Yue et al. On the lower bound of minimizing polyak-łojasiewicz functions. COLT 2023.

---

> > > > ### Comment · Reviewer_WD9n · 2025-08-06
> > > > **Thanks**
> > > >
> > > > Thank you for your elaborate responses to the questions and concerns raised in the review.
> > > >
> > > > The condition that I meant is the one displayed as Equation (3.1), as a part of Assumption 3.1. I remember seeing this condition in a talk, but unfortunately cannot track down a published location right now. In case I find it somewhere, I'll mention it to the authors (so that you can have a more complete coverage of related work).
> > > >
> > > > I support the acceptance of this paper.

---

> > > > > ### Author Response · Authors · 2025-08-08
> > > > > **Thank you**
> > > > >
> > > > > Thank you very much for your support. We would greatly appreciate it if you would consider increasing your score. Please share any relevant literature that you think will improve the literature review and place our contribution better.

---

### Note · Authors · 2025-08-12

Dear ACs and Reviewers,

We thank the reviewers for their constructive feedback. We summarize our main contributions and key discussions below.

### Contributions

1. We introduced a sufficient condition for Equation (3.1), the key inequality, on Riemannian manifolds via generalized geodesic convexity (Proposition 3.8). (3.1) is widely used in Euclidean optimization, and, as Reviewer WD9n noted, its meaningful interpretation in Riemannian manifolds was previously unknown.

2. We extended silver step-size accelerated GD to non-negatively curved Riemannian manifolds. Our method avoids the curvature upper bound and diameter bounds required in previous accelerated RGDs.

3. Our method yields the first provable accelerated discretized algorithm in Wasserstein space, a non-negatively curved space without a curvature upper bound. All experiments--Wasserstein spaces (potential energy minimization, mean-field neural network training) and the Rayleigh quotient maximization problem on the sphere (a standard benchmark)--show consistent performance across repeated runs.

### Technical Difficulties

Two main obstacles arise when extending Euclidean silver step-size analysis to the Riemannian setting:

1. Deriving a meaningful geometric interpretation of (3.1), achieved in Proposition 3.8.

2. The absence of closed-form expressions for displacements between non-consecutive iterates in RGD, a key component in Euclidean silver step-size analysis. We showed pairwise relationships suffice to recover the guarantees (Remark 5.5).

### Responses to Key Feedback

> Proof restricted to $n = 2^k-1$.

We showed the rate holds for $n \neq \ell(2^k-1)$ in the strongly convex case (see our replies to Reviewer Y9D4 and sSNV). In the convex case, while the restriction remains, it is not significant since the number of iterations $n$ can be freely chosen.

> Generalized geodesic convexity seems restrictive.

While its full implication needs further study, examples (Appendix D.1) show this concept is non-vacuous in general Riemannian manifolds.

> Restricted to non-negatively curved manifolds.

Extension to negative curvature is nontrivial (see our reply to Reviewer WD9n). We focus here on the non-negative cases, which already have broad applications.

> Oscillations along trajectory.

These stem from occasional large step-sizes but are known apriori due to the silver step-size pattern. For $n = 2^k-1$ the last iterate is always stable and achieves the convergence rate in our theorems.

---

### Decision · Program_Chairs · 2025-09-17

**Decision:**

Accept (poster)

**Comment:**

The paper introduces a "silver step‑size" schedule for Riemannian gradient descent and shows that, on non‑negatively curved manifolds satisfying the newly defined generalized geodesic convexity and smoothness assumptions, this simple step‑size rule yields accelerated in the convex case and an improved condition‑number dependence in the strongly‑convex case. The technical contribution rests on "Proposition 3.8" (which has central to the discussion between authors and reviewers also), which provides a geometric interpretation that enables adapting the Euclidean silver‑step analysis to the curved setting although no closed‑form expressions for non‑consecutive iterates are available. The authors further demonstrate that their method provides the first provable accelerated algorithm various settings; concerns are limited to (i) the restriction to non‑negative curvature, (ii) the strength and unfamiliarity of the generalized geodesic convexity condition, (iii) modest experimental breadth, and (iv) lack of a lower‑bound optimality proof.  However it seems to be consensus that that these issues do not outweigh the solid theoretical advances and overall merit of the paper. I concur and recommend acceptance.